# jsmetrics v0.2.0: a Python package for metrics and algorithms used to identify or characterise atmospheric jet-streams.

Tom Keel[1, 2], Chris Brierley[1], and Tamsin Edwards[2]

[1]Department of Geography, University College London, Gower Street, London, UK
[2]Department of Geography, King's College London, 40 Bush House, London, UK

**Correspondence:** Tom Keel (thomas.keel.18@ucl.ac.uk)

**Abstract.** The underlying dynamics controlling the jet streams are complex, but it is expected that they will have an observable response to changes in the larger climatic system. A growing divergence in regional surface warming trends across the planet, which has been both observed and projected since the start of the $20^{th}$ century, has likely altered the thermodynamic relationships responsible for jet stream formation and control. Despite this, the exact movements and trends in the changes to the jet streams generally remain unclear and without consensus in the literature. The latest IPCC report highlighted that trends both within and between a variety of observational and modelling studies were inconsistent (Gulev et al., 2021; Lee et al., 2021). Trends in the jet streams were associated with *low* to *medium confidence*, especially in the Northern Hemisphere.

However, what is often overlooked in evaluating these trends is the confused message in the literature around how to first identify, and then characterise, the jet streams themselves. We classify the methods for characterising jet streams in the literature into three broad strategies: statistics that isolate individual values from the wind speed profile (*jet statistics*), methods for quantifying the sinuosity of the upper air (*waviness metrics*), and algorithms that identify a mask related to the coordinates of fast flowing wind throughout the horizontal and/or vertical plane (*jet core algorithms*). While each approach can capture particular characteristics and changes, they are subject to the spatial and temporal specifications of their definition. There is therefore value in using them in combination, to assess parametric and structural uncertainty, and to carry out sensitivity analysis. Here, we describe *jsmetrics* version 0.2.0, a new open-source Python 3 module with standardised versions of 17 metrics that have been used for jet stream characterisation. We demonstrate the application of this library with two case studies derived from ERA5 climate reanalysis data.

## 1 Introduction

Jet streams are instantaneous features of the Earth's general atmospheric circulation. They manifest as fast-flowing ribbons of air, usually found near the thermodynamic boundary between the troposphere and stratosphere — the tropopause (Vallis, 2019). As their features are chaotic and loosely defined at any given scale, there is no universal process to capture jet streams in data (see recent reviews in Maher et al., 2020; Bösiger et al., 2022). As such, many strategies have been adopted to capture aspects of the jet stream. Among the most commonly used approaches is to develop algorithms, indices, and statistics (here

known as metrics) which isolate and characterise regions in the atmosphere expected to be synonymous with jet streams within a given spatio-temporal scale. We divide these common approaches into three broad types:

1. *Jet statistics* — Statistics for isolating individual quantities synonymous with the jet stream from upper-level wind speed within a given time window (e.g. latitude, speed, width; Section 2.1);

2. *Waviness metrics* — Statistics and algorithms for determining the 'waviness' of upper-level mean flow within a given time window. These metrics only have meaning at an integrated global scale (Section 2.2);

3. *Jet core algorithms* — Methods that return a mask of coordinates related to the jet location, e.g., identifying the maximum wind speed throughout the horizontal and/or vertical plane within a given time window (Section 2.3).

The differences between these types of approaches could lead to confusion about the trends shown in the planet's jet streams across a range of modelling and observational studies. While the variety of metrics developed can be used to improve understanding of the interactions of the jet stream with other components of the climate system, we argue that any understanding is inherently methodology-dependent. As such, this has made it difficult to understand the past and future behaviours of jet streams.

Here, we aim to address the need for a method of combining and/or comparing the various methods for jet stream identification. The tool we introduce, *jsmetrics*, is an open-source Python 3 package built upon *xarray* that implements 17 existing metrics used for jet stream identification or characterisation. We first review the different metrics included with the package (Section 2), before discussing the design of the package (Section 3) and demonstrating an application (Section 4). We conclude by discussing further potential uses of the package and future directions for work on jet stream identification (Section 5).

## 1.1 Background

Although the identification of jet streams is dependent on the definition used, in general they can be characterised as strong localised winds within regions of the maximal thermal wind shear occurring where there are extreme temperature and pressure gradients (Vallis, 2019). The Earth's atmospheric circulation gives rise to two processes that develop strong thermal wind shear and therefore jet streams: *eddy-driven processes* (relating to the behaviour of transient eddies in the mid-latitudes; Held, 1975) and *thermally-driven processes* (relating to conservation of angular momentum at the poleward edge of the thermally-driven Hadley Cell; Held and Hou, 1980).

While eddy-driven processes tend to produce jet features that are deeper and more variable in their location and strength, thermally-driven processes produce jet features that are more shallow, narrow, and less latitudinally variable (Harnik et al., 2014; Lachmy and Harnik, 2014; Madonna et al., 2017; Menzel et al., 2019; Stendel et al., 2021). The position of thermally-driven processes is connected to the edges of the Hadley Cell it, although recent work suggests this only a loose connection (Menzel et al., 2019).

However, jet streams are often driven by a combination of both processes, so it is perhaps better to consider entirely eddy-driven or entirely thermally-driven jets as two ends of a spectrum (Lee and Kim, 2003; Manney et al., 2014; Spensberger and

Spengler, 2020; Spensberger et al., 2023). Tropospheric jet streams in observations often exist in "merged states", especially across the mid-latitudes (Stendel et al., 2021), but diagnostics included in this package are not yet able to disaggregate the two "primary" types of jets. As thermally-driven components of the jet streams may dominate wind speeds in the upper reaches of the troposphere, using metrics that isolate lower-level winds magnifies the relative presence of eddy-driven components, and this has been a common strategy for identifying these processes (see Section 2; Hallam et al., 2022; Spensberger et al., 2023). Deeper, eddy-driven jets might stretch from the top of the troposphere to the atmospheric boundary layer, and tend to be more barotropic (Held, 1975; Held and Hou, 1980; Madonna et al., 2017). The *jsmetrics* package, introduced in this paper, focuses exclusively on metrics for tropospheric jet streams.

Jet streams play an influential role in the climate system. They help control, modify, and drive pressure systems across the planet, and their features are often directly involved in the development of cold waves, heat waves, weather bombs and weather persistence. It is important that we are able to assess uncertainties involved in representing the jet streams in data, and further, to know how they are responding to climate change (Gulev et al., 2021; Lee et al., 2019). Understanding how jet streams operate between seasons, between phases in climate oscillations, and in response to human activities could enable projections about the regimes of (extreme) surface weather across timescales (Harnik et al., 2016; Manney and Hegglin, 2018; Cohen et al., 2021).

## 2 Strategies for characterising jet streams

Despite their importance to climate studies, features of jet streams are generally quite difficult to identify and characterise in data-space because they act in chaotic ways in the atmosphere (Barnes and Polvani, 2015; Peings et al., 2017). Any given metric, used in isolation, roots the understanding of the jet stream to a given context and within a given spatial and temporal frame (e.g., Manney et al., 2011; Woollings et al., 2018).

In general, the metrics included within the *jsmetrics* package have been developed in relative isolation from each other to answer specific questions about the jet streams' form, position, and/or trends over time and space. In this version of *jsmetrics* (v0.2.0), we include 17 methods from the literature. This initial set of metrics was included based on, first, their ease to implement into Python, and second, the frequency of their usage in the literature. In Section 1, we made a distinction between metrics in three broad categories, discussed in further detail in this section.

### 2.1 Jet statistics

Jet statistics is a group that broadly encompasses all statistics and indices that extract individual values from upper-air wind synonymous with features of jet streams and within a given time window and spatial reference. Most commonly, this includes metrics that extract a jet latitude (e.g. the latitude of maximum wind speed in a given spatial reference) and/or jet speed (maximum wind speed in a given spatial reference), but there are also methods for other characterisations such as jet width and jet depth. These metrics are generally not designed to capture individual events or general form in the jet such as troughs or ridges, but instead to capture the general climatological characteristics of a jet stream, such as its position and speed (e.g., Koch et al., 2006; Barton and Ellis, 2009; Rikus, 2018). As such, they are most useful for understanding the general regimes

**Table 1.** Jet statistics from the literature included in the *jsmetrics* package ($u$- and $v$- refer to the zonal and vertical wind components)

| Study | Variable(s) | Pressure (hPa) | Temporal | Method |
|---|---|---|---|---|
| Archer and Caldeira (2008) (AC08) | $u, v$ | 100-400 | Monthly | Mass-flux weighted mean latitude |
| Woollings et al. (2010) (W10) | $u$ | 700-925 | Daily | Low-pass then Fourier filter over max wind speed |
| Barnes and Polvani (2013) (BP13) | $u$ | 700-850 | Daily | Low-pass filter then quadratic interpolation |
| Grise and Polvani (2014) (GP14) | $u$ | 850 | Daily | Quadratic interpolation of max wind speed |
| Barnes and Polvani (2015) (BP15) [1] | $u$ | 700-925 | Daily | Fit a parabola around wind speed profile |
| Barnes and Simpson (2017) (BS17) | $u$ | 700 | 10-day average | Maximum wind speed |
| Bracegirdle et al. (2018) (B18) | $u$ | 850 | Annual & Seasonal | Cubic-spline interpolation of max wind speed |
| Ceppi et al. (2018) (C18) [2] | $u$ | 850 | Monthly | Centroid of wind speed profile |
| Zappa et al. (2018) (Z18) [2] | $u$ | 850 | Monthly | Extends Ceppi et al. (2018) |
| Kerr et al. (2020) (K20) [3] | $u$ | 500 | Daily | Smoothed max wind speed by longitude |

[1] adapted from Barnes and Polvani (2013); [2] extended to include jet speed in Screen et al. (2022) [3] adapted from Barnes and Fiore (2013).

of jet streams and so have been adopted to evaluate latitudinal shifts, slowing or speeding up of the jet as well as narrowing or widening of the jet stream's operating range (Martin, 2021; Hallam et al., 2022). In Table 1, we review the 10 jet latitude metrics from the literature that feature in the *jsmetrics* package.

Jet statistics (Table 1) have typically been developed for pressure levels relatively close to the surface (700-925 hPa) and primarily with one variable: the zonal component of wind ($u$). As thermally-driven components of the jet streams may dominate wind speeds in the upper reaches of the troposphere, using lower level winds, as these methods do, is mostly motivated by magnifying the relative presence of eddy-driven components (Hallam et al., 2022). Jets dominated by eddy-driven components tend to be more barotropic, so extend further down towards the surface than the shallower thermally-driven and more latitudinally fixed, subtropical jets (Held, 1975; Held and Hou, 1980; Madonna et al., 2017). However, by isolating lower-level winds, these methods may miss aspects of jet streams whose eddy-driven components do not extend throughout the atmospheric column within the method's given time window. They also do not capture behaviour near the level of maximum wind speed, nor the presence of multiple jet streams (Melamed-Turkish et al., 2018; Manney et al., 2021).

In each case, the jet statistics available in *jsmetrics* all centre around extracting individual quantities from upper-level wind to characterise the jet stream in a given temporal and spatial frame. Most commonly, this involves extracting 'latitude' and/or 'speed' quantities at the point of fastest zonal wind, either for an entire study region (all metrics expect K20), or by longitude (K20). While each jet statistics produces outputs that are directly comparable to each other, a degree of variation is provided by how each method achieves their outputs. Metrics from GP14, BS17, B18, C18, and Z18 use various smoothing functions (quadratic, cubic spline and centroid) to downscale the resolution for jet speed and latitude estimate (commonly to a resolution of 0.01 degrees). W10, BP13, and BS17 express jet latitude estimate as an anomaly from the seasonal cycle to distinguish seasonal modes of the jet latitude and their preferred positions over a study area.

**Table 2.** Jet waviness metrics from the literature included in the *jsmetrics* package ($u$- and $v$- refer to the zonal and vertical wind components; $zg$ refers to the gravity-adjusted geopotential height)

| Study | Variable(s) | Pressure-level (hPa) | Temporal | Method |
|---|---|---|---|---|
| Francis and Vavrus (2015) (FV15) | $u$, $v$ | 500 | Daily | Meridional circulation index |
| Cattiaux et al. (2016) (C16) | $zg$ | 500 | Daily | Sinuosity metric |

Each of the methodologies is relatively adjustable and fast to compute (compared to the other metrics in the package), so they can be used to produce quick diagnostics of fast-flowing wind over a given time period and region. Notably, these types of metrics have been employed mainly to evaluate shifts in position and speed of the jet streams at relatively longer time scales (intra-seasonal and interannual) to evaluate their response to changes in polar-tropical temperature gradients in a warming world (e.g. Barnes and Simpson, 2017; Zappa et al., 2018).

Approaching any day-to-day spatial variation shown in the jet stream with this form of metric is generally regarded to be limited (Koch et al., 2006; Rikus, 2018). And when considering that the jet streams are inherently 3-dimensional and multifaceted structures, it is restrictive to view wind speed at one isolated slice of the atmosphere (Strong and Davis, 2005, 2006). As such, jet latitude metrics are typically less useful for diagnosing trends in synoptic-scale events (Manney and Hegglin, 2018), such as cold-air outbreaks. Further, these metrics are developed to find a single-jet structure (one stream), so they are less appropriate for studying splitting and merging in the jet (Hallam et al., 2022).

## 2.2 Waviness metrics

Waviness metrics can be considered to be more derived methods that describe the general nature of the winds in the upper parts of the troposphere. They look to quantify waves, meridional excursions and/or sinuosity within the structure of a single global jet stream. They broadly describe propagation of Rossby waves in the structure of the upper-level mean flow, and they do not necessarily isolate which parts of the mean flow are 'jet streams', nor do they diagnose the eddy or thermal processes driving them (Martin, 2021). Two jet waviness metrics feature in the *jsmetrics* package: (1) Francis and Vavrus (2015) that calculates the Meridional Circulation Index by comparing the ratio of meridional wind component to total wind speed, and (2) Cattiaux et al. (2016) that calculates sinuosity by comparing the length of a geopotential height contour corresponding to the 500 hPa average over 30°-70°N to the 50°N latitude circle. We outline the two waviness metrics in Table 2.

These metrics consider the jet stream as a continuous pan-global feature, as opposed to a regional, split or emergent structure (Molnos et al., 2017; Martin, 2021). This conceptualisation is more observable in upper-air mean flow at seasonal and longer time aggregations (Koch et al., 2006; Spensberger et al., 2017). By framing the identification of jet streams as being about their propagation in Rossby waves, these metrics move towards diagnosing the propensity for peaks and troughs and thus can be used as a proxy to describe the poleward/equatorward transport of the underlying surface air masses (Hanna et al., 2017; Vavrus et al., 2017). Waviness metrics have been used to evaluate trends of jet stream flow in response to the warming world

**Table 3.** Jet core algorithms from the literature included in the *jsmetrics* package

| | Study | Variable(s) | Pressure-level (hPa) | Temporal | Method |
|---|---|---|---|---|---|
| | Koch et al. (2006) (K06) | $u, v$ | 100-400 | Daily | Event-based jet stream climatology and typology |
| | Archer and Caldeira (2008) (AC08) [1] | $u, v$ | 100-400 | Monthly | Mass-flux weighted wind-speed |
| | Schiemann et al. (2009) (S09) | $u, v$ | 100-500 | 6-hourly | Local maxima and above 30 $ms^{-1}$ |
| | Manney et al. (2011) (M11) [2] | $u, v$ | 100-400 | Daily | Wind speed maxima and jet core separation |
| | Pena-Ortiz et al. (2013) (PO13) | $u, v$ | 700-850 | Month-Yearly | Local wind maxima |
| | Kuang et al. (2014) (K14) [3] | $u, v$ | 200-250 | any | Jet occurrence and jet occurrence centres |

[1] also include a method for extracting the jet latitude statistic; [2] method refined in Manney and Hegglin (2018) to include physically-based method to distinguish subtropical and polar jets; [3] adapted from Ren et al. (2011).

and whether this has encouraged extreme weather (Francis and Vavrus, 2015; Hanna et al., 2017; Vavrus et al., 2017; Cohen et al., 2020). The notion that a 'wavier' jet stream leads to more extreme (winter) weather in response to the warming world is a highly contested topic (Francis, 2017; Manney and Hegglin, 2018; Cohen et al., 2020, 2021), but it is suggested that the slower progression of the jet stream in a 'wavier' regime encourages surface weather systems to take a longer path and broader across the planet's latitudes and as such encourage the transport of colder air to be pushed further equatorward and vice versa. Robust conclusions about changes in jet waviness so far have been difficult to establish due to variation in the region and years studied, as well as the methodology used (e.g. Barnes, 2013; Barnes and Simpson, 2017; Blackport et al., 2019; Blackport and Screen, 2020).

## 2.3 Jet core algorithms

Jet core algorithms are rule-based methods which return a mask of coordinates associated with jet streams in the upper-air wind throughout the horizontal and/or vertical plane. Their outputs consist of a multidimensional collection of points describing coordinates associated with the main body of the jet streams, known variously as 'jet cores', 'jet occurrences' or 'jet centres' (here we refer to them all as jet cores). Using these coordinates, it is possible to then mask/extract further dynamical information e.g. pressure, altitude, or speed at the locations of the jet cores. Most commonly, the jet core algorithms extract coordinates using wind-speed thresholds before applying further rule-based algorithms to classify the jet cores further (e.g. into types of jet core occurrence, into local maxima, into zonally continuous structures, etc.). We review the 6 jet core algorithms featured in the *jsmetrics* package in Table 3.

Typically, these algorithms are more computationally expensive than the other types of strategies outlined in this research. However, they provide relatively more detail about the features in the jet streams at a synoptic scale (Molnos et al., 2017; Kern et al., 2018). We note that, the implementations of SO9, M11, PO13, and K14 can provide three-dimensional outputs for each time step including altitude coordinates about the jet cores they extract, and K06 and ACO8 instead return mass-weighted output which provide two-dimensional jet cores for each time step.

The determination of jet cores varies between the algorithms, and they have been selected on: (i) predefined maximum speeds expected for jet streams (varying over 27-40 $ms^-1$; Koch et al., 2006; Strong and Davis, 2007; Schiemann et al., 2009; Manney et al., 2011; Pena-Ortiz et al., 2013; Kuang et al., 2014), (ii) in relation to wind-speeds of neighbouring data points (local wind-speed maxima; Schiemann et al., 2009; Manney et al., 2011; Pena-Ortiz et al., 2013; Kuang et al., 2014), or (iii) retaining continuity of a core across longitudes and/or pressure-levels (e.g. Molnos et al., 2017). By relying on defined wind-speed thresholds and local maxima, these methods can discount the influence of multiple streams of jet streams, i.e. if they are only selecting the 'maximum' jet speeds (Spensberger et al., 2017; Rikus, 2018). Furthermore, they may also underestimate positions of the jet cores in different seasons, in climate regimes different from the present (e.g. SSP5-8.5), and within different phases of the given climate oscillations (e.g. Woollings et al., 2010; Madonna et al., 2017; Manney and Hegglin, 2018; Manney et al., 2021). We expect jets to be faster and the eddy-driven and thermally-driven components to be more latitudinally separated in the winter versus summer, although this relationship also expresses significant regional variation (Manney and Hegglin, 2018; Maher et al., 2020; Manney et al., 2021).

Different processes are known to drive the jet streams that form over the planet (Ahrens and Henson, 2021), but, in the Northern Hemisphere especially, these processes are known to exist in combination and interact (Li and Wettstein, 2012; Madonna et al., 2017; Maher et al., 2020). Broadly, this has made it difficult to isolate the relationship between changes to the different processes driving jet streams and the patterns shown in upper-level wind conditions (Molnos et al., 2017; Manney and Hegglin, 2018; Hallam et al., 2022).

While there is no clear-cut method to separate eddy- and thermally-driven components of the jet stream (or the subtropical jet from the polar jet), some jet core algorithms make a consideration that the jet streams are driven by two mechanisms and attempt to separate them. K06 subdivide jet core 'events' by depth. PO13 develop a method based on latitude to distinguish between merged and separate states of the polar and subtropical jets after the initial detection of jet cores, but were only able to separate the Northern Hemisphere subtropical jet in Jan-Feb. The M11 method was extended by Manney and Hegglin (2018) by introducing a physical-based identification of the subtropical jet (based on the thermal tropopause altitude), to more robustly separate it from the polar jet. Manney et al. (2014) found that separating the M11 cores by a latitude criterion to be effective only at a climatological scale. Although not currently implemented in jsmetrics, Christenson et al. (2017) and Spensberger et al. (2023) propose methods which use the potential temperature of jet cores to distinguish eddy from thermally driven jets. Finally, we also note that are some methods that have been developed exclusively for the subtropical jet (see Maher et al. (2020) for a review of such methods), and envisage these could be incorporated in a future release of *jsmetrics*.

## 3 Description of jsmetrics

*jsmetrics* is a package containing implementations of various metrics and algorithms for identifying and/or characterising jet streams, written in Python 3. The package can be installed from the Python Package Index (PyPI) repository using `pip` and is also available on GitHub. *jsmetrics* is published under the GNU v3.0 licence.

The main focus of the package is to standardise the methods used to either characterise or identify jet streams in atmospheric data such that they can be compared with each other. The hope is that a tool allowing for this inter-compatibility would help the research community to both help quantify what different metrics show about jets as features of atmospheric circulation, but also to provide a platform for researchers to edit existing, and develop new, metrics and algorithms in a standardised framework. The design of this framework is discussed in this section, and there are more details about how to add new metrics to the package in section 5.1.

## 3.1 Design

The package is built using *xarray* — an open-source Python package for working with labelled multidimensional arrays that has become a popular package for Earth Science research (due in part to its ability to interface with NetCDF4 and GRIB data formats; Hoyer et al., 2023). As the package is built from *xarray*, each individual metric and algorithm in the package is stubborn about its inputs — only accepting an *xarray Dataset* or *DataArray* object as an input. Further, the inputs are expected to contain dimensions and variables with standardised names conforming to the 'controlled vocabulary' of Taylor et al. (2011) (e.g. ua, va, zg, plev, lat, lon). Whilst the current iteration of jsmetrics is only compatible with data with standard pressure levels (plev), for future development of the package, it is a priority to include compatibility with other vertical coordinate systems. The use of the standard inputs in this way allows the package to have a logical output, i.e., xarray dataset containing additional variables computed by the given jet stream metric.

The design philosophy of this package was to decompose and de-couple each metric and algorithm into a collection of base functions that each perform one specific part of the methodology, e.g. to calculate a climatology, calculate a zonal mean or extract cells with wind-speed matching given criteria. This design decision was taken to allow metrics to share components, potentially making subsequent metrics easier to verify and implement, and also to improve bug detection and traceability. The package is built such that existing metrics can be modified by replacing the statistical filtering method used, the wind speed threshold limit, or by tweaking the steps of an algorithm, for example. Unfortunately, this flexibility requires making all base functions as simple and one-use as possible, which has sometimes led to a decrease in readability. For example, it became necessary to keep some base functions more specific, which may make some of these harder to use without a familiarity with the package and/or more advanced Python knowledge. We hope to have alleviated any loss of readability, with the use of more verbose naming conventions throughout the package and detail in the individual method's docstring.

### 3.1.1 Flexibility

*jsmetrics* was designed in a way that does not predefine any sub-setting of input data or to be stubborn about receiving data of a given resolution, i.e., it can meet specifications defined by the various definitions of the methodologies of the metrics provided in the literature. Instead, the package passes the handling of sub-setting of the data onto the user. As such, each metric can be run on the same data without requiring sub-setting. In cases where not sub-setting is nonsensical (i.e., methods that can only run on one pressure level or require specific temporal or spatial resolution), then the user is notified. Because of this, each metric is flexible, so it is possible to change the resolution of the input data, the spatial region or the number of pressure levels

used. The motivation was to open up the possibility of sensitivity analysis with the metrics and the quantification of parametric uncertainty of the metrics. If there are any adjustments or difference between the literature's implementation and the (Python) implementation provided by *jsmetrics*, we have made a note of these in the metric's docstring under 'Notes', available in the online documentation (see Keel, 2023).

### 3.1.2 Package organisation

An aim in designing the layout of the *jsmetrics* package was to keep it well-organised, hierarchical, and easy to navigate. Also, to hide all the implementation-level detail of each metric within a function sharing the given metrics name. To achieve this, we break the package down into 3 main folders: *core* — containing all the main functions for the package, *metrics* — containing the implementations of the jet stream metrics, *utils* — containing scripts with utility functions for general data, spatial and wind related operations. We break down the *metrics* folder further in Table 4. Notably, during the process of designing this package, it became important to distinguish between three distinct types of methods described earlier; here stored in three files: *jet_statistics.py*, *waviness_metrics.py* and *jet_core_algorithms.py*. These files contain the instructions (functions) to calculate a given metric or algorithm at a high-level of abstraction. In each case, they call sub-functions in three component files, and these component files can call upon various utility functions available within the *utils* folder. The implementation details of each metric are kept intentionally hidden (and de-coupled from the metric itself) in sub-functions to allow for readability and also to allow for the construction of new metrics and/or edit the existing ones. Finally, the package also provides a specification file, *details_for_all_metrics.py*, which details all the data sub-setting needed to replicate the specification from which the method was built on, i.e. Woollings et al. 2010 was built from zonal wind speed (*ua*) data at 700-925 hPa between 15-75 degrees N and 120-180 degrees W. This file also provides a description of the metric including citation details.

### 3.2 Development

The process by which metrics have been added to the *jsmetrics* package is diagrammatically represented in Figure 1. This process applies to metrics already added to the package, but also serves as a guide for adding metrics in the future, with a code review on GitHub being an essential part of the development of this project. As shown, we break down the process into 4 successive stages (which we organise in GitHub as a kanban board under 'projects').

As shown in Figure 1, after the identification of a relevant metric (*Not started*), we first produce a pseudocode implementation on paper using the description of the method from the respective paper (*In progress*). After this, we translate the pseudocode to Python in Jupyter-Notebooks, where we refactor the code so that it runs as fast and independently as possible (with an emphasis on minimising third-party packages/libraries, i.e. using only NumPy, xarray and base Python). In this stage, we start to write documentation (docstrings) for each function and class, and plan unit tests for when the metric is moved over to *jsmetrics*. After writing the implementation, we validate its accuracy by reproducing the results from the given study where possible in stage 3 (*Undergoing validation*). After which we either debug the method further if it fails the validation, or write unit tests, finish the documentation, and integrate the metric into the *jsmetrics* package if it succeeds. As of version 0.2.0, ten jet statistic

**Table 4.** File layout of the *metrics* folder in the *jsmetrics* package.

| File | Purpose |
|---|---|
| *higher-level of abstraction* | |
| jet_core_algorithms.py | Stores all the instructions to run the jet core algorithms. |
| jet_statistics.py | Stores all the instructions to run the jet statistics. |
| waviness_metrics.py | Stores all the instructions to run the waviness metrics. |
| *lower-level of abstraction* | |
| jet_core_algorithms_components.py | Sub-functions for the jet core algorithms. |
| jet_statistics_components.py | Sub-functions for the jet statistics. |
| waviness_metrics_components.py | Sub-functions for the waviness metrics. |
| *specification file* | |
| details_for_all_metrics.py | Stores all the data sub-setting specifications and descriptions for each algorithm and metric |

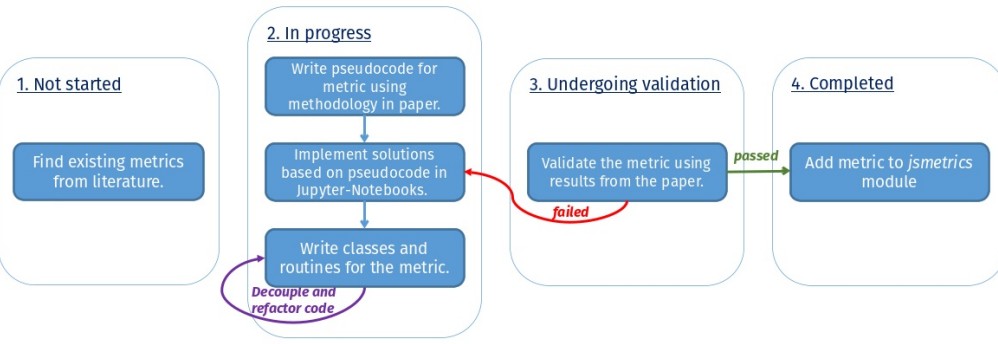

**Figure 1.** Stages involved with developing the *jsmetrics* package.

metrics, two jet waviness metrics and six jet core algorithms have been added to the package. We have detailed the progress status of each metric included, and this is available via ReadTheDocs (see Keel, 2023).

## 255  4   Application of jsmetrics to ERA5 reanalysis data

Having covered some key features of *jsmetrics*, the aim of this section is to introduce how to install the package and to demonstrate its application on a climate data set — here chosen to be the European Centre for Medium-Range Weather Forecast's ERA5 (Hersbach et al., 2020). For the demonstration here, a limited amount of knowledge about Python is needed to repli-

cate our results, as the *jsmetrics* package is built to be simple and user-friendly. For more advanced use of this package, we recommend some working knowledge of Python and *xarray*.

### 4.0.1 Experiment setup, installation, and input data

*jsmetrics* is compatible with Python version 3.7 or later and can be installed via PyPI using the command `pip install jsmetrics`. Installing via pip automatically collects and installs all the dependencies required for the package, but the source code is also accessible via GitHub. More detail about installing *jsmetrics* is provided in its documentation (https://jsmetrics.readthedocs.io/en/latest/, last accessed 5th December 2023). To introduce the features of the package, we look at two case studies using data from the ERA5 climate reanalysis (Hersbach et al., 2020), which we have accessed via the Climate Data Store API. We have provided a link to the scripts we used for extracting data from the Climate Data Store API in *data availability* at the end of this document.

### 4.1 Case study 1: Comparison of winter jet latitude and jet speed estimations

In this first case study, we use lower tropospheric *u*-component wind data (in m/s) from the ERA5 climate reanalysis to compare the daily latitudinal position of the jet stream over the North Atlantic, North Pacific, and Southern Hemisphere as determined by 8 jet statistics available in *jsmetrics* (Figure 2). The data are in NetCDF version 4.0 and consist of 1.0°by 1.0°global u-component wind speed for each winter day (DJF or JJA) between 1st January 1979 and 31st January 2022 at the pressure levels 700, 775, 850 and 925 hPa. In this figure, each violin plot is produced from 3912 (DJF) or 3956 (JJA) data points representing the daily winter day averages during this 43-year period. The thicker black line in the centre of each violin plot indicates the interquartile range, and the thinner line indicates the 95% confidence interval. The white dot represents the median and the shading which forms the body of each violin is a Kernel Density Estimation, with wider sections representing a higher probability of occurence. The latitude-longitude bounds of each region included in this figure are not consistently defined across the literature, and so we vary these according to each metric's respective study. We exclude two metrics from this section: AC08, as this method uses *v*-component wind speed, and K20, as the methodology does not specifically look at any of these three regions.

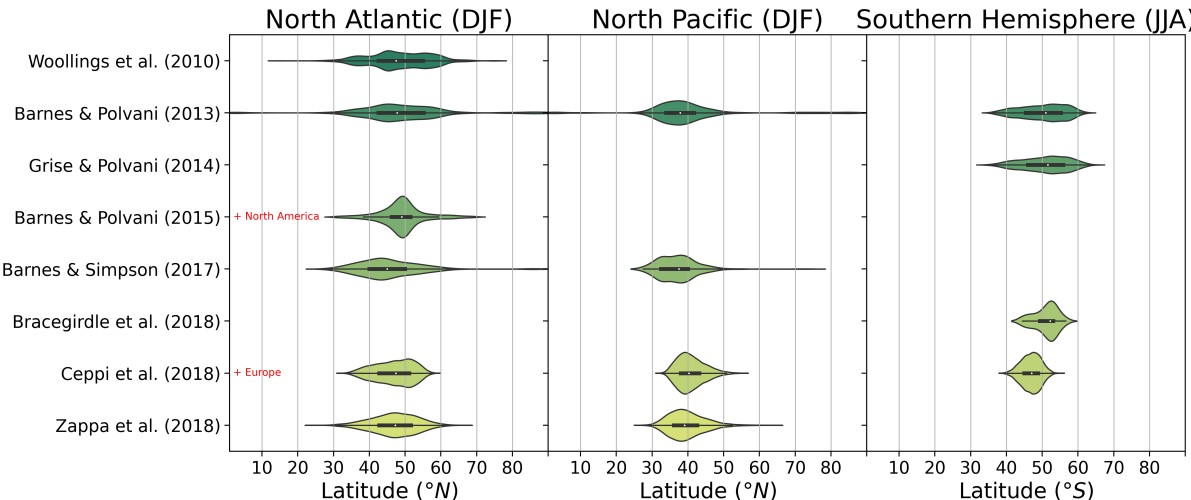

**Figure 2.** A comparison of the daily mean position of the jet stream during the winter months between 1st January 1979 and 31st January 2022 in three study regions as specified by 8 jet latitude metrics available in the *jsmetrics* package. The thicker horizontal lines inside each violin represent the interquartile range, and the thinner lines represent the 95% confidence interval. The white dot represents the median, and the shading which forms the body of each violin is a Kernel Density Estimation. The region of the North Atlantic is combined with North America in Barnes and Polvani (2015), and with Europe in Ceppi et al. (2018) (Data: ERA5 climate reanalysis product; Hersbach et al., 2020).

As shown in this figure, the distribution of daily latitude of the jet stream in the winter is shown to be relatively wider in the North Atlantic region (Interquartile Range (IQR) varying between 5.32-12.5°N across the metrics) than in the North Pacific (IQR=5.14-7.79°N) and Southern Hemisphere (IQR=3.85-9.94°S) across the metrics. For the North Atlantic, we show that the

degree of uncertainty arising from choice of jet latitude statistic (or the *metric uncertainty*) on the mean position is 3.81°N (between 45.26-49.08°N). In contrast, the uncertainty arising from internal variability, for which we use IQR as a proxy, is 7.18°N (between 5.32-12.5°N) across the metrics, implying the internal variability has a relative larger impact on uncertainty associated with the jet position. The mean position of the jet stream across 1979-2022 is shown to be between 37.24-40.81°N for the North Pacific and between 46.85-51.01°S for the Southern Hemisphere, as estimated by these metrics.

In Figure 2, some general differences between the metrics may arise due to differences in the region definition, e.g. BP15 & C18, and differences in pressure level from which the metric has been calculated, e.g. BS17 (see Table 1). Further, while W10, BP13 and BP15 adopt a similar methodology and look at data from pressure levels between 700-925 hPa. GP14, BS17, C18, Z18, and B18 use one pressure level (either 700 hPa or 850 hPa). The motivation for using relatively low-level pressure levels (between 700-925 hPa) is to remove the signal of thermally-driven parts of the jet stream and isolate the eddy-driven

parts (which act as an important control on various aspects of the mid-latitude climate; Hallam et al., 2022). Eddy-driven jet streams tend to be deeper and thus are more likely to extend down towards the surface than thermally-driven jets, which tend to

be shallower and generally higher up in the troposphere (Held, 1975; Held and Hou, 1980; Madonna et al., 2017; Spensberger et al., 2023).

The above example demonstrates that when viewing jet latitude estimations in this manner, researchers may be able to evaluate metric uncertainties arising from differences in methods used to characterise jet streams. These figures highlight some preliminary divergence in metric uncertainty across different regions of the globe arising from various existing metrics.

## 4.2 Case study 2: Identifying the jet stream across North America during the February 2021 North American Cold Wave

For the second case study, we examine the representation of the jet stream across North America during the 2021 North American Cold Wave event, which occurred between 6th and 21st February 2021. This event was associated with an anomalous cold air outbreak over North America occurring in late January 2021 (Cohen et al., 2020, 2021; Rao et al., 2021) and has been linked with a (strong) negative phase of the Pacific–North American pattern (Hsu et al., 2022). For this section, we have used 6-hourly averaged $u-$ and $v-$component wind-speed data from the ERA5 climate reanalysis (Hersbach et al., 2020) at a $1°$ by $1°$ grid for the pressure levels 100, 250, 300, 400, 500 hPa accessed via the Climate Data Store API. We isolate just one 6-hour period from the cold wave: 0000 UTC on 15th February 2021 and compare wind speed at 250 hPa to 5 jet core algorithms from the *jsmetrics* package in Figure 3. We note that the vertical resolution and grid spacing of the data used in this case study, may not be adequate for some of the methods to effectively capture jet cores (see a discussion of vertical resolution and grid spacing in Manney et al. (2017)). Instead, the data has been selected based on faster algorithm run time and reproducibility of the figures included in this section. Finally, we have selected jet cores at 250 hPa from M11 and PO13 for comparison with wind speed and K14, but we acknowledge that these two algorithms also return jet core outputs at different altitudes.

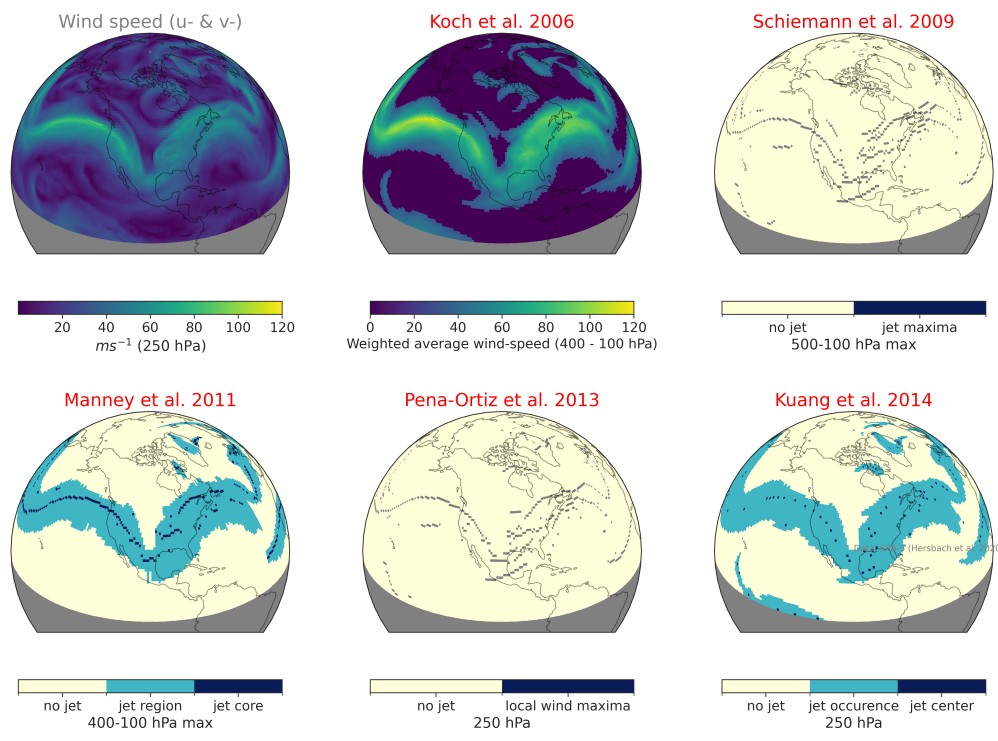

**Figure 3.** Comparison of the estimation of the jet stream position during the North American Cold Wave event at 0000 UTC on 15th February 2021 as estimated by 5 jet core algorithms available in the *jsmetrics* package. The top left panel shows the 250 hPa resultant wind speed as calculated from *u-* and *v-* component winds (Data: ERA5 climate reanalysis product; Hersbach et al., 2020).

When viewing the upper-level jet stream over North America at this given instance of the North American Cold Wave event and between 5 unique *jet core algorithm* metrics, it is clear that each metric is identifying the same broad pattern — a well-defined singular band across North America and a trough that extends down towards Texas. Notably, S09, M11, P013, and K14 all use a 30 $ms^{-1}$ threshold (but not in the same way) and both S09 and PO13 select only cells of local 'maxima'; M11 and K14 also extract regions around each core/maxima. There are only slight visual differences between the jet cores in PO13 and S09, because both algorithms make use of a wind-speed threshold of 30 $ms^{-1}$ to extract local maxima in the altitude/latitude plane, but S09 isolate *jet cores* only where the $u-$component wind is also shown to be above 0 $ms^{-1}$. M11 use an additional algorithm after initial discovery of local maxima to divide jet cores occurring within the same local maxima region based on whether: (1) the two or more cores are more than 15°of latitude apart and (2) whether the wind speed drops more than 15 $ms^{-1}$ between those cores; otherwise these jet cores in the same region will be considered part of the same core, at the location of the

largest of the local wind speed maxima. As such, the jet core output from M11 at 250 hPa may vary slightly from other similar methods (e.g. S09 & PO13), as the jet cores in each may be associated with different altitudes. K14 also relies on checking for local maxima, but within the longitude/latitude plane. The methodology checks for jet occurrence and jet centers, which are defined in grid cells, whereby wind speeds above 30 $ms^{-1}$ are local maxima (so they have a higher wind speed than all the surrounding 8 grid cells). As such, this algorithm distinguishes between two different categories of jet stream occurrences: making the assumption that the centres of jet streams are important features in their own right, as opposed to regions where a given wind threshold is exceeded (Kuang et al., 2014).

With this case study, we demonstrate the slight differences in the estimations of the jet stream from various jet core algorithms, and suggest that the difference at the 6-hourly scale will likely be amplified when aggregating into coarser time resolutions.

Next, we look at two waviness metrics during the North American Cold Wave event in Figure 4. As estimated by FV15, a large negative MCI (MCI less than 0 indicates Northerly flow) patch is shown over Western North America during the 10-day period encompassing the North American Cold Wave event. C16 provides a more limited view of the event, but shows some variation in the temporal profile of the upper air sinuosity (jet waviness) during the North American Cold Wave event, with relatively higher and more sustained sinuosity in the upper-air flow being associated between 0000 UTC on 13th February and 1800 UTC on 15th February.

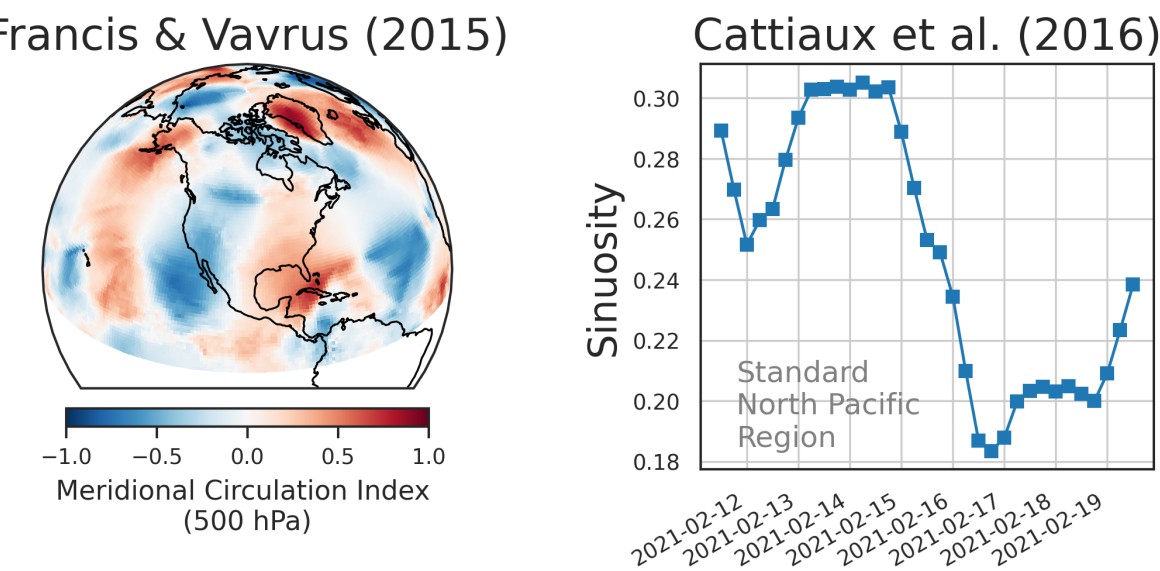

**Figure 4.** A comparison of jet waviness during the North American Cold Wave event between 1200 UTC on 11th February and 1200 UTC on 19th February 2021 as estimated by two waviness metrics available in *jsmetrics*. The metric from Francis and Vavrus (2015) is a mean of MCI calculated for each 6 hour time step between 1200 UTC on 11th February and 1200 UTC on 19th February 2021. When MCI is 0, the wind is purely zonal, and when MCI is 1 (-1), the flow is from the South (North). The sinuosity metric from Cattiaux et al. (2016) is calculated over a standard North Pacific region (0-90°N & 120°E-120°W) (Data: ERA5 climate reanalysis product; Hersbach et al., 2020).

## 4.3 Other potential uses

The *jsmetrics* package is designed to be flexible with both the inputs and the calculation of a given metric. While a user can change the exact specifications by which some metrics are calculated (e.g. changing wind-speed thresholds and filter window sizes), users can also pass different subsets/specifications of data into the metrics (e.g. different spatial-temporal regions and resolutions). As such, this opens up the possibility to do sensitivity analysis to explore or evaluate:

1. *metric uncertainty* — by comparing the estimations of the jet stream using multiple statistics or algorithms on a single dataset.

2. *parametric uncertainty* — by comparing the estimation of the jet stream from a given metric using slightly different specifications, i.e. filter window-sizes, thresholds, etc.

3. *input uncertainty* — by comparing the estimation of the jet streams in different domains (pressure levels, spatial-temporal resolution) and with different datasets.

In Figures 5 and 6, we demonstrate a simple evaluation of *metric uncertainty* using the same dataset and metrics as case study 1 (Section 4.1: winter jet latitude and speed), but with a single set of specifications: vertical levels of 700-925 hPa, for the four fixed regions of North Atlantic (15-75°N & 60-0°W), North Pacific (0-90°N & 120°E-120°W), Northern Hemisphere (0-90°N), Southern Hemisphere (0-90°S). We include the extension to C18 and Z18 proposed by Screen et al. (2022), to produce an associated jet speed for those methods. To demonstrate the sensitivity of the jet statistics to parameters in their definition, we also show in Figure 6 the distribution of jet speed over the various regions used in the respective studies (see Table 1), rather than a single common region.

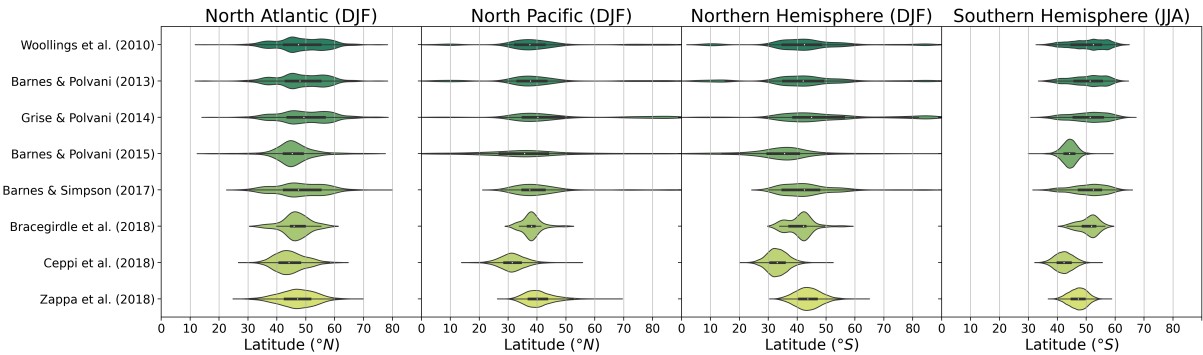

**Figure 5.** A comparison of the 700-925 hPa winter daily mean position of the jet stream between 1st January 1979 and 31st January 2022 in four standard regions as estimated by 8 jet latitude metrics available in the *jsmetrics* package. The thicker horizontal lines inside each violin represent the interquartile range, and the thinner lines represent the 95% confidence interval. The white dot represents the median, and the shading which forms the body of each violin is a Kernel Density Estimation (Data: ERA5 climate reanalysis product; Hersbach et al., 2020).

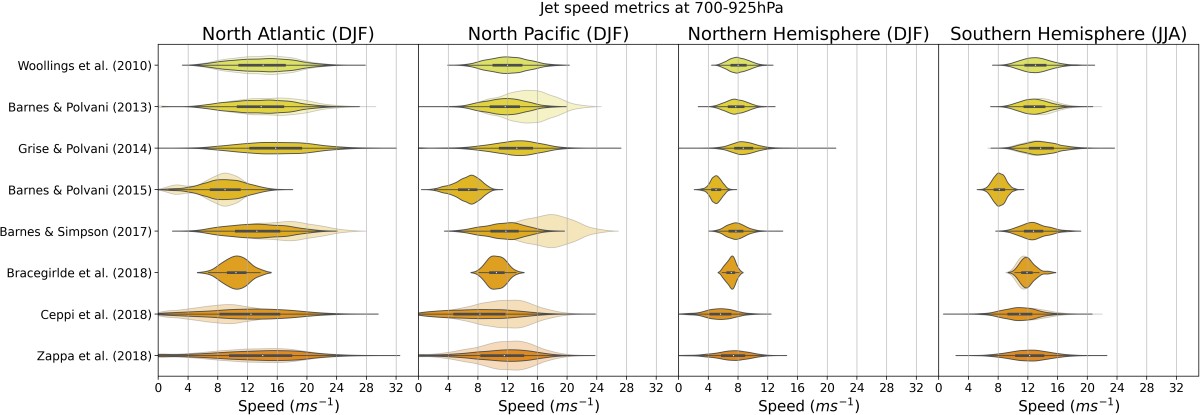

**Figure 6.** A comparison of the 700-925 hPa winter daily mean jet speed between 1st January 1979 and 31st January 2022 in four standard regions as estimated by 8 jet speed metrics available in the *jsmetrics* package (Data: ERA5 climate reanalysis product; Hersbach et al., 2020). Where the original studies used a different regional definition (see Table 1), the distribution of jet speed using that altered region is shown underneath the distribution determined from the common standard region.

As shown in these figures, there are clear divergences in the distribution of daily jet latitude position and jet speed estimated by the various metrics using the same dataset. In Figure 5, the mean jet position varies more in the Northern Hemisphere (33.22-49.75°N) than in the North Atlantic (44.62-49.55°N), North Pacific (31.57-46.81°N), or Southern Hemisphere (42.37-50.59°S). The estimation of jet mean position from C18 over this period is found to perform less similarly to the other jet latitude statistics, showing the most equatorward estimation in each region.

In Figure 6, the strongest and most variable jet speed estimations are shown in the North Atlantic (8.92-15.31 $ms^{-1}$), versus the North Pacific (6.56-11.51 $ms^{-1}$), Northern Hemisphere (5.04-7.77 $ms^{-1}$) or Southern Hemisphere (8.15-13.07 $ms^{-1}$). We also find that the BP13, BS17, and C18 methods are sensitive to the original definitions of the North Pacific region (Figure 6).

In viewing the jet statistics in this manner, we hope to have demonstrated that using any one metric in isolation is associated
with a significant level of metric uncertainty — so estimates of how much a jet has shifted will strongly depend on the metric. In particular, Figure 5 shows that some metrics show more variation in their estimates across multiple regions than others. As such, the *jsmetrics* package could be used to evaluate the sensitivity of each metric to varying definitions of regions.

Jet streams are chaotic actors in the atmosphere, and as such, there is no universal strategy to capture their features at any timescale in data (e.g. Maher et al., 2020; Bösiger et al., 2022, and references therein). Therefore, in the next example, we
explore the effect of *input uncertainty* by using a jet core algorithm on data with different time averaging periods. We use the K14 metric, which classifies jet occurrence centres in the upper-air wind (200-250 hPa). These centres are defined as grid cells where wind speed is above 30 $ms^{-1}$ and a local maxima compared with the surrounding eight grid cells. We examine the effect of six different time averaging periods (all centred on 1200 UTC on 15th February 2021) on characterisation of the North American Cold Wave event in February 2021, using the same data detailed in Section 4.2.

Figure 7 shows a clear trough in upper-level jet occurrence and jet occurrence centres extending south towards Texas in mid-February, but the extent to which this feature is visible depends on the timescale used. This feature is robust up to about 4 days, but a trough structure becomes less clear in the jet occurrences and jet centers beyond that. We expect large-scale and persistent features of the jet stream (in this case a stationary/standing wave over North America) to be more defined/stable at broader time scales if the weather system remains and the features of the jet stay in place over a region. Note that this metric

finds jet features over Greenland at the finer time scales, but these features are lost with temporal averaging.

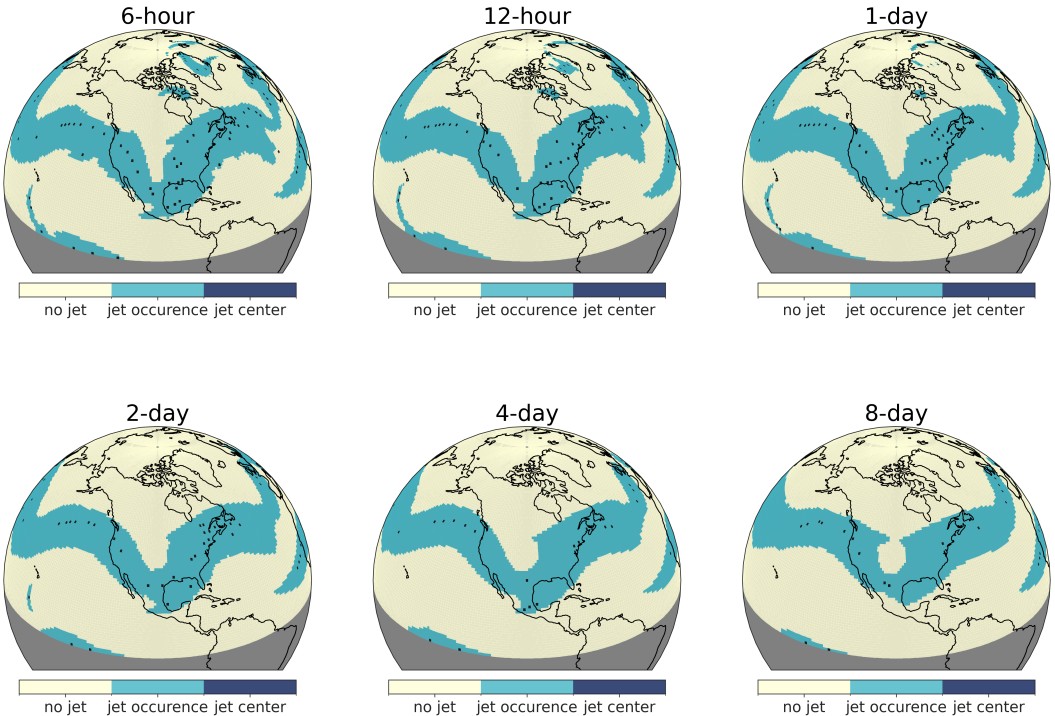

**Figure 7.** Jet occurrence and jet occurrence centre points as determined by the algorithm from Kuang et al. (2014) at 250 hPa at 6-hour, 12-hour, 1-day, 2-day, 4-day and 8-day averaging periods during the North American Cold Wave event centring on 1200 UTC on 15th February 2021 (Data: ERA5 climate reanalysis product; Hersbach et al., 2020).

Next, we compare the 8-day mean with the count of 6-hourly means of the jet occurrence centres from K14 around the North American Cold Wave event between 1200 UTC on 11th of February 2021 and 1200 UTC on 19th of February 2021. We use a 2-sigma Gaussian filter around the 32 6-hourly jet centres to smooth the counts in each 1° by 1° grid cell. The comparison (Figure 8) demonstrates the losses and gains of time averaging: some features are diluted using the *mean*, while *counts* show

more detail but can also include more noise.

These examples highlight the care needed in study design. Using only one temporal scale, without considering the effect of temporal averaging on jet features (given the current lack of knowledge about which scales are appropriate for a given purpose), is likely to underestimate uncertainty in estimation of the jet streams.

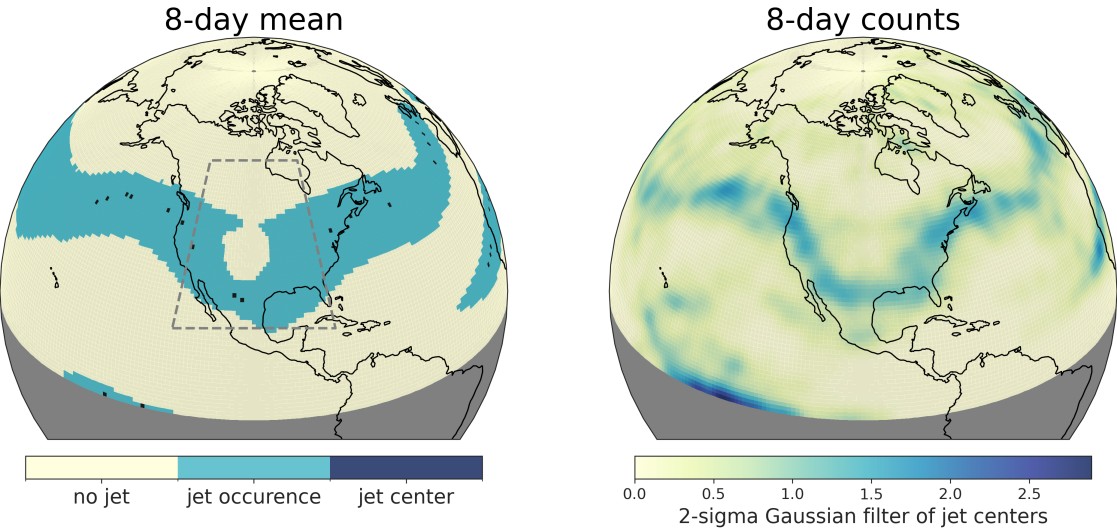

**Figure 8.** A comparison of 8-day daily counts versus mean of jet occurrence centres as determined by the algorithm from Kuang et al. (2014) during the North American Cold Wave event centring on 1200 UTC on 15th February 2021 (Data: ERA5 climate reanalysis product; Hersbach et al., 2020).

In our last example, we extract a single value – the latitude of the jet stream over a study area – to compare the estimations of
395 6 jet core algorithms to the estimation of the latitude of the jet stream to 7 metrics available in *jsmetrics* that are purpose-built for extracting a jet latitude. We use 8 days of the 2021 North American Cold Wave and the region outlined in Figure 8 (120-80°W, 20-60°N) to do this (Figure 9). To create an estimate for jet latitude from the jet core algorithms, we first compute the estimation of jet cores using a given algorithm and use these locations as a mask to extract wind speed values for each day. Using these values, we then extract the zonally-averaged maximum wind speed and define the associated latitude as the jet latitude value
at the native resolution. For consistency's sake, we use a single method to extract the latitude from the multidimensional field returned by the algorithms in this case study. This is the latitude of the maximum wind in the region (despite other options being available to do this for the multidimensional fields, e.g. Manney et al. (2011) would select all the indexes of returned jet cores). Future versions of jsmetrics could contain a variety of procedures that process the outputs of jet core algorithms into jet statistics. AC08, BS17 & B18 all produce a single value of jet latitude due to the temporal resolution used for this example.
These estimates are 37.42°N, 32°N, & 31.78°N, respectively. This figure shows the jet latitude to be generally more polewards as determined by the jet core algorithms compared with the jet latitude metrics. This is most likely due to the altitude of the methodology, as the jet core algorithms are looking at the upper troposphere and the jet statistics algorithms are looking at the lower troposphere (Tables 1 & 3). Notably, only a few of the metrics produce a bimodal distribution of the jet latitude, which is observed in the maximum zonal wind speed profile during this period, but this includes none of the jet core algorithms, which
use a wind speed threshold.

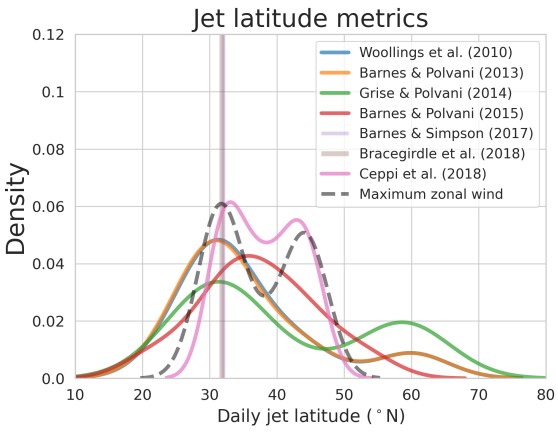 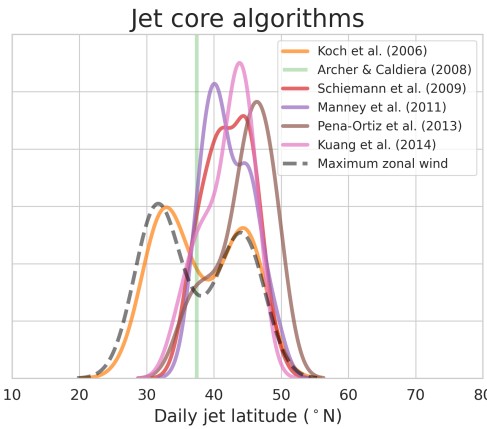

**Figure 9.** A comparison of 6-hourly latitude of maximum wind speed estimations from jet latitude metrics and jet core algorithms available in *jmsetrics* during the North American Cold Wave event between 80–120°W and 20–60°N between 1200 UTC on 11th February and 1200 UTC on 19th February 2021. *Maximum zonal wind* is the zonally-averaged maximum wind speed, calculated using *u*- and *v*-component wind (Data: ERA5 climate reanalysis product; Hersbach et al., 2020).

## 5 Future Work

The *jsmetrics* package is a work in progress, but aims to be a flexible and useful research tool for comparing and refining existing jet metrics, as well as a platform for developing new metrics in the future. Apart from adding new metrics to the module, detailed in section 5.1, there are a few directions for the current use of the *jsmetrics* package. As a package, *jsmetrics* provides
no scripts for running analysis of various jet stream metrics in combination, as we have demonstrated in section 4. Therefore, one direction for the use of *jsmetrics* is scripts or a module built on top of *jsmetrics* that is made to run a comparison of multiple metrics. For the analysis in section 4, we used scripts that make use of specification files (like *details_for_all_metrics.py*) that detail the data sub-setting, expected input variables and the function to run. We then wrote a script containing an 'AnalysisRunner' class to actually handle the experiment and loop over and calculate the metrics in a manner specified by the specification
files on a given dataset. As outlined in section 3, this is made possible as the package does not attempt to subset the input data: instead, it is expected that the user handles the quality and specification of data passed into *jsmetrics*. Running metrics in combination opens up the possibility of evaluating the input, metric, and parametric uncertainty associated with the estimations of the jet stream latitude, speed, waviness, or location (depending on the experiment, and which metrics are currently in the package).
Another direction is to write a script to run the analysis on multiple datasets, built on top of modules using specification files. This could be used not only to evaluate jet stream estimates in different input reanalysis datasets (as in Pena-Ortiz et al., 2013; Manney et al., 2017, 2021), but also in multiple climate model projections (e.g. the CMIP6 multi-model ensemble; Eyring et al., 2016), to search for coherent patterns and emergent observational constraints of future jet-stream behaviour.

Other metrics libraries and packages are written in Python and developed for use with NetCDF4 and xarray datasets. There is the potential to include various metric implementations within Python's xclim — a Python library of derived climate variables and climate indicators, based on xarray (Logan et al., 2022). Further, a comparison of various jet stream metrics as calculated with the *jsmetrics* package has the potential to be integrated as a recipe for the ESMVal Tool for evaluating CMIP6 model data (Andela et al., 2022).

## 5.1 Adding metrics

The *jsmetrics* package has a guide to contributing available on ReadTheDocs (https://jsmetrics.readthedocs.io/en/latest/contributing. html, last accessed: $16^{th}$ October 2023). This project is a strictly open-source project and has a strong copy-left licence (GNU General Public Licence v3.0). The *jsmetrics* package is designed to be easy to contribute to and there is an emphasis on future metrics being built upon a collection of generalised sub-functions that can be shared with similar metrics e.g., for calculating zonal mean wind speed, or applying a low-pass filter. Because of the inherent similarity of some existing metrics currently implemented in *jsmetrics*, we recommend first looking for similar metrics that have been implemented and viewing how they are defined within this package. The aim of adding any new metric should be to try to minimise the amount of repeating code and to standardise the components of the metrics as much as possible so that they can run with slightly altered inputs, i.e., with different wind speed thresholds, different filter window sizes etc. We recommend experimenting with various designs of any prospective addition to *jsmetrics* in a Jupyter-notebook and to prioritise fast and simple implementations of that given metric.

We have leant into the capabilities of GitHub to log the progress of any given metrics. We open a new GitHub Issue to log and describe a new potential metric and GitHub Projects to track the progress of a given metric in a manner explained in Section 3.2 and in Figure 1. In Table 5 we outline some further metrics that are in the process of being implemented or could be implemented in the future. It is possible that as the package expands, there is an opportunity to refine the categories developed to contain and define different types of metrics and also those that look at different types of jet streams i.e. low-level, eddy-driven, thermally-driven jets etc. Finally, we note that some metrics may be too complex for the remit of this package (e.g. Kern et al., 2018; Kern and Westermann, 2019; Bösiger et al., 2022). When developing the package, we avoided metrics that use variables describing different aspects of the upper-level flow synonymous with (characteristics of) jet streams, such as wind shear (e.g. Lee et al., 2019) and magnitude of atmospheric waves (e.g. Chemke and Ming, 2020). Similarly, we did not include any potential metrics that require a training element to run and those that are currently very computationally expensive (e.g. Limbach et al., 2012; Molnos et al., 2017).

## 6 Conclusions

In this work, we have introduced the features of *jsmetrics* — a Python package containing an implementation of 17 metrics or algorithms used to identify atmospheric jet streams, and we have demonstrated its use on climate reanalysis data. The motivation for developing this software comes from a desire to standardise, and make openly available, various methods used to identify and characterise jet streams such that they can be used in combination, compared and contrasted. It is hoped that this

**Table 5.** Techniques for identification or characterisation of jet streams in the literature not yet implemented in the *jsmetrics* package ($u$-, $v$- and $w$- refer to the zonal, meridional and vertical wind components; $zg$ refers to the gravity-adjusted geopotential height)

| Study | Variable(s) | Pressure-level (hPa) | Temporal | Method |
|---|---|---|---|---|
| | | Jet statistics | | |
| Strong and Davis (2007) [1] | $u$-, $v$- | 100-500 | 6-hourly | Surface of max wind speed |
| Barton and Ellis (2009) | $u$- | 300 | Daily | Latitude of maximum wind speed |
| Harnik et al. (2014) | $u$- | 300 hPa | Daily | Jet Latitude Index |
| Messori and Caballero (2015) | $u$-, $v$- | 200–400 & 700–925 | any | Jet Angle Index & Jet Latitude Index |
| Simpson et al. (2018) | $u$- | 700 | Monthly | 20-year running mean zonal wind |
| Liu et al. (2021) | $u$- | 250 & 850 | Daily | Extends Barnes and Polvani (2013) |
| Mangini et al. (2021) [2] | $u$- | 700-900 | Daily | Jet clusters using K-means |
| Blackport and Fyfe (2022) | $u$- | 700 | Daily | Extends Barnes and Polvani (2015) |
| Hallam et al. (2022) | $u$- | 250 | Daily | Maximum wind speed by longitude |
| | | Jet waviness metrics | | |
| Screen and Simmonds (2014) [3] | $zg$ | 500 | Daily | Wave amplitude metrics |
| Martineau et al. (2017) [4] | $zg$ | 500 | Daily | Local finite wave activity |
| | | Jet core algorithms | | |
| Gallego et al. (2005) | $zg$ | 200 | Daily | Geo-strophic stream-line algorithm |
| Chenoli et al. (2017) | $u$-, $v$- | 100-300 | Daily | Extends Pena-Ortiz et al. (2013) |
| Spensberger et al. (2017) [5] | $PV$ | 280-380 K PVU | 6-hourly | Jet axis algorithm |
| Rikus (2018) | $u$- | 0-1000 | any | Discrete object algorithm |

[1] adapted from Strong and Davis (2005, 2006); [2] adapted from Madonna et al. (2017); [3] adapted from Screen and Simmonds (2013); [4] adapted from Chen et al. (2015); Huang and Nakamura (2016); [5] adapted from Berry et al. (2007);

software can open up new avenues for researchers for evaluating both the location and characterisation of the jet streams and also open up a more comprehensive quantification of various uncertainties associated with using different methods, datasets, and specifications (metric, parametric, input uncertainty, respectively)

We have tried to keep the package as simple to use and install as possible for those who wish to use the package as a research tool, but there is also a lot of scope for the package to be built upon and extended. As we outline in Section 5.1, the process of adding new metrics to the package is relatively formulaic and extensively logged on GitHub. The package provides a collection of generalised functions that form components of the metrics, so it is easy enough to edit aspects of existing metrics included in the module and also to develop new metrics from these generalised functions. Furthermore, the metrics included in the package make no explicit attempt to change or subset the input data to the original specifications of the paper they stem from, so they are adaptable to different regions, times, scales, and to future data products.

*Code and data availability.* The up-to-date version of *jsmetrics* is available at: https://github.com/Thomasjkeel/jsmetrics. *jsmetrics* is also accessible on PyPi via the Python **pip** package manager. It is archived at: https://zenodo.org/record/7377570. All data used are available from ERA-5 climate re-analysis available from the Climate Data Store.

*Author contributions.* TK undertook this research under the supervision of CB & TE. All author contributed to writing the manuscript.

*Competing interests.* The authors declare that they have no conflict of interest.

*Acknowledgements.* We would firstly like to thank all of those who were involved in curating the jet stream metrics and algorithms that form a part of the jsmetrics software, including guidance from Paulo Ceppi, Gloria Manney and James Screen, among others. We are also very grateful and would like to extend a special thanks to Gloria Manney and one anonymous reviewer for their thorough review of the preliminary manuscripts and software. Finally, we would like to acknowledge and thank Raquel Alegre and others at UCL ARC, Clair
Barnes at the Grantham Institute, and all the co-developers of the xclim software, for the inspiration and assistance they provided throughout the development of jsmetrics.

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
