# Peer review of "jsmetrics v0.2.0: a Python package for metrics and algorithms used to identify or characterise atmospheric jet-streams."

_EGUsphere, 2023_

## Referee Comment (RC2)

Review of "jsmetrics v0.1.1: a Python package for metrics and algorithms used to identify or characterise atmospheric jet-streams", by Keel, et al.
Reviewed by Gloria Manney

**Recommendation:** I raise some questions about the maturity of the jsmetrics package/documentation and thus the current suitability for publication in GMD. Substantial revisions and clarifications in the manuscript, and improvements in the code documentation, would be needed before publication.

**General Comments:** I was initially very excited to read this manuscript and try the jsmetrics package. I think the idea of putting together this sort of package is a great one. After reading the paper, installing (via pip, so presumably latest stable version as of late July 2023) and trying out the package, and looking at the documentation both on GitHub and the site given in Keel (2023), I have several reservations about this package as it now stands regarding its maturity, its interpretation and implementation of some of the methods, the frugality of the information returned by the package, and the opaqueness of the documentation and usage guidance. In particular,

(1) Of the paper types listed as suitable for GMD, the category this would fit into is "methods for assessment of models". The description of this paper type includes "..description of a fully fledged software tool…". I question whether this package can at present be considered "fully fledged". For example, of the 17 metrics (summing the three types) described and exemplified in the paper, on the website, four of them are listed as complete, one as "in progress" with a note indicating "need help", and the rest as "to verify". As far as I can tell (e.g., extrapolating from Fig. 1 in the paper), "to verify" means that the implementation has not yet been verified against results from the original paper. Even if thus limited only to the approaches discussed in the paper (where the reader is likely to assume that they have all been fully tested), I question whether this can be considered a mature software package. I recognize that the authors are hoping to make this a community project and entrain outside contributors, but if this manuscript is a call for participants – without a basic foundation of mature software that is to be built upon – then I am not sure it is appropriate for publication in GMD at the present state of maturity of the software package.

(2) I question the distinction (and the need for one) between "jet statistics" and "jet core algorithms". All of these methods aim to extract basic information such as the jet latitude (and altitude for several of them) and windspeed. For example, the method of Manney et al. (2011, hereinafter M11; though the package is formally named JETPAC, that was not mentioned in the first couple of papers) was based on extending that of Koch et al. to characterise the jets in the latitude/altitude plane rather than only getting horizontal locations and vertically averaged dynamical characteristics. But the two papers (as well as Manney et al., 2014) have very similar aims and show very similar diagnostics (eg., frequency distributions of jet cores and associated winds) – so why is one "jet statistics" algorithm and the other a "jet core algorithm"? I see no need for a distinction here.

(3) Obviously, there is one method herein with which I'm very familiar (and to a lesser degree several of the other methods, either because mine built on them, they (apparently) built on mine, or I reviewed the original papers. I looked closely at their implementation of the M11 method, and I cannot from the information provided (nor from running an example of their implementation) convince myself whether or not what they are doing closely resembles the M11 method: In the paper (lines 137–140)l they do not list M11 in the category "(ii) in relation to wind-speeds of neighbouring data points (local wind-speed maxima…", but this method is fundamentally based on first finding all local wind-speed maxima (in a latitude / altitude plane) and then applying thresholding and other criteria. In fact, the M11 method is fundamentally nearly identical to the Peña-Ortiz et al (2013; hereinafter PO2013) method (the latter paper showed internal evidence that they must have known of M11, but did not cite it), which is listed in both categories that would be appropriate for M11.

In what appears to be the primary function to identify the cores, there is a comment saying "Core are discovered where 8-cells are above boundary threshold… Paper uses 100-400 hPa." The M11 method inherently operates on a slice of the windspeed (and other fields) with a latitude-like and an altitude-like coordinate (and then is generally run at each longitude gridpoint for each day); jet cores are identified in that latitude/altitude plane by finding the local maxima (as points with higher windspeed than the nearest neighbors in altitude and latitude) and testing for exceedance of the core threshold. Finding the "edges" of the jet regions is no more complicated (or informative!) than finding the latitude gridpoint on each side and vertical coordinate (eg, altitude) gridpoint above and below where the windspeed drops below the boundary threshold value. Other criteria that determine whether to consider two cores in the same "region" (that is, where the windspeed does not drop below the boundary threshold between them) separate are latitude distance between the cores (which this package appears to implement) and the amount by which the windspeed drops along a line connecting the cores in altitude / latitude space (which I didn't find mentioned herein or in the documentation). 100-400 hPa is the region to which the jet **core** altitude is limited, but the jet regions surrounding any core are identified at altitudes that may be above and/or below that – wherever they occur. This procedure is described in detail in M11 and Manney et al (2014), and completely but in less detail in Manney et al (2017, 2021; latter J Clim, DOI: 10.1175/JCLI-D-20-0947.1) and Manney & Hegglin (2018). It is not clear to me that either the description in this manuscript and the jsmetrics code, or the implementation in the code are doing something that is fundamentally similar to this method.

Further, in lines 155–157, they ignore the update to the M11 method provided by Manney & Hegglin (2018) (and used in all succeeding papers using the method, including Manney et al, 2021), whereby the subtropical jet is identified (and thus distinguished from polar jets) using a physically-based method. In the preceding lines, they also fail to note that PO13 themselves concluded that their method for separating subtropical and polar jets did not work well; which is consistent with the statement in Manney et al, 2014 regarding the division of subtropical and polar jets by latitude (which they used in the one case where they did not simply analyze all the jet cores at each longitude) being useful only for very broad climatological studies.

I've gone into (probably tiresomely) lengthy detail here because this case (and the difference in the descriptions of M11 and PO13, which are very similar methods) makes me question whether there may be other metrics for which the implementation may not accurately reflect the original method. If I had been putting together a package of this sort, I would have contacted the authors of the original papers and tried to verify my understanding of the metrics and what was fundamental to them – has this been done for any of the methods? You probably would not have gotten a 100% response rate, but many authors (like me) would have been happy to confirm the algorithm and to support your efforts to the extent we could find the time.

(4) The information that is provided as the output from running these algorithms does not include some of the most fundamental outputs. Continuing to use M11 as an example, the fundamental output provided by M11 (ignoring their very complete characterization of numerous other dynamical fields at the jet core locations) is not just jet core latitude, but also jet core winds (windspeed, zonal and meridional winds), and jet core altitude (PO13, at least, also provide these characteristics, and I presume others do as well); in M11 the characterization of the boundaries of the jet regions is secondary information, they focus on full characterization of the jet cores. These are all very fundamental quantities that are used in virtually all studies of jets and their trends (omitting altitude for methods that don't provide that – but for methods that do it is one of a several fundamental characteristics that are expected to change with climate change, see, e.g., discussion / references in Manney & Hegglin 2018) and in most cases (certainly for M11) these quantities "come along" with (because they are a fundamental part of) the jet core identification / characterisation. Thus it seems to me an important (and easily remedied) omission to not provide these quantities in the output.

(5) I find the documentation – in the GitHub project, in https://jsmetrics.readthedocs.io/en/latest/, and in the code itself – opaque, hard to navigate, and incomplete (though the last may be an artifact of the previous issues). Indeed, part of point (3) above could be because I could not find complete descriptions of the algorithm as implemented for each of the methods. Another example is the "Usage" information, which says that to run the Kuang et al (2014) algorithm, after reading/formatting a u & v wind dataset in uv_data, you would type:
k14 = jsmetrics.jetstream_algorithms.kuang_et_al_2014(uv_data)
Whereas I had to type (syntax obtained from looking through the code):
k14 = jsmetrics.metrics.jet_core_algorithms.kuang_et_al_2014(uv_data)
(I cannot find any objects with names containing "jetstream_"). I did, finally, find some end-to-end usage examples (that have correct syntax) by following several layers of links from the main README.rst file in the GitHub project – this information should be in https://jsmetrics.readthedocs.io/en/latest/ (in a corrected / expanded usage section, and either in an examples section like I've seen in other packages I've installed with this format of documentation, or with the API reference information as examples for particular metrics / algorithms).
In "Metrics & Algorithms" in the documentation (link above):
- The "all metrics" link gives a 404 error.

- There do not appear to be any API-type descriptions of the "Metrics", thus zero guidance on how to run them; the API-type descriptions of the "Algorithms" are not complete enough to be useful. For instance, basic information for formatting the input fields (whether the algorithm needs one level or multiple levels; if it works on one level at a time, does it handle, and how does it handle, multiple levels) should be given. There is also little or no information about what the output Dataset contains in each case. There should be a full API reference that describes each routine in enough detail that a user with basic familiarity with python can run them. (I ran several of them, but generally had to go to source code and even then guess at some things to see how to do that.)
- In the API-type information given for the "Algorithms", the descriptions of the methods are not clear (see discussion re M11 above), nor sufficiently detailed to know what the implementation actually does. The code itself, for the vast majority of routines that actually do things (as opposed to just set up inputs, etc) are going to be very opaque to anyone who isn't fluent in the full object-oriented syntax / usage – that class includes many, many python users in the atmospheric science / climate community (partly because python can be used so effectively without even getting to using it in true object-oriented implementations). I'm not criticizing how the code is written, but doing it this way makes it critical to document what it actually does so the user doesn't have to be able to read the code at that level. (The page at the "issues" link gives descriptions of the algorithms directly from the original papers, but nothing that shows how that is "translated" into the implementation here. Being able to see the pseudo-code that is mentioned in the paper would be extremely helpful as one way to do this.)

Further details are given in the specific comments. Altogether, the package and its documentation appear sufficiently immature to question "announcing" it in a paper.
* * *
**Specific Comments** (in order of appearance in the manuscript, not importance):

Lines 1–17, The abstract would benefit from spending less time on motivation and more on describing the actual results of the paper (just 3 / 17 lines are devoted to what the paper actually does).

Lines 26–33, See general point (2).

LInes 53–55, Should cite Lee & Kim (2003) and Manney et al (2014) in addition to Spensberger and Spengler (2020) (indeed, much of the latter's discussion was based on the discussion in those earlier papers).

Line 81, not all of these (eg, Barton & Ellis use one specific pressure level) describe the "atmospheric column".

Lines 91–92, these also do not capture the behavior near the level of maximum wind speed, see, e.g., discussion in Melamed-Turkish et al. (2018) and Manney et al (2021).

Lines 93–95, how is "extracting the latitude of the jet stream as the point of fastest zonal wind" different from setting "a value of jet latitude and jet speed for each longitude"? Are you saying in the first statement that the maximum is found as a function of latitude and longitude? (Note also, per comment on metrics versus algorithms, if you add "and jet altitude" the latter accurately describes the M11 and PO13 methods, and probably others, that are categorized as "jet core algorithms".)

Line 125, since this is indeed a "highly contested" topic, it might be good to cite some papers led by other authors, e.g., Manney & Hegglin 2018 have a fairly detailed paragraph in their introduction, citing numerous still-relevant references; Francis (2017, BAMS, DOI:10.1175/BAMS-D-17-0006.1) would also be appropriate.

Table 3 and discussion thereof, it would be helpful to distinguish between methods that provide jet altitude as well as jet latitude information (eg, M11, PO13) and those that provide only jet latitude information and that often over a vertically-averaged range (eg, Koch et al, Archer & Caldeira, Schiemann et al, Kuang et al)

Lines 139–140, per general comment (3), M11 also fall into category (ii)

LIne 142–146, It could be just the wording, but this reads as if you are saying the methods of Manney et al (including the important refinement in Manney & Hegglin, 2018 of a physically-based method to distinguish subtropical and polar jets) does not "resolve" different reasons and seasons, when in fact they specifically focused (particularly Manney et al, 2014, 2017, 2021 and Manney & Hegglin, 2018) on analysing regional and seasonal variations. Manney & Hegglin (2018; which shows polar / subtropical jet separation in different seasons) and Manney et al (2021; which, while focusing on ENSO relationships, has several figures looking at subtropical / polar jet relationships) would be good references to cite along with Maher et al (2020) for showing seasonal variations in polar and subtropical jet stream locations and separation.

Lines 154–157, PO13 used a simple latitude criterion and demonstrated/stated that it didn't work well. Manney et al (2014) did not distinguish but simply showed and discussed the spectrum of jets (I have no idea what you mean by that being "a more emergent form" and it doesn't separate the jets into "groups") – except for one climatological figure where they used a latitude criterion and discussed the fact that that was only useful for very broad climatological studies (consistent with PO13). If you are going to discuss this topic, you should note that Manney & Hegglin (2018) developed a physically-based identification of the subtropical jet and then defined the polar jets with respect to that – this is what the package originally developed by M11 has been using since then (eg, Manney et al, 2021, who discuss subtropical / polar jet relationships in the context of ENSO variations) (it would also be useful to have this method implemented in jsmetrics, though it does require tropopause altitude information, so would

require reading additional fields, which at its current state, jsmetrics seems reluctant to do.) It might also be worth noting that some methods by their nature selectively identify the subtropical jet (e.g., though it isn't one of the ones currently implemented in jsmetrics, Maher et al., 2020), and that some (e.g., several papers by Winters and others (e.g., Winters et al, 2020, DOI: 10.1175/MWR-D-19-0353.1, and references therein) based on the method described by Christenson et al (2017, https://doi.org/10.1175/JCLI-D-16-0565.1) distinguish them by postulating that they occupy different altitude ranges (in a sense, a refinement of the common procedure of using low level wind maxima to identify the polar jet, but one that can be used while still capturing the level of the core of that jet, and in a similar spirit to Koch et al.).

Lines 190–191, I was unable to find this, though I believe I looked through the website of Keel (2023) and the associated GitHub project (except for the broken link mentioned in the general comments) thoroughly. This might have been helpful in assessing more definitively whether the implementation contained the fundamental aspects of the method.

Lines 204–206, the "details_for_all_metrics.py" file contains nothing in the "description" field for the vast majority of the methods (and the one I saw that was not empty did not give a description of the method). In the entry for M11, it makes it sound like only levels between 400 and 100hPa are needed to implement it, whereas (see general comment (3)) a deeper vertical range is required.

Lines 212–213, being able to see the pseudocode would have been very helpful in not only assessing to what degree the implementation matched the original (as mentioned above), but also in understanding what the implementation actually did in general.

Lines 225–227, as already noted, I found the manner in which the "implementation-level detail" was hidden made the package quite the opposite of user-friendly. Also, the implementation is critical for the user to understand where it means that the method may differ conceptually from that in the original paper. The user needs to know what each method, as implemented in jsmetrics, actually does, so if the implementation is hidden there has to be very thorough and complete documentation of that.

Line 237–239, by "7 metrics available in jsmetrics" it appears you mean the ones you identify as "statistics". Since most (if not all) of the "jet core algorithms" provide (at least in their original implementations) wind speed (a more accurate measure of "...speed of the jet stream…" than maxima in the u-component) it would help explain why you group these together if you note that these are all applied to the lower troposphere.

Line 241, you need to explain what the "violin plots" represent, either here or in the caption for the figure where they are first shown, in detail – that is, what the width and width variations mean, what the thin and thick lines along the centre mean, and what the (barely visible) white dot in the centre of the thick line means. Not everyone is familiar with violin plots and they are not obvious – you have to orient the reader so they can make sense of what you subsequently say based on interpretation of those plots.

Lines 242–243 and Figs. 2 & 3, I see no reason to show these defined inconsistently since you are not comparing them with the original papers here (which would be the only reason not to use consistent definitions). It would make much more sense to show these only once, as they are shown in Figs. 5 & 6, so that they can actually be compared with each other. This would also eliminate the problem of not showing some of them in some regions.

Lines 265–266, also, these lower tropospheric altitudes are far below the altitude of the core (maximum winds) of these jet streams, and regional averages would be lower than at an individual longitude grid point.

Lines 268–269, you should define what you mean by "structural uncertainties".

Line 278, these five pressure levels represent vertical spacing / resolution that is inadequate to resolve upper tropospheric jet cores – see, e.g., comparison in Manney et al, 2017, of calculations using the M11 method on reanalysis model levels with the same reanalysis on standard pressure levels (which are poorer resolution than those in the models, but better than those used here).

Figure 4, why don't you show the jet core wind speeds for all those methods that calculate it (both M11 and PO13 methods do)?  Also it would be helpful to look at the jet core altitude information from those methods that calculate it (at least the two just mentioned) and see how close that actually is to 250 hPa – this would tell you how different you'd expect a jet core identified by those methods at whatever the actual altitude of the jet core is to agree with the winds at 250hPa.

Line 282, I'm not sure how really unique they are given that M11 is an extension of Koch et al and PO13 is nearly identical to M11.

Lines 287– 292, this does not seem like a particularly clear or even accurate description.  M11 simply identify all (well, up to five in each hemisphere) jet cores at each longitude. They do not assess splitting (or merging), which I think(?) is what you are trying to get at in line 290 – note, however, that a faithful (but independent) implementation of the method by Homeyer & Bowman, 2013 does add a post-hoc algorithm such as you mention to "string the jets together" in longitude and identify splitting and merging. (A complete implementation of the current M11-based method would also provide subtropical / polar jet information.)

Lines 293–296. "Surrounding 8 grid cells" in what space (latitude / longitude / altitude?).  If restricted to latitude / altitude, M11 also do this, and most certainly "make the assumption that the centres of the jet streams are important features in their own right" – that is, in fact, the central motivation for their method.  (This would also hold for PO13.)

Line 298, I don't think it is intuitive that the differences would be amplified when aggregating into coarser time resolutions. In fact my first thought was that differences might be in some sense averaged or filtered out. Why do you expect this?

Lines 306–307, parametric uncertainty testing is a fundamental part of development of any algorithm, so some of this is typically mentioned in the original papers.

Lines 308–309, Manney et al (2017) focused on this; in addition, Manney & Hegglin (2018), and Manney et al (2021), and PO13 all used multiple input datasets (mostly reanalyses in these cases).

Lines 310–314, See comment on Figs. 2 & 3 – having both is redundant, and Figs. 5 & 6 provide far more useful information.

Lines 323, 326, and 340, by "temporal aggregations", do you mean you are averaging over that number of days before doing the jet identification / characterisation? (It looks like that to me since if you plot the points for all of those days / times of day I'd expect to see more points on the plot for each successively larger time interval.) If by "temporal aggregations" you mean "time mean", then just say "time mean". (Also, assuming that's the case, in the Fig. 7 caption replace "time scales" with something like "averaging periods".)

Lines 330–332, I'd expect more *transient* features to be "washed out" for longer averaging periods and more *persistent* features to remain better defined – what you are saying sounds contrary to this.

Lines 343–344 and 346–348, most (all?) of the "jet core algorithms" are also "purpose built" for extracting jet latitudes as one of their primary products (and jet wind speed as another) – e.g., these were a focus of results in PO13 and in the Manney-led papers from 2014 on. Why do you not just use the jet latitude and wind speed that those algorithms provide? It doesn't make sense to me to compare algorithms by comparing something obtained via a method they didn't use.

I have little to say in terms of specifics on Sections 5 and 6 – since they are very general discussion I think the related issues (such as the lack of success of the package as it currently stands, particularly with the current state of the documentation, in being "simple to use").
* * *
**Minor / Technical points** (typos, grammar, wording, etc; wasn't particularly reading for this, so just what I happened to notice):

Throughout, "ERA-5" needs to be replaced with "ERA5" as that acronym does not have a hyphen in it.

Line 1, I see no need to say "this planet's" – so far as I know the jet streams are complex on all the planets (eg, Earth, Venus, Jupiter, Saturn, possibly others) we've studied them one (and given enough data, similar methods might be used for other planets)!

Line 11, I think you could and should leave out "or reduces" ("highlights" by itself covers that by implication).  Also "exist" should be "exists" and "characterise" should be "characterises" (refers to "Each", which is considered singular).

Line 23, "most popular" is a subjective assessment, you could just say something like "Among the most commonly used approaches is to develop…"

LInes 46–47, should be something like "...where there are extreme temperature and vertical pressure gradients…"  (and I don't think "vertical" is needed here).

Line 68, delete comma after "detect".

LIne 71, I think this should be plural possessive, thus "jet streams' "

Line 89, delete comma after "and"

Lines 109–110, It would be clearer (because this is the point that Manney & Hegglin specifically discussed) to move the Manney & Hegglin reference to immediately follow "synoptic-scale events".

Line 115, "propagations" should be "propagation"

Line 116, wording not clear, do you mean "...are 'jet streams' such as those driven by eddy- or thermal processes…"  Also doesn't "driven by eddy or thermal processes" pretty much cover all 'jet streams', thus not really needed? (Unless you mean to say something like "...are 'jet streams' nor do they diagnose the eddy or thermal processes driving them…")

Lines 179–181, I'm not sure this is even worth saying, since it would be unacceptable for any package aimed at community use not to have this flexibility.

LIne 237, should note that it is lower tropospheric u-component data that are used here.

Line 238, need serial (Oxford) comma after "North Pacific"

Line 255, "these metric" should be "these metrics" and "which" should be "that".

Line 283, "which" should be "that"

Lines 285, 288, 289, probably other places, in this usage "jet-cores" should be "jet cores" (two words, not hyphenated).

Line 301, delete comma after "calculation".

Line 322, at least add "e.g.," before these two references (and / or "and references therein" after) since there are many, many papers that discuss this.

Line 353, do you mean that all the jet core algorithms use a wind speed threshold (in which case there should be a comma between "algorithms" and "which") or that it includes none of the ones that do use a windspeed threshold (in which case "which" should be "that")?

Line 381, should be "...similar metrics, e.g., for calculating…" (unless you are saying the following is an exhaustive list, in which case "i.e." is appropriate but the commas are still needed).

Line 382, 394, 396 "which" should be "that".

---

## Author Response (AR1)

*jsmetrics v0.1.1: a Python package for metrics and algorithms used to identify or characterise atmospheric jet-streams.*

**Response to Editor**

Dear Editor,

We feel we have dealt with the changes suggested by the two reviewers. This includes a substantial amount of work which we have put into updating jsmetrics' documentation and the text and figures in the manuscript to alleviate concerns raised by the reviewers. The *jsmetrics* software has been updated to version 0.2.0 since the initial submission and a range of detailed examples of use cases has been added to the online documentation. Specifically, we have added 450 lines of text and 3,412 words to the documentation (available here: https://jsmetrics.readthedocs.io/en/latest/) since version 0.1.1.

It is worth highlighting at this point, that we have also moved over to using daily resolution data in figures instead of monthly resolution. This has resulted in changes to those figures, although the difference is rarely substantial.

We now respond individually to each specific reviewer comment.

**Response to Reviewer #1**

We thank reviewer 1 for their time and effort in giving thorough feedback to our manuscript. We have found your comments very helpful, and we hope these have gone towards improving the manuscript significantly. Please find our response to your comments below.
* * *
**General Comments:**

*Reviewer: Overall, the manuscript is close to being in a publishable form. It is well written, and the software presented appears to address a legitimate need in the larger community. I especially appreciate that they lay out a process for the expansion of the software into a tool that might really serve the community well.*

**Author Response:** Thank you for this comment.

*Reviewer:* *"I have mostly minor concerns, the largest of which is the lack of demonstration of a nontrivial element of the software. The authors do not give any motivation for this omission, although it could be justified. Still, it left this reviewer with the impression that either the manuscript or the package was a bit incomplete without some acknowledgement or demonstration of the full capabilities of the software. See more detailed comments about this below".*

**Author Response:** We acknowledge this concern and we have, in conjunction with preparing revisions to this manuscript and edits to the documentation, motivated the lack of demonstration of a non-trivial element of the software within the manuscript. Firstly, since the original submission, we have invested a significant amount of time working on various detailed worked examples in the online documentation (find these here: https://jsmetrics.readthedocs.io/en/latest/usage.html), to help onboard any of prospective users and provide a framework for using the package for wider research purposes. For this, we have also now providee examples for every metric currently included in the package, and these are available in the form of Jupyter notebooks from https://github.com/Thomasjkeel/jsmetrics-examples.

Secondly, since our original submission, we have also been in a process of preparing a manuscript using *jsmetrics* on the outputs of the CMIP6, and as part of this we have now released some of the 'analysis runner' scripts (which you ask for later into your review) which have helped us run *jsmetrics* in batch on JASMIN. These scripts are available at: https://github.com/Thomasjkeel/jsmetrics-analysis-runner. However, we would like to note that we have not yet found time to provide these scripts with a level of documentation that we have for the *jsmetrics* package itself (as of Oct 2023).

*Reviewer:* *"Another larger concern is the authors do not provide a rationale for which packages were chosen for the initial release. This is easily remedied, but it does make evaluation of the completeness of the package difficult. At present, there are several metrics broadly used by the community that are left in the "Future Work" section (Table 5) which might greatly increase the usefulness and adoption of this software. A more thorough justification of which metrics were implemented before first release (which, by extension, helps explain why some were not) would alleviate these concerns".*

**Author Response:** We agree, and so have made changes to the text (lines 76-77) to included a rationale for which packages were chosen:

*"These have become the initial set of metrics included based on firstly, their ease to implement into Python, and secondly on their relative popularity in the wider literature".*

*Reviewer:* *"Finally, a suggestion for the authors, which they may freely reject, is to consider more quantitative ways of evaluating the discrepancies between metrics. Given the present manuscript generally only compares broad patterns/trends seemingly drawn from visual inspection of the figures, there is not much that can be robustly concluded from the comparisons done here. I recognise that the purpose of the manuscript is more about the software than its application, but perhaps a more robust and consequential application would make the software more attractive to potential users.*

**Author Response:** Thank you for this suggestion, we have made some edits to the text that consider the quantitative discrepancies (mean and IQR) between the metrics. These changes are made on: lines 271-278, describing Figure 2; lines 348-356, describing Figure 5+6.

*Reviewer: One suggestion in this vein is to compare the results of a metric against quantifiable parts of its definition to demonstrate how the definition of a metric may explain discrepancies across metrics. This is an important part of identifying structural uncertainty -- identifying its primary source. The manuscript at present raises this question and posits some answers, while some more rigorous answer of the question would be possible with minimal additional analysis. I think this would be a nice addition to the manuscript, but the authors may legitimately determine it is out of scope. If there are concurrent efforts to do this kind of extension as a separate work, perhaps they could be detailed more in the conclusions".*

**Author Response:** Thank you for your suggestion, although we hope in our above response we have provided a suitable alternative to this. We decided against focusing on an analysis of the quantifiable part of an individual metric's definition because, as Dr Manney highlighted in her review, this sensitivity analysis is often done in developing the original methodology, and further, it may not be entirely and uniformly possible for each metric. We have thus deemed it to be out of the scope of our intention with this manuscript, unfortunately.
* * *
**Specific Comments:**

*Reviewer: Line 11: "Each of these…" subject-verb agreement*

**Author Response:** We have since removed this sentence from the manuscript because of comments made in Dr Manney review to include more description of the results and findings of the paper.

*Reviewer: Line 50: Perhaps some mention should be made that these metrics focus entirely on tropospheric jet streams*

**Author Response:** Thank you for this suggestion, it was perhaps not very clear that all metrics we focus on tropospheric jet streams exclusively. We have added: *"The jsmetrics package, introduced in this paper, focuses exclusively on metrics for tropospheric jet streams"* (lines 61-62).

*Reviewer: Line 53: While generally true for the existence, not always the case for variability/forced response. See Menzel et al. 2019.*

**Author Response:** This is a very good point, and thank you for pointing us in the direction of this research. We have edited the text to highlight this debate (lines 49-53):

*"While eddy-driven processes tend to produce jet features that are deeper and more variable in their location and strength, thermally-driven processes produce jet features that are more shallow, narrow, and less latitudinally variable (Harnik et al.,2014; Lachmy and Harnik, 2014; Madonna et al., 2017; Menzel et al., 2019; Stendel et al., 2021). The position of the thermally-driven processes*

*are connected to the edges of the Hadley Cell it, although recent work suggests this only a loose connection (Menzel et al., 2019)".*

*Reviewer:* Line 63: Suggest rephrasing.

**Author Response:** Thank you for your suggestion, we have rephrased this to (lines 65-66):

*"So it is important that we are able to assess uncertainties involved in representing the jet streams in data, and further, to know how they are responding to climate change (Gulev et al., 2021; Lee et al., 2019)".*

*Reviewer: Line 65: Suggest rephrasing to improve clarity and specificity. Something like "Understanding how jet streams operate between seasons, between phases in climate oscillations, and in response to human activities could enable projections about the regimes of (extreme) surface weather across timescales."*

**Author Response:** We have updated the text (lines 65-67)

*Reviewer: Line 73: One general comment - it would be helpful to get some more justification from the authors about the specific methods used in the initial release. What was the motivation for including the methods that are here and not some of the ones mentioned which are planned for future implementation? Whether methods were chosen based on an assessment of their relative importance or relative ease of implementation, or some combination of these or other factors would be helpful to be aware of. In particular, there are some important but more complex methods which are not yet implemented (e.g., jet latitude index, local wave activity etc.) and the reader is left to assume about why they were not implemented in the initial release. There is a clearly a lot of resources invested into the metrics included, and some discussion of the strategy for choosing this would be warranted*

**Author Response:** We thank the reviewer for their suggestion and have now made amendments to the text to include a justification of which specific methods were included in the initial release. As we chose these based on relative ease of implementation, we hope to have made this clear in the text (lines 76-77):

*"These have become the initial set of metrics included based on firstly, their ease to implement into Python, and secondly on their relative popularity in the wider literature".*

*Reviewer: Line 81: suggested rephrase: "approaching understanding" -> "understanding"*

**Author Response:** Done (line 86).

*Reviewer: Line 90: suggested rephrase: "However, by isolating lower-level winds, these methods may miss aspects of jet streams whose eddy-driven components do not extend throughout the atmospheric column within the method's given time window."*

**Author Response:** Thank you, we have edited this text (lines 95-97).

*Reviewer:* *Line 101: suggested rephrase: "so they can be used"*

**Author Response:** Done (lines 107-108).

*Reviewer:* *Line 110: suggested rephrase: "so they are less appropriate"*

**Author Response:** Done (line 116).

*Reviewer:* *Line 124: It may be worth noting that use of the meridional circulation index itself is highly debated (see Barnes 2013). Also consider referencing Blackport and Screen 2020, Blackport et al. 2019 to capture both sides of the debate*

*&*

*Line 125: suggest rephrasing (run-on)*

**Author Response:** Yes, we agree, and in accordance with comments made by the other reviewer, we have updated the text to include a more detailed discussion of the debate around jet waviness from lines 129-137. Regarding the additional references we have added these in our note of the debate in lines 135-137.

*Reviewer:* *Line 135: suggested rephrase: "they provide relatively more detail"*

**Author Response:** Done (line 147).

*Reviewer:* *Line 142: suggested rephrase: "these methods can discount the influence of multiple jet streams…"*

**Author Response:** Done (line 156).

*Reviewer:* *Line 147: Perhaps another limitation of threshold-based metrics is that they may not operate properly in climate regimes very different from the present (e.g., RCP8.5)*

**Author Response:** Thank you for your suggestion. We have included this further limitation of threshold-based metrics to lines 157-160:

 *"Furthermore, they may also underestimate positions of the jet cores in different seasons, in climate regimes different from the present (e.g. SSP5-8.5), and within different phases of the given climate oscillations (e.g. Woollings et al., 2010; Madonna et al., 2017; Manney and Hegglin, 2018; Manney et al., 2021)."*

*Reviewer:* *Line 170* Perhaps xarray's interoperability with netCDF and GRIB data formats could be advertised.

**Author Response:** Thank you for this suggestion, we have updated the text to include a mention of this (line 190):

*"The package is built using xarray — an open-source Python package for working with labelled multidimensional arrays that has become a popular package for Earth Science research (due to its ability to interface with NetCDF4 and GRIB data formats; Hoyer et al., 2023)".*

*Reviewer:* *Line 180: In general I think this design choice is a good one, but it does come with trade-offs which might be acknowledged/mitigated. One concern with this design choice is the atomization of code which can make maintenance and readability more challenging as it requires a maintainer or reviewer to search through many potential code paths. A small example from the codebase. Jsmetrics.metrics.jet_statistics_components.get_atm_mass_at_one_hPa is essentially the same function as calc_atmospheric_mass_at_kPa but does a basic unit conversion prior to the calculation (i.e., divides by 10). The granularity of the component functions has some trade-off with the eventual or actual reusability, and how flexibility in the codebase is implemented is something to keep in mind or even potentially discuss further at this junction.*

**Author Response:** We thank the reviewer for this suggestion. This philosophical decision can make up for an awkwardness, we now appreciate although it is justifiable, it requires a lot more documentation than was earlier provided. Which has been made clear by the comments by Dr Manney. In response to this, we have made substantial edits to the documentation provided online and in the code. In the manuscript we have added some new information to highlight potential limitations of using flexible base function (lines 201-205):

*"Unfortunately, this flexibility requires making all base functions as simple and one-use as possible, which has sometimes led to a decrease in readability. For example, it became necessary to keep some base functions more specific, which may make some of these harder to use, without a familiarity with the package nor more advanced Python knowledge. We hope to have alleviated any loss of readability, with the use of more verbose naming conventions throughout the package and detail in the individual method's docstrings".*

*Reviewer:* *Line 255: This explanation requires some clarification. Based on Table 1, the Barnes and Simpson method is the only one that is not using an interpolation to a higher resolution. The others are using parabolic/quadratic/cubic interpolations. Yet the way this sentence currently reads, it sounds as if Barnes and Simpson's interpolation scheme is somehow what makes it different from the others. If this is not intended, please clarify. If this is the explanation, it is unclear to me how the quadratic interpolation scheme is different enough from the others to cause such a discrepancy. Also I think Table 1 should then include some mention of the interpolation scheme if it is significant enough to cause such variation between metrics. It seems to me, based on Table 1, an alternative explanation might be the altitude utilized for jet identification. Figure 5 backs up this interpretation - when Barnes and Simpson is run on the same levels as the others, it perform much more consistently with the other metrics. This is discussed in the paragraph following the explanation,*

*but it is currently not clear how these two explanations are connected, or it seems as if the difference in pressure levels is a second-order explanation rather than first-order.*

**Author Response:** Thank you for highlighting this issue. To resolve this, in conjunction with comments made by the other reviewer, we have adjusted the text to remove this original explanation, and instead broadly highlighted the differences arising from the data inputs (pressure level and region) (now lines 279-282). We would also like to note that we have moved from monthly data to daily data, so the interpretation of this figure we originally provided is now not valid, and as such, we felt it may not be in the scope of this section to cover discrepancies arising from the method's interpolation strategies

*Reviewer: Line 255 (cont.): I think a lot of my confusion here could be cleared up by more careful language enumerating exactly which metrics are being discussed. If the constant referencing of different metrics becomes cumbersome, you may consider abbreviating each metric name in Table 1. Maybe something like BP15, BP13, GP17, etc. I leave that decision up to the authors, but I think having a small, concrete label for each algorithm might help with clarity overall.*

**Author Response:** Thank you very much for this suggestion, we have made changes throughout the text to include new metric codes (e.g. W10, BP13, BP15, etc.), which we hope make it clearer which metric is being discussed.

*Reviewer: Line 260: However, there is generally good agreement between Bracegirdle 2018 and Barnes and Polvani 2013, one of which uses a single level and the other does not, so I'm not sure this blanket explanation works for all algorithms. I think focusing the explanation on the largest discrepancies is best. Alternatively, consider plotting statistics as a function of the input parameters, such as variance as a function of altitude or resolution. This would then very clearly support any explanations regarding discrepancies between metrics.*

**Author Response:** As we mention in our responses above, we have changed the interpretation of this figure in this section, so we no longer highlight agreement of the methods based on single/multi-level data inputs (new analysis provided in lines 279-282).

*Reviewer: Line 299: I notice there is not currently a demonstration of the jet waviness metrics, which makes me wonder a bit about their inclusion into the software. I recognize there are only two at present, which make comparison difficult, and these are likely the more complex algorithms to implement of all of the included metrics. Still, perhaps some application to the cold wave event in 4.2 which features a prominent trough could at least display jsmetrics capabilities. Otherwise, it feels like they should be left in the future work column if they are not complete enough to get even some (possibly trivial) scientific result from. My opinion is a bit mixed here, as in general I don't want to suggest much additional analysis, but I think doing a bit more with these is worth the authors' consideration.*

**Author Response:** Thank you, we have now addressed this and made figures to represent the two waviness metrics included in jsmetrics in Figure 4 (see below). We have edited the text to include an interpretation of this figure in lines 325-329. We acknowledge that this may be a relatively,

trivial example of their use to diagnose the Texas Cold wave, although we hope to have alleviated your broader concern for their omission originally.

[Figure]

*Figure 4. A comparison of jet waviness during the Texas Cold Wave event between 12:00 11th-19th February 2021 as estimated by two waviness metrics available in jsmetrics. The metric from Francis and Vavrus (2015) is a mean of MCI calculated for each 6 hour time step between 12:00 11th-19th February 2021. When MCI is 0, the wind is purely zonal, and when MCI is 1 (-1), the flow is from the South (North). (Data: ERA5 climate reanalysis product; Hersbach et al., 2020).*

**Reviewer:** *Line 327: The Figure caption notes that each time window is centred on the same time, but it might be nice to include a description of the temporal window selection in the main text as well.*

**Author Response:** Thank you for this suggestion, we have now made edits to the text to include this note about the temporal window selection (line 367).

**Reviewer:** *Figure 7/8: One consideration which should be made here is the choice to use a contour plot for what is essentially categorical data (jet detected/not detected). Contour plots will interpolate between grid points in ways that may distort the underlying data when it is discrete instead of continuous. While the authors may find compelling reasons to keep the figures as they are, please at least consider using a grid-coloring map (i.e., pcolormesh) instead of a contour plot to avoid generating details which may not be present in the data. An exception here may be the 8-day counts in Figure 8. I'm not familiar enough with the post-processing technique used here to know whether the counts should be discrete or continuous, although I would generally expect counts to be discrete.*

**Author Response:** Thank you for this suggestion, you make a good point. We have changed the figures from a contour plot to 'pcolormesh' now. This makes changes to Figure 3, 7 and 8. For

Figure 8, the 8-day counts includes data that is continuous (as we fit a Gaussian filter over the counts), so we have kept the plot as a contour plot.

*Reviewer:* *Line 347: Does the estimation of latitude from jet core algorithms use any interpolation to higher resolution as is common with the jet statistic algorithms? I think whether this is done on native resolution or a higher one should be explicitly stated; it may play some part in their overall discrepancy.*

**Author Response:** Yes, we use the native resolution of the data in each case for the jet latitude estimation of the winds from the jet core algorithms. We agree it was an oversight not to mention this, and we have updated the text to highlighted this (lines 389).

*Reviewer:* *Figure 9: The text states (349) that Barnes and Polvani 2015 produces a single estimate, however it appears here to have a PDF. There are no colors corresponding to Barnes and Simpson or Bracegirdle; I suspect that their colors appear together as the grey vertical line. However, Barnes and Simpson should have a PDF, so I think there is an issue with the legend here. And if two metrics do fall at exactly the same location, please come up with a better way to distinguish them or perhaps make a note in the text, it is somewhat confusing as currently displayed.*

**Author Response:** Thank you, this was an oversight in the original manuscript that has now been fixed (see new Figure 9 below). Also, we now include a note in text to help readers distinguish the values between the methods returning a single estimate of jet latitude over this time frame: AC08, BS17 and B18 (lines 391-392). BP15 should produce a PDF.

[Figure]

[Figure]

*Reviewer:* *Line 361: is this analysis runner available to package downloaders? I could not find it in the repo. If not, why not provide some helpful scripts, perhaps pared down versions of the ones used for this work?*

**Author Response:** Development work was still continuing on the analysis runner during the time of the original submission, but this has now been completed (in conjunction with another manuscript on the North Pacific Jet we have been preparing). We are happy to release this code, and have done so now (find the code here: https://github.com/Thomasjkeel/jsmetrics_analysis_runner/tree/main).

However, we have not found time to provide it with a level of documentation that we have for the jsmetrics package itself (as of Oct 2023).
* * *
**Response to Reviewer #2**

Dear Gloria,

Thank you for your giving your time to provide feedback for our manuscript. You have provided very helpful comments and suggestions, and these have gone a long way towards improving our manuscript.

Please find our response to your comments below. In this response, we hope to have addressed both changes to this manuscript, and changes to the *jsmetrics* online documentation (with which we will provide *before and after* screenshots throughout this response).
* * *
**General Comments:**

*Reviewer: Recommendation: I raise some questions about the maturity of the jsmetrics package/documentation and thus the current suitability for publication in GMD. Substantial revisions and clarifications in the manuscript, and improvements in the code documentation, would be needed before publication.*

**Author Response:** We recognise your concern here. We had originally thought that the package had been sufficiently developed for publication. In light of both your, and the other reviewers, comments, this was not the case. Since our submission, we have invested significant effort into providing more complete online an in-code documentation and introduced new usage guidelines with examples. Each of these will be highlighted in this response using *before and after* screenshots.

We also hope to have made substantial clarifications in the manuscript that you call for here, especially around the approach we take to introduce and describe the content of the package.

*Reviewer: "After reading the paper, installing (via pip, so presumably latest stable version as of late July 2023) and trying out the package, and looking at the documentation both on GitHub and the site given in Keel (2023), I have several reservations about this package as it now stands regarding its maturity,its interpretation and implementation of some of the methods, the frugality of the information returned by the package, and the opaqueness of the documentation and usage guidance".*

**Author Response:** We appreciate these reservations about the package at the time of your review. Since then, we have invested a lot of time dramatically improving docs (hosted at: https://jsmetrics.readthedocs.io/en/latest/index.html). Below, we provide screenshots of the documentation at version 0.1.4 (Published 8th August 2023) and then again at version 0.2.0 (Published 14th October 2023). We note that, the original manuscript describes version 0.1.1, but version 0.1.4 included no updates since the documentation of 0.1.1.

Below we show changes to the usage guidelines and an example of the changes to the docstring of the Koch et al. 2006 metric:

*Usage guidelines updates:*

*Before (Screenshot is of version 0.1.4 (Published 8th August))*

[Figure]

*Current (Screenshot is of version 0.2.0 (Published & Accessed 14th October))*

[Figure]

_Metric descriptions (docstring) updates:_

*Old (Screenshot is of version 0.1.4 (Published 8th August))*

**jsmetrics.metrics.jet_core_algorithms.koch_et_al_2006**(*data, ws_threshold=30*)    [source]

Calculates the weighted average windspeed and applies a threshold to identify the jet. The actual methodology uses 100-400 hPa and 30 ms^-1 as the windspeed threshold.

weighted average windspeed = 1/(p2-p1) integral(p2, p1)(u^2+v^2)^(1/2)dp where p1, p2 is min, max pressure level

Method from Koch et al (2006) https://doi.org/10.1002/joc.1255

| Parameters: | **data : xarray.Dataset** |
|---|---|
| | Data containing u- and v-component wind |
| | **ws_threshold : int or float** |
| | Windspeed threshold for jet-stream (default: 30 ms-1) |
| **Returns:** | **weighted_average_ws : xarray.Dataset** |
| | data containing weighted average ws above windspeed threshold |

`jsmetrics.metrics.jet_core_algorithms.koch_et_al_2006`*(data, ws_threshold=30)*    [source]

This method follows a two-step procedure used to detect jet-event occurences (here: 'jet_events_ws').

The weighted average windspeed ($avel$) for the jet events is calculated as follows:

$$avel = \frac{1}{p2 - p1} \int_{p1}^{p2} (u^2 + v^2)^{1/2} \, dp$$

where $p1, p2$ is min, max pressure level.

After calculating $avel$, in a second step a windspeed threshold to isolate jet events (the default is $30ms^{-1}$).

This method was first introduced in Koch et al (2006) (https://doi.org/10.1002/joc.1255ᶜ) and is described in section 2.2.2 of that study. The original methodology provides a third step (to produce a climatology of jet events), but this has been ignored in this implementation. Instead, we have provided an example of how to calculate this after running this method in 'Examples' below.

Please see 'Notes' below for any additional information about the implementation of this method to this package.

| Parameters: | data : xarray.Dataset |
|---|---|
| | Data which should containing the variables: 'ua' and 'va', and the coordinates: 'lon', 'lat', 'plev' and 'time'. |
| | **ws_threshold : int or float** |
| | Windspeed threshold used to extract from weighted average (default: 30 ms-1) |
| Returns: | output : xr.Dataset |
| | Data containing the output variable 'jet_events_ws' |

**Notes**

The original methodology uses windspeed between 100-400 hPa to calculated the weighted average and 30 meters per second as the windspeed threshold.

This equation for this method is provided on pg 287 of the Koch et al. 2006 paper.

In the original paper, they accumulate the jet events into two-class jet typology (described in section 2.2.3 of Koch et al. 2006)

**Examples**

```python
import jsmetrics
import xarray as xr

**Load in dataset with u and v components:**
uv_data = xr.open_dataset('path_to_uv_data')

**Subset dataset to range used in original methodology (100-400 hPa)):**
uv_sub = uv_data.sel(plev=slice(100, 400))

**Run algorithm:**
koch_outputs = jsmetrics.jet_core_algorithms.koch_et_al_2006(uv_sub, ws_threshold=30)

**Produce climatology of jet occurence events for each season and each month:**
koch_month_climatology = koch_outputs.groupby("time.month").mean("time")
koch_season_climatology = koch_outputs.groupby("time.season").mean("time")
```

*Reviewer: "(1) Of the paper types listed as suitable for GMD, the category this would fit into is "methods for assessment of models". The description of this paper type includes "..description of a fully fledged software tool…". I question whether this package can at present be considered "fully fledged". For example, of the 17 metrics (summing the three types) described and exemplified in the paper, on the website, four of them are listed as complete, one as "in progress" with a note indicating "need help", and the rest as "to verify". As far as I can tell (e.g., extrapolating from Fig. 1 in the paper), "to verify" means that the implementation has not yet been verified against results from the original paper. Even if thus limited only to the approaches discussed in the paper (where the reader is likely to assume that they have all been fully tested), I question whether this can be considered a mature software package. I recognize that the authors are hoping to make this a community project and entrain outside contributors, but if this manuscript is a call for participants – without a basic foundation of mature software that is to be built upon – then I am not sure it is appropriate for publication in GMD at the present state of maturity of the software package".*

**Author Response:**

In improving the documentation provided for this package, we also hope to have alleviated some of your broader concern regarding the implementation and interpretation of the methods in jsmetrics. We now include detail descriptions of the methods and relevant examples of their use within their docstring (see screenshot in our previous response). We have also added a notes section to each method which detail more specific implementation details (see example from the Koch et al. 2006 algorithm below).

Regarding the concern for metric validation, we have been able to finalise and verify 12 of the 17 included in the package as of now. We have achieved this either by producing the same results from the original paper, with presenting the outputs to the original authors (Tim Woollings and Paulo Ceppi), or by carefully studying the equations given in the original papers. For the remaining left to verify, including your M11 metric, we hope to continue a dialogue with you, and take your advice to continue reach out to other authors whose metrics we felt needed some clarification.

*Example of Notes and Examples section of a metric's docstring (Screenshot is of version 0.2.0 (accessed 14ᵗʰ October 2023)):*

**Notes**

The original methodology uses windspeed between 100-400 hPa to calculated the weighted average and 30 meters per second as the windspeed threshold.

This equation for this method is provided on pg 287 of the Koch et al. 2006 paper.

In the original paper, they accumulate the jet events into two-class jet typology (described in section 2.2.3 of Koch et al. 2006)

**Examples**

```python
import jsmetrics
import xarray as xr

**Load in dataset with u and v components:**
uv_data = xr.open_dataset('path_to_uv_data')

**Subset dataset to range used in original methodology (100-400 hPa)):**
uv_sub = uv_data.sel(plev=slice(100, 400))

**Run algorithm:**
koch_outputs = jsmetrics.jet_core_algorithms.koch_et_al_2006(uv_sub, ws_threshold=30)

**Produce climatology of jet occurence events for each season and each month:**
koch_month_climatology = koch_outputs.groupby("time.month").mean("time")
koch_season_climatology = koch_outputs.groupby("time.season").mean("time")
```

*Current Metric verification status:*

| Metric/Algorithm | Status | Metric/Algorithm | Status |
|---|---|---|---|
| Gallego et al. 2005 | To start | Strong & Davis 2005 | To start |
| Koch et al. 2006 | Complete | Archer & Caldiera 2008 | Complete |
| Schiemann et al. 2009 | Complete | Woollings et al. 2010 | Complete |
| Manney et al. 2011 | To verify | Allen et al. 2012 | To start |
| Barnes & Polvani 2013 | Complete | Pena-Ortiz et al. 2013 | To verify |
| Screen & Simmonds 2013 | In progress* | Kuang et al. 2014 | To verify |
| Barnes & Polvani 2015 | Complete | Francis & Vavrus 2015 | Complete |
| Cattiaux et al. 2016 | To verify | Barnes & Simpson 2017 | Complete |
| Chenoli et al. 2017 | In progress | Grise & Polvani 2017 | Complete |
| Molnos et al. 2017 | In progress* | Adam et al. 2018 | To start |
| Bracegirdle et al. 2018 | Complete | Ceppi et al. 2018 | Complete |
| Kern et al. 2018 | To start* | Rikus 2018 | In progress |
| Zappa et al. 2018 | Complete | Kern & Westermann 2019 | To start |
| Kerr et al. 2020 | To verify | Maher et al. 2020 | To start |
| Winters et al. 2020 | To start | Martin 2021 | To start* |
| Bosiger et al. 2022 | To start | Local Wave Activity | In progress* |

\* == help needed

*Reviewer: "(2) I question the distinction (and the need for one) between "jet statistics" and "jet core algorithms". All of these methods aim to extract basic information such as the jet latitude (and altitude for several of them) and windspeed. For example, the method of Manney et al. (2011, hereinafter M11; though the package is formally named JETPAC, that was not mentioned in the first couple of papers) was based on extending that of Koch et al. to characterise the jets in the latitude/altitude plane rather than only getting horizontal locations and vertically averaged dynamical characteristics. But the two papers (as well as Manney et al., 2014) have very similar aims and show very similar diagnostics (eg., frequency distributions of jet cores and associated winds) – so why is one "jet statistics" algorithm and the other a "jet core algorithm"? I see no need for a distinction here".*

**Author Response:** Thank you for your comment, we have since clarified our distinction between 'jet statistics' and 'jet core algorithms' (lines 9-12 & 25-31). We would like to again highlight that this decision was made due to the dimensionality of the outputs. Whereas the jet statistics return a single value, the jet core algorithms work to provide a multidimensional mask of the outputs. Our

motivation was that this would allow the user to use the mask provided by a given jet core algorithm to extract other quantities (winds, altitude, etc.), and we have now provided examples in the documentation of exactly this.

We have updated the description for jet statistics and jet core algorithms when they are first introduced in the introduction (lines 9-12), and within Section 2.1 (lines 80-81) & 2.3 (lines 139-141), to reflect this clarification/distinction.

We also have updated the online documentation (https://jsmetrics.readthedocs.io/en/latest/metrics.html).

*Current description of jet statsitics, jet core algorithms and waviness metrics in the online documentation (accessed 16th October 2023):*

[Figure]

*Reviewer: "(3) Obviously, there is one method herein with which I'm very familiar (and to a lesser degree several of the other methods, either because mine built on them, they (apparently) built on mine, or I reviewed the original papers. I looked closely at their implementation of the M11 method, and I cannot from the information provided (nor from running an example of their implementation) convince myself whether or not what they are doing closely resembles the M11 method*

*[…]*

*I've gone into (probably tiresomely) lengthy detail here because this case (and the difference in the descriptions of M11 and PO13, which are very similar methods) makes me question whether there may be other metrics for which the implementation may not accurately reflect the original method. If I had been putting together a package of this sort, I would have contacted the authors of the original papers and tried to verify my understanding of the metrics and what was fundamental to them – has this been done for any of the methods? You probably would not have gotten a 100% response rate, but many authors (like me) would have been happy to confirm the algorithm and to support your efforts to the extent we could find the time".*

**Author Response:** You were correct, it was a mistake and oversight to originally include the Manney et al. 2011 method in this manuscript, as it was the only method in the 'in progress' phase

of development at the time. In particular, the originally implemented code listed as M11 was just an initial foundation and boilerplate. Since your review, we have taken care to read and re-assess the description of M11 from Manney et al., 2011, 2014 and Manney & Hegglin, 2018.

We include a step-by-step breakdown of how we have implemented the method to Python in a jupyter-notebook, available as part of the examples: https://github.com/Thomasjkeel/jsmetrics-examples/blob/main/manney11_algorithm.ipynb

This method, like 5 of the 17 others in this package, will be verified with the authors of the original metric (that is to say, in the 'to verify' state, we are confident that we have re-produced the metric, but think we need some further clarification for all the outputs). As such, we hope to continue a dialogue with Dr Manney about this implementation.

We would like to highlight that, the scope for this manuscript was to introduce the package and the foundation it provides for researchers working on and with jet stream metrics. While there are 17 at this time, it is likely that the package will continue to be in a relative state of continuous development, with new methods being added and finalised over time. We hope that the improvements to the documentation and clarification of how we have implemented the method will suffice for the description of the software tool as it stands.

*Reviewer: In the paper (lines 137–140)l they do not list M11 in the category "(ii) in relation to wind-speeds of neighbouring data points (local wind-speed maxima…", but this method is fundamentally based on first finding all local wind-speed maxima (in a latitude / altitude plane) and then applying thresholding and other criteria. In fact, the M11 method is fundamentally nearly identical to the Peña-Ortiz et al (2013; hereinafter PO2013) method (the latter paper showed internal evidence that they must have known of M11, but did not cite it), which is listed in both categories that would be appropriate for M11. In what appears to be the primary function to identify the cores, there is a comment saying "Core are discovered where 8-cells are above boundary threshold… Paper uses 100-400 hPa." The M11 method inherently operates on a slice of the windspeed (and other fields) with a latitude-like and an altitude-like coordinate (and then is generally run at each longitude gridpoint for each day); jet cores are identified in that latitude/altitude plane by finding the local maxima (as points with higher windspeed than the nearest neighbors in altitude and latitude) and testing for exceedance of the core threshold. Finding the "edges" of the jet regions is no more complicated (or informative!) than finding the latitude gridpoint on each side and vertical coordinate (eg, altitude) gridpoint above and below where the windspeed drops below the boundary threshold value. Other criteria that determine whether to consider two cores in the same "region" (that is, where the windspeed does not drop below the boundary threshold between them) separate are latitude distance between the cores (which this package appears to implement) and the amount by which the windspeed drops along a line connecting the cores in altitude / latitude space (which I didn't find mentioned herein or in the documentation).*

**Author Response:** We have corrected these lines and M11 has now been included in category (ii) (now lines 153-154). As we mentioned, we have also overhauled the M11 method in the package, and the description provided in the docstring of the M11. Below, we show a before and current views of the M11 documentation. We now include a more verbose description of M11 and the outputs provided, along with a note about the parameters that a user can adjust.

*Before (Screenshot is of version 0.1.4 (Published 8th August))*

**jsmetrics.metrics.jet_core_algorithms.manney_et_al_2011**(*data, ws_core_threshold=40, ws_boundary_threshold=30*)   [source]

Looks to get seperate jet cores based on boundary and threshold. Core are discovered where 8-cells are above boundary threshold Paper uses 100-400 hPa.

Method from Manney et al. (2011) https://doi.org/10.5194/acp-11-6115-2011 Also see Manney et al. 2011, 2014, 2017 and Manney & Hegglin 2018

NOTE: Currently takes a long time i.e. 7.6 seconds per time unit with 8 plevs (i.e. 7.6 seconds per day) on AMD Ryzen 5 3600 6-core processor

**Parameters:**      **data : xarray.Dataset**

Data containing u- and v-component wind

**ws_core_threshold : int or float**

Threshold used for jet-stream core point (default=40)

**ws_boundary_threshold : int or float**

Threshold for jet-stream boundary point (default=30)

**Returns:**       **output : xarray.Dataset**

Data containing jet-cores (ID number relates to each unique core)

```
jsmetrics.metrics.jet_core_algorithms.manney_et_al_2011(data, jet_core_plev_limit,
jet_core_ws_threshold=40, jet_boundary_ws_threshold=30, ws_drop_threshold=25, jet_core_lat_distance=15)
    [source]
```

This method detects jet cores (within an altitude range see 'jet_core_plev_limit') and a boundary region surrounding those cores based on two windspeed thresholds. Two checks are applied after initial detection of cores to check whether boundaries with more then one core are part of the same feature (the default threshold for these boundaries is 30 m/s, see 'jet_boundary_ws_threshold'). The two checks seperate cores based on whether the cores are more than a certain distance apart (default is 15 degrees, see 'jet_core_lat_distance') and whether the windspeed between two given cores does not drop below a windspeed threshold (default is 25 m/s, see 'ws_drop_threshold')

**This method returns four outputs**

1. **jet_core_mask** – Regions within each latitude-altitude slice that are local maxima and have windspeeds above the 'jet_core_ws_threshold'
2. **jet_region_mask** – Regions above, below, left and right of any given jet core with windspeed above the 'jet_boundary_ws_threshold'
3. **jet_region_contour_mask** – All contigious regions of windspeeds emcompassing a jet core above the 'jet_boundary_ws_threshold' (i.e. not just above, below, left and right)
4. **ws** – Resultant wind speed calculated from 'ua', 'va' inputs.

This method was originally introduce in Manney et al. (2011) (https://doi.org/10.5194 /acp-11-6115-2011⤴), and is described in Section 3.1 of that study. This method is also known as the JETPAC (Jet and Tropopause Products for Analysis and Characterization) software package, and available in its original form at request to NASA JPL.

Please see 'Notes' below for any additional information about the implementation of this method to this package.

| Parameters: | data : xarray.Dataset |
| --- | --- |
| | Data which should containing the variables: 'ua' and 'va', and the coordinates: 'lon', 'lat', 'plev' and 'time'. |
| | **jet_core_plev_limit: tuple or array** |
| | Sequence of two values relating to the pressure level limit of the jet cores (original paper uses 100hPa 400 hPa) |
| | **jet_core_ws_threshold : int or float** |
| | Threshold used for jet-stream core point (default=40 m/s) |
| | **jet_boundary_ws_threshold : int or float** |
| | Threshold for jet-stream boundary point (default=30 m/s) |
| | **ws_drop_threshold : int or float** |
| | Threshold for drop in windspeed along the line between cores (default: 25 m/s) |
| | **jet_core_lat_distance : int or float** |
| | Threshold for maximum distance between cores to be counted the same (default: 15 degrees) |
| Returns: | **output : xarray.Dataset** |
| | Data containing the four output variables: 'ws', 'jet_region_mask', 'jet_region_contour_mask', and 'jet_core_mask' |

**Notes**

The implementation of this method varies slightly from the original, because this method will return a mask rather than dynamical values, the intention was to allow these masks to be used to subset other variables such as windspeed (see 'Examples' for demonstration of how to use the mask).

There is an update to this method introduced in Manney & Hegglin 2018 to include physically-based method to extract the subtropical jet is identified (and thus distinguished from polar jets).

'jet_region_above_ws_threshold_mask' is provided here as a alternative to using a contour to check which regions encompass jet cores.

**Examples**

```python
import jsmetrics
import xarray as xr

**Load in dataset with u and v components:**
uv_data = xr.open_dataset('path_to_uv_data')

**Subset dataset to range appropriate for original methodology (100-1000 hPa)):**
uv_sub = uv_data.sel(plev=slice(100, 1900))

**Run algorithm:**
manney_outputs = jsmetrics.jet_core_algorithms.manney_et_al_2011(uv_sub, ws_core_threshold=4

**Use the jet core mask to extract the jet windspeeds**
manney_jet_ws = manney_outputs.where(manney_outputs['jet_core_mask'])['ws']
```

*Reviewer: 100-400 hPa is the region to which the jet core altitude is limited, but the jet regions surrounding any core are identified at altitudes that may be above and/or below that – wherever they occur. This procedure is described in detail in M11 and Manney et al (2014), and completely but in less detail in Manney et al (2017, 2021; latter J Clim, DOI: 10.1175/JCLI-D-20-0947.1) and Manney & Hegglin (2018). It is not clear to me that either the description in this manuscript and the jsmetrics code, or the implementation in the code are doing something that is fundamentally similar to this method.*

**Author Response:** To account for this we have now explicitly included a parameter of the M11 method: *'jet_core_plev_limit'* which is described in the docstring (shown in screenshots above). This method also returns a message to the user to set this parameter to be able to run M11:

*"Please provide a pressure level limit for the jet cores returned by this metric. As an example, the original methodology uses a limit of 100-400 hPa. To replicate this, pass the parameter jet_core_plev_limit=(100,400)."*

*Reviewer: Further, in lines 155–157, they ignore the update to the M11 method provided by Manney & Hegglin (2018) (and used in all succeeding papers using the method, including Manney et al, 2021), whereby the subtropical jet is identified (and thus distinguished from polar jets) using a physically-based method. In the preceding lines, they also fail to note that PO13 themselves concluded that their method for separating subtropical and polar jets did not work well; which is consistent with the statement in Manney et al, 2014 regarding the division of subtropical and polar jets by latitude (which they used in the one case where they did not simply analyze all the jet cores at each longitude) being useful only for very broad climatological studies.*

**Author Response:** We have now updated the manuscript to include a mention of only being able to seperate Jan-Feb NH Jets from PO13 (lines 170-172), and note of the physically based subtropical jet separation introduced Manney & Hegglin (2018) (lines 172-174). We also include a mention of this update in the Notes of the method's docstring (see screenshots shown in the previous responses).

*Lines 170-174:*

*"PO13 develop a method to distinguish between merged and separate states of the polar and subtropical jets after the initial detection of jet cores, but were only able to separate the Northern*

*Hemisphere subtropical jet in Jan-Feb. The M11 method was extended by Manney and Hegglin (2018) which introduces a physical-based identification of the subtropical jet (based on the thermal tropopause altitude), to more robustly separate it from the polar jet."*

*Reviewer: "(4) The information that is provided as the output from running these algorithms does not include some of the most fundamental outputs. Continuing to use M11 as an example, the fundamental output provided by M11 (ignoring their very complete characterization of numerous other dynamical fields at the jet core locations) is not just jet core latitude, but also jet core winds (windspeed, zonal and meridional winds), and jet core altitude (PO13, at least, also provide these characteristics, and I presume others do as well); in M11 the characterization of the boundaries of the jet regions is secondary information, they focus on full characterization of the jet cores. These are all very fundamental quantities that are used in virtually all studies of jets and their trends (omitting altitude for methods that don't provide that – but for methods that do it is one of a several fundamental characteristics that are expected to change with climate change, see, e.g., discussion / references in Manney & Hegglin 2018) and in most cases (certainly for M11) these quantities "come along" with (because they are a fundamental part of) the jet core identification / characterisation. Thus it seems to me an important (and easily remedied) omission to not provide these quantities in the output".*

**Author Response:** Thank you for your advice, and we have since developed a more careful description of the scope and distinction of the *jet core algorithms* included in jsmetrics (lines 30-31 & 139-141). We have also provided more detailed usage cases in the documentation. Again, we reiterate that the package was developed with a generality in mind, so all the jet core algorithms, including M11, provide a mask which can be used to extract any fundamental quantities (such as the altitude, which is inherently provided by the multidimensional mask output of the jet core algorithms in jsmetrics). We hope to have supplemented the standardisation and usability of the jet core algorithms, with in-code and online documentation which now details more thoroughly how we intended these metrics to be used i.e. allowing for them to be run to extract the fundamental outputs of their original use, as well as for a more varied use and cross-metric comparisons.

*Reviewer: "(5) I find the documentation – in the GitHub project, in https://jsmetrics.readthedocs.io/en/latest/, and in the code itself – opaque, hard to navigate, and incomplete (though the last may be an artifact of the previous issues). Indeed, part of point (3) above could be because I could not find complete descriptions of the algorithm as implemented for each of the methods. Another example is the "Usage" information, which says that to run the Kuang et al (2014) algorithm, after reading/formatting a u & v wind dataset in uv_data, you would type: k14 = jsmetrics.jetstream_algorithms.kuang_et_al_2014(uv_data) Whereas I had to type (syntax obtained from looking through the code): k14 = jsmetrics.metrics.jet_core_algorithms.kuang_et_al_2014(uv_data) (I cannot find any objects with names containing "jetstream_"). I did, finally, find some end-to-end usage examples (that have correct syntax) by following several layers of links from the main README.rst file in the GitHub project – this information should be in https://jsmetrics.readthedocs.io/en/latest/ (in a corrected / expanded usage section, and either in an examples section like I've seen in other packages I've installed with this format of documentation, or with the API reference information as examples for particular metrics / algorithms)".*

**Author Response:** We hope that you will find that we have significantly improved the ReadTheDocs (**https://jsmetrics.readthedocs.io/en/latest/**) since your review and with this we

hope to address each aspect of your comments here. It was a big oversight to rush the online documentation ready for this manuscript submission, so we hope that it is much more sufficient now and that we have been able to highlight the changes in screenshots in this response.

We also note that we have put work into provide Jupyter Notebooks which detail a uses of all of the jet statstics, waviness metrics and jet core algorithms included in *jsmetrics* (available at: https://github.com/Thomasjkeel/jsmetrics-examples).

*Reviewer: "The "all metrics" link gives a 404 error."*

**Author Response:** Thank you for pointing this out, we have now checked and fixed this link on the readthedocs and Github.

*Reviewer: "There do not appear to be any API-type descriptions of the "Metrics", thus zero guidance on how to run them; the API-type descriptions of the "Algorithms" are not complete enough to be useful. For instance, basic information for formatting the input fields (whether the algorithm needs one level or multiple levels; if it works on one level at a time, does it handle, and how does it handle, multiple levels) should be given. There is also little or no information about what the output Dataset contains in each case. There should be a full API reference that describes each routine in enough detail that a user with basic familiarity with python can run them. (I ran several of them, but generally had to go to source code and even then guess at some things to see how to do that.) In the API-type information given for the "Algorithms", the descriptions of the methods are not clear (see discussion re M11 above), nor sufficiently detailed to know what the implementation actually does.*

**Author Response:** We have approached this by improving and adding a code example to the documentation of each metric, available here: https://jsmetrics.readthedocs.io/en/latest/metrics.html. This should hopefully meet a requirement of helping users with only a basic familiarity with Python , to use the package and be able to interpret the method and outputs.

*Reviewer: The code itself, for the vast majority of routines that actually do things (as opposed to just set up inputs, etc) are going to be very opaque to anyone who isn't fluent in the full object-oriented syntax / usage – that class includes many, many python users in the atmospheric science / climate community (partly because python can be used so effectively without even getting to using it in true object-oriented implementations). I'm not criticizing how the code is written, but doing it this way makes it critical to document what it actually does so the user doesn't have to be able to read the code at that level. (The page at the "issues" link gives descriptions of the algorithms directly from the original papers, but nothing that shows how that is "translated" into the implementation here. Being able to see the pseudo-code that is mentioned in the paper would be extremely helpful as one way to do this.)"*

**Author Response:** We hope to have alleviated some of your concern by spending a lot more time working on the in-code and online documentation for each method as well as providing worked examples. One major change is the minimal worked examples now available on ReadTheDocs (https://jsmetrics.readthedocs.io/en/latest/usage.html), with explanations for each of the three types of methods in the package. We have included figures and in-line comments to help readability and to onboard potential users with an interested in using this package for research.

*Reviewer: Further details are given in the specific comments. Altogether, the package and its documentation appear sufficiently immature to question "announcing" it in a paper.*
* * *
**Specific Comments:**

*Reviewer: Lines 1–17, The abstract would benefit from spending less time on motivation and more on* describing the actual results of the paper (just 3 / 17 lines are devoted to what the paper actually does).

**Author Response:** We have updated the abstract to include the distinction between the types of methods introduced in this project (lines 9-12). It now reads:

*"We propose that there have been three broad strategies for characterising jet streams in the literature, including statistics that isolate a single value from the wind speed profile (jet statistics), methods for quantifying the sinuosity of the upper air (waviness metrics), and algorithms that identify the coordinates of fast flowing wind (jet core algorithms)."*

*Reviewer: Lines 26–33, See general point (2).*

**Author Response:** We have updated the text here to clarify the three broad types and to better align with the categorisation/modules provided in *jsmetrics* also (lines 25-31).

*Reviewer: Lines 53–55, Should cite Lee & Kim (2003) and Manney et al (2014) in addition to Spensberger and Spengler (2020) (indeed, much of the latter's discussion was based on the discussion in those earlier papers).*

**Author Response:** Thank you for pointing in the direction of these papers, the text has been edited to include these citations (lines 54-56).

*Reviewer: Line 81, not all of these (eg, Barton & Ellis use one specific pressure level) describe the "atmospheric column".*
**Author Response:** We have removed "atmospheric column" from the sentence to avoid confusion (line 84-86) Sentence now reads:
*"These metrics are generally not designed to capture individual events or general form in the jet such as troughs or ridges, but instead to capture the general climatological characteristics of a jet stream, such as its position and speed (Koch et al., 2006; Barton and Ellis, 2009; Rikus, 2018)".*

*Reviewer: Lines 91–92, these also do not capture the behavior near the level of maximum wind speed, see, e.g., discussion in Melamed-Turkish et al. (2018) and Manney et al (2021).*
**Author Response:** Thank you for your suggestion and pointing us towards these papers. We have edited the text to include mention of further limitations of the jet statistics (lines 97-98):
*"They also do not capture any behaviour near the level of maximum wind speed, nor the presence of multiple jet streams (Melamed-Turkish et al., 2018; Manney et al., 2021)."*

*Reviewer: Lines 93–95, how is "extracting the latitude of the jet stream as the point of fastest zonal wind" different from setting "a value of jet latitude and jet speed for each longitude"? Are you*

*saying in the first statement that the maximum is found as a function of latitude and longitude? (Note also, per comment on metrics versus algorithms, if you add "and jet altitude" the latter accurately describes the M11 and PO13 methods, and probably others, that are categorized as "jet core algorithms".)*

**Author Response:** We have attempted to remove confusion originally produced by these two sentences. We have edited the text (line 100-101):

*"Most commonly, this involves extracting 'latitude' and 'speed' quantities at the point of fastest zonal wind, either for an entire study region (all metrics expect K20), or by longitude (K20)".*

***Reviewer:*** *Line 125, since this is indeed a "highly contested" topic, it might be good to cite some papers led by other authors, e.g., Manney & Hegglin 2018 have a fairly detailed paragraph in their introduction, citing numerous still-relevant references; Francis (2017, BAMS, DOI:10.1175/BAMS-D-17-0006.1) would also be appropriate.*

**Author Response:** Thank you for the suggestion. Along with comments made by the other reviewer, we have edited this part to include more discussion on jet waviness (lines 131-137):

*"The notion that a 'wavier' jet stream leads to more extreme (winter) weather in response to the warming world is a highly contested topic (Francis, 2017; Manney and Hegglin, 2018; Cohen et al., 2020, 2021), but it is suggested that the slower progression of the jet stream in a 'wavier' regime encourages surface weather systems to take a longer path and broader across the planet's latitudes and as such encourage the transport of colder air to be pushed further equatorward and vice versa. Robust conclusions about changes in jet waviness so far have been difficult to establish due to variation in the region and years studied, as well as the methodology used (e.g. Barnes, 2013; Barnes and Simpson, 2017; Blackport et al., 2019; Blackport and Screen, 2020)".*

***Reviewer:*** *Table 3 and discussion thereof, it would be helpful to distinguish between methods that provide jet altitude as well as jet latitude information (eg, M11, PO13) and those that provide only jet latitude information and that often over a vertically-averaged range (eg, Koch et al, Archer & Caldeira, Schiemann et al, Kuang et al)*

**Author Response:** Thank you for this suggestion. We have edited the text to include a note of the dimensionality of the outputs returned by the jet core algorithms (lines 148-150). We hope this exists as an improvement on our original manuscript, where we may have failed to convey this to the reader. *Updated text:*

*"We note that, the implementations of SO9, M11, PO13, and K14 can provide three-dimensional outputs for each time step including altitude coordinates about the jet cores they extract, and K06 and ACO8 instead return mass-weighted output which provide two-dimensional jet cores for each time step".*

***Reviewer:*** *Lines 139–140, per general comment (3), M11 also fall into category (ii)*
**Author Response:** Yes, we agree and have amended the text (lines 153-154).

***Reviewer:*** *Line 142–146, It could be just the wording, but this reads as if you are saying the methods of Manney et al (including the important refinement in Manney & Hegglin, 2018 of a physically-based method to distinguish subtropical and polar jets) does not "resolve" different reasons and seasons, when in fact they specifically focused (particularly Manney et al, 2014, 2017, 2021 and Manney & Hegglin, 2018) on analysing regional and seasonal variations. Manney & Hegglin (2018; which shows polar / subtropical jet separation in different seasons) and Manney et*

*al (2021; which, while focusing on ENSO relationships, has several figures looking at subtropical / polar jet relationships) would be good references to cite along with Maher et al (2020) for showing seasonal variations in polar and subtropical jet stream locations and separation.*

**Author Response:** Thank you for this insight. We have amended the text to reflect the research done in those paper, as well as edits suggested by RC1 (lines 157-162):

*"Furthermore, they may also underestimate positions of the jet cores in different seasons, in climate regimes different from the present (e.g. SSP5-8.5), and within different phases of the given climate oscillations (e.g. Woollings et al., 2010; Madonna et al., 2017; Manney and Hegglin, 2018; Manney et al., 2021). We expect jets to be faster and the eddy-driven and thermally-driven components to be more latitudinally separated in the winter versus summer, although this relationship also expresses significant regional variation (Manney and Hegglin, 2018; Maher et al., 2020; Manney et al., 2021)".*

*Reviewer: Lines 154–157, PO13 used a simple latitude criterion and demonstrated/stated that it didn't work well. Manney et al (2014) did not distinguish but simply showed and discussed the spectrum of jets (I have no idea what you mean by that being "a more emergent form" and it doesn't separate the jets into "groups") – except for one climatological figure where they used a latitude criterion and discussed the fact that that was only useful for very broad climatological studies (consistent with PO13). If you are going to discuss this topic, you should note that Manney & Hegglin (2018) developed a physically-based identification of the subtropical jet and then defined the polar jets with respect to that – this is what the package originally developed by M11 has been using since then (eg, Manney et al, 2021, who discuss subtropical / polar jet relationships in the context of ENSO variations) (it would also be useful to have this method implemented in jsmetrics, though it does require tropopause altitude information, so would require reading additional fields, which at its current state, jsmetrics seems reluctant to do.)*

**Author Response:** We have edited the text of the manuscript to include mention of issues found with the jet seperation in PO13 (lines 170-172). We have made a clearer note of the physically based subtropical jet separation introduced Manney & Hegglin (2018) (lines 172-174). We have also indicated (lines 174-175). These sentences now read:

*"PO13 develop a method to distinguish between merged and separate states of the polar and subtropical jets after the initial detection of jet cores, but were only able to separate the Northern Hemisphere subtropical jet in Jan-Feb. The M11 method was extended by Manney and Hegglin (2018) which introduces a physical-based identification of the subtropical jet (based on the thermal tropopause altitude), to more robustly separate it from the polar jet. Manney et al. (2014) found that separating the M11 cores by a latitude criterion to be effective only at a climatological scale".*

You are correct in that *jsmetrics* is relevantly reluctant to introduce additional field to the methods at this stage, but we would like to work with you to introduce the physically based identification of the jets as an add-on to the M11 implementation of the package.

*Reviewer: It might also be worth noting that some methods by their nature selectively identify the subtropical jet (e.g., though it isn't one of the ones currently implemented in jsmetrics, Maher et al., 2020), and that some (e.g., several papers by Winters and others (e.g., Winters et al, 2020, DOI: 10.1175/MWR-D-19-0353.1, and references therein) based on the method described by Christenson et al (2017, https://doi.org/10.1175/JCLI-D-16-0565.1) distinguish them by postulating that they occupy different altitude ranges (in a sense, a refinement of the common procedure of using low level wind maxima to identify the polar jet, but one that can be used while still capturing the level of the core of that jet, and in a similar spirit to Koch et al.).*

**Author Response:** Thank you for the suggestion, we have added a note about the review from Maher et al. 2020 (lines 175-177):

*"Finally, we also note that are some methods that have been developed exclusively for the subtropical jet (see Maher et al. (2020) for a review of such methods), and envisage these could be incorporated in a future release of jsmetrics".*

*Reviewer: Lines 190–191, I was unable to find this, though I believe I looked through the website of Keel (2023) and the associated GitHub project (except for the broken link mentioned in the general comments) thoroughly. This might have been helpful in assessing more definitively whether the implementation contained the fundamental aspects of the method.*

**Author Response:** Thank you for alerting us to this. We have edited the text to highlight that any relevant information is now included in the docstrings, with a full description of the what a given method does including equations taken from the original method (and the section it is first described in). We hope that this has alleviated your concern.

*Reviewer: Lines 204–206, the "details_for_all_metrics.py" file contains nothing in the "description" field for the vast majority of the methods (and the one I saw that was not empty did not give a description of the method). In the entry for M11, it makes it sound like only levels between 400 and 100hPa are needed to implement it, whereas (see general comment (3)) a deeper vertical range is required.*

**Author Response:** We have now filled out the "description" field for this file. All methods are covered in that file, and you can find this file here: https://github.com/Thomasjkeel/jsmetrics/blob/main/jsmetrics/details_for_all_metrics.py

*Reviewer: Lines 212–213, being able to see the pseudocode would have been very helpful in not only assessing to what degree the implementation matched the original (as mentioned above), but also in understanding what the implementation actually did in general.*

**Author Response:** We understand your concern here, although as the pseudocode was written on paper, this would be a significant investment of time to provide online. Instead, we hope that the overhaul to the in-code and online documentation we have provided will be sufficient. We have been careful to introduce and describe each method (in isolation), that makes it obvious where we have extracted the method from, and how we have implemented in jsmetrics. Furthermore, the examples we now provide for each of the metric in their docstring, should help the user be able to work with the code.

*Reviewer: Lines 225–227, as already noted, I found the manner in which the "implementation-level detail" was hidden made the package quite the opposite of user-friendly. Also, the implementation is critical for the user to understand where it means that the method may differ conceptually from that in the original paper. The user needs to know what each method, as implemented in jsmetrics, actually does, so if the implementation is hidden there has to be very thorough and complete documentation of that.*

**Author Response:** We have removed this mention of hiding implementation-level detail to avoid confusion. Consequently, we would like to emphasise that the implementation of methods to *jsmetrics* should be more evident in the in the code and in the inline and online documentation now. We have been careful to write the descriptions and provide helpful inline comments for each metric

now. This includes adding steps (e.g. *Step 1 Calculate monthly means, Step 2 … etc.*) to each method with similarly named functions just underneath, such that it reads a close to English as possible (e.g. for *koch_et_al_2006 # Step 1. Get all pressure levels (hPa) as list* is the comment above code that runs the *get_all_hPa_list(data)* subroutine). This should help any prospective user (of any degree of Python competency) to have an idea of how a given metric works and how the implementation has been segmented in the code.

For an example, see the jet core algorithms file: https://github.com/Thomasjkeel/jsmetrics/blob/main/jsmetrics/metrics/jet_core_algorithms.py

*Reviewer: Line 237–239, by "7 metrics available in jsmetrics" it appears you mean the ones you identify as "statistics". Since most (if not all) of the "jet core algorithms" provide (at least in their original implementations) wind speed (a more accurate measure of "...speed of the jet stream…" than maxima in the u-component) it would help explain why you group these together if you note that these are all applied to the lower troposphere.*

**Author Response:** Yes, we mean jet statistics, we have made the change to the text (line 264). Thank you.

*Reviewer: Lines 265–266, also, these lower tropospheric altitudes are far below the altitude of the core (maximum winds) of these jet streams, and regional averages would be lower than at an individual longitude grid point.*

**Author Response:** In response to your later comment, we have removed the jet speed figure (was Figure 3), and also these lines from the text.

*Reviewer: Lines 268–269, you should define what you mean by "structural uncertainties".*

**Author Response:** We have renamed this type of uncertainty: "metric uncertainty", and have changed the text here define this term, and in places previously mentioning structural uncertainties (first mentioned on line 274, and further outlined on line 335).

*Reviewer: Line 278, these five pressure levels represent vertical spacing / resolution that is inadequate to resolve upper tropospheric jet cores – see, e.g., comparison in Manney et al, 2017, of calculations using the M11 method on reanalysis model levels with the same reanalysis on standard pressure levels (which are poorer resolution than those in the models, but better than those used here).*

**Author Response:** While we agree with your comment here, we believe that for this case study we are primarily trying to demonstrate broad and non-metric specific differences between the jet core algorithms. Using a smaller amounts of data at a coarser resolution also speeds up the computation of the methods for anyone wishing to reproduce the figures in this section. We have made changes though, and added a comment about vertical resolution and the comparison in Manney et al. 2017 to lines 299-301:

*"We note that the vertical resolution and grid spacing of the data used in this case study, may not be adequate for the methods to effectively capture jet cores (see a discussion of vertical resolution and grid spacing in Manney et al. (2017))".*

*Reviewer: Figure 4, why don't you show the jet core wind speeds for all those methods that calculate it (both M11 and PO13 methods do)? Also it would be helpful to look at the jet core altitude information from those methods that calculate it (at least the two just mentioned) and see how close that actually is to 250 hPa – this would tell you how different you'd expect a jet core identified by those methods at whatever the actual altitude of the jet core is to agree with the winds at 250hPa.*

**Author Response:** Thanks for your suggestion. Although we have not included jet core altitude information (we purely want to emphasise that the jet core algorithms in jsmetrics return the coordinates of jet features), in this figure (in the interest of keeping the comparison simple), instead we have selected the outputs of PO13 and M11 at the 250 hPa level for a closer comparison with wind speeds (see Figure 4 below). We also add a note about the altitude information returned by the outputs of M11 and PO13.

lines 301-304:

*"Finally, we have selected jet cores at 250 hPa from M11 and PO13 for comparison with wind speed and K14, but we acknowledge that these two algorithms also return jet core outputs at different altitudes".*

Figure 4:

[Figure]

*Reviewer: Line 282, I'm not sure how really unique they are given that M11 is an extension of Koch et al and PO13 is nearly identical to M11.*

**Author Response:** We appreciate that the results in Figure 4 of original manuscript were not accurate, we have updated this figure which should emphasis the similarity of these algorithm now.

*Reviewer: Lines 287– 292, this does not seem like a particularly clear or even accurate description. M11 simply identify all (well, up to five in each hemisphere) jet cores at each longitude. They do not assess splitting (or merging), which I think(?) is what you are trying to get at in line 290 – note, however, that a faithful (but independent) implementation of the method by Homeyer & Bowman, 2013 does add a post-hoc algorithm such as you mention to "string the jets together" in longitude and identify splitting and merging. (A complete implementation of the current M11-based method would also provide subtropical / polar jet information.)*

**Author Response:** You are correct, we have changed the interpretation in this section to better reflect more true workings of the M11 method (lines 312-316):

*"M11 use an additional algorithm after initial discovery of local maxima to divide jet cores occurring within the same local maxima region based on whether: (1) the two or more cores are more than 15 degrees of latitude apart and (2) whether the wind speed drops more than 15 ms−1 between those cores, otherwise these jet cores in the same region will be considered part of the same core. As such, the jet core output from M11 at 250 hPa may vary slightly from other similar methods (e.g. S09 & PO13), as the jet cores may be associated with different altitudes".*

*Reviewer: Lines 293–296. "Surrounding 8 grid cells" in what space (latitude / longitude / altitude?). If restricted to latitude / altitude, M11 also do this, and most certainly "make the assumption that the centres of the jet streams are important features in their own right" – that is, in fact, the central motivation for their method. (This would also hold for PO13.)*

**Author Response:** We have edited the text to include a mention that this is surrounding in the latitude/altitude plane. We have also added this for PO13 and M11 too, as you mention (lines 309-312 and 312-316: see above):

*"There are only slight visual differences between the jet cores in P13 and S09, because both algorithms make use of a wind-speed threshold of 30 ms−1 to extract local maxima in the altitude/latitude plane, but S09 isolate jet cores only where the u−component wind is also shown to be above 0 ms−1".*

*Reviewer: Line 298, I don't think it is intuitive that the differences would be amplified when aggregating into coarser time resolutions. In fact my first thought was that differences might be in some sense averaged or filtered out. Why do you expect this?*

**Author Response:** Yes, this is an oversight in our wording, thank you for pointing this out. We have adjusted the text to reflect the opposite sentiment (lines 322-324):

*"With this case study, we demonstrate the slight differences in the estimations of the jet stream from various jet core algorithms, and suggest that the difference at the 6-hourly scale will likely be amplified when aggregating into coarser time resolutions".*

*Reviewer: Lines 306–307, parametric uncertainty testing is a fundamental part of development of any algorithm, so some of this is typically mentioned in the original papers.*
*&*

*Lines 308–309, Manney et al (2017) focused on this; in addition, Manney & Hegglin (2018), and Manney et al (2021), and PO13 all used multiple input datasets (mostly reanalyses in these cases).*

**Author Response:** We agree, although not all algorithms provide details about this or allow others to change the parameters. We are including parametric uncertainty in this list of potential directions for sensitivity analysis just to highlight the broader possibilities with *jsmetrics*.

*Reviewer: Lines 310–314, See comment on Figs. 2 & 3 – having both is redundant, and Figs. 5 & 6 provide far more useful information.*

**Author Response:** Thank you for the suggestion, we chosen to remove Figure 3, and combine information originally given here with Figure 6 instead. We have edited the text and figure caption to reflect this (lines 337-338).

*Reviewer: Lines 330–332, I'd expect more transient features to be "washed out" for longer averaging periods and more persistent features to remain better defined – what you are saying sounds contrary to this.*

**Author Response:** We have edited our analysis here (lines 370-374):

*"This feature is robust up to about 4 days, but a trough structure becomes less clear in the jet occurrences and jet centers beyond that. We expect large-scale and persistent features of the jet stream (in this case a stationary/standing wave over North America) to be more defined/stable at broader time scales if the weather system remains and the features of the jet to stay in place over a region".*

*Reviewer: Lines 323, 326, and 340, by "temporal aggregations", do you mean you are averaging over that number of days before doing the jet identification / characterisation? (It looks like that to me since if you plot the points for all of those days / times of day I'd expect to see more points on the plot for each successively larger time interval.) If by "temporal aggregations" you mean "time mean", then just say "time mean". (Also, assuming that's the case, in the Fig. 7 caption replace "time scales" with something like "averaging periods".)*

**Author Response:** We have updated the text, thank you for the suggestion to rephrased "temporal aggregation" to "time averaging" (lines 364, 374, 378 and 381).

*Reviewer: Lines 343–344 and 346–348, most (all?) of the "jet core algorithms" are also "purpose built" for extracting jet latitudes as one of their primary products (and jet wind speed as another) – e.g., these were a focus of results in PO13 and in the Manney-led papers from 2014 on. Why do you not just use the jet latitude and wind speed that those algorithms provide? It doesn't make sense to me to compare algorithms by comparing something obtained via a method they didn't use.*

**Author Response:** We have opted to include this figure, although methodologically obsolete, simply to demonstrate the possibilities of *jsmetrics*. To alleviate your concern that M11 and PO13 provide their own methods we have added lines 390-391:

*"We note that the papers that spawned some jet core algorithms e.g. M11 & PO13 provide their own jet latitude extraction method, but these have not yet been implemented in jsmetrics".*

*Reviewer: I have little to say in terms of specifics on Sections 5 and 6 – since they are very general discussion I think the related issues (such as the lack of success of the package as it currently stands, particularly with the current state of the documentation, in being "simple to use").*

**Author Response:** We hope to have alleviated your concerns here with the aforementioned improvements to the documentation and clarifications made in the manuscript.
* * *
*Minor / Technical points (typos, grammar, wording, etc; wasn't particularly reading for this, so just what I happened to notice):*

*Reviewer: Throughout, "ERA-5" needs to be replaced with "ERA5" as that acronym does not have a hyphen in it.*

**Author Repsonse:** Thank you for point this out, the manuscript has now been updated with this correction.

*Reviewer: Line 1, I see no need to say "this planet's" – so far as I know the jet streams are complex on all the planets (eg, Earth, Venus, Jupiter, Saturn, possibly others) we've studied them one (and given enough data, similar methods might be used for other planets)!*

**Author Response:** Done (line 1).

*Reviewer: Line 11, I think you could and should leave out "or reduces" ("highlights" by itself covers that by implication). Also "exist" should be "exists" and "characterise" should be "characterises" (refers to "Each", which is considered singular).*

**Author Response:** We have removed this sentence.

*Reviewer: Line 23, "most popular" is a subjective assessment, you could just say something like "Among the most commonly used approaches is to develop…"*

**Author Response:** Thank you, we have edited the text (line 23).

*Reviewer: Lines 46–47, should be something like "...where there are extreme temperature and vertical pressure gradients…" (and I don't think "vertical" is needed here).*

**Author Response:** We agree, the text has been updated (lines 44-45).

*Reviewer: Line 68, delete comma after "detect".*

**Author Response:** Done (line 71).

*Reviewer: Line 71, I think this should be plural possessive, thus "jet streams'"*

**Author Response:** Done (line 75).

*Reviewer: Line 89, delete comma after "and"*

**Author Response:** Done (line 94).

*Reviewer: Lines 109–110, It would be clearer (because this is the point that Manney & Hegglin specifically discussed) to move the Manney & Hegglin reference to immediately follow "synoptic-scale events".*

**Author Response:** Done (line 115).

*Reviewer: Line 115, "propagations" should be "propagation"*

**Author Response:** Done (line 127).

*Reviewer: Line 116, wording not clear, do you mean "...are 'jet streams' such as those driven by eddy- or thermal processes…" Also doesn't "driven by eddy or thermal processes" pretty much cover all 'jet streams', thus not really needed? (Unless you mean to say something like "...are 'jet streams' nor do they diagnose the eddy or thermal processes driving them…")*

**Author Response:** We have updated the text (lines 120-122).

*"They broadly describe propagation of Rossby waves in the structure of the upper-level mean flow, and they do not necessarily isolate which parts of the mean flow are 'jet streams', nor do they diagnose the eddy or thermal processes driving them (Martin, 2021)".*

*Reviewer: Lines 179–181, I'm not sure this is even worth saying, since it would be unacceptable for any package aimed at community use not to have this flexibility.*

**Author Response:** I have left in for now, because I think this is worth hammering home (especially as we want to follow clean software design principles) (lines 199-201). Also, based on comments by the other reviewer, we have also added a limitation to jsmetrics' design philosophy (lines 201-205).

*Reviewer: Line 237, should note that it is lower tropospheric u-component data that are used here.*

**Author Response:** Done (line 261)

*Reviewer: Line 238, need serial (Oxford) comma after "North Pacific"*

**Author Response:** Done (line 263).

*Reviewer: Line 255, "these metric" should be "these metrics" and "which" should be "that".*

**Author Response:** We have removed this sentence based on comments made by the other reviewer.

*Reviewer: Line 283, "which" should be "that"*

**Author Response:** Done (line 307).

*Reviewer: Lines 285, 288, 289, probably other places, in this usage "jet-cores" should be "jet cores" (two words, not hyphenated).*

**Author Response:** We have changed all occurrence to 'jet cores'.

*Reviewer:* Line 301, delete comma after "calculation".

**Author Response:** Done (line 331).

*Reviewer:* Line 322, at least add "e.g.," before these two references (and / or "and references therein" after) since there are many, many papers that discuss this.

**Author Response:** Done (line 363).

*Reviewer:* Line 353, do you mean that all the jet core algorithms use a wind speed threshold (in which case there should be a comma between "algorithms" and "which") or that it includes none of the ones that do use a windspeed threshold (in which case "which" should be "that")?

**Author Response:** Done (line 396).

*Reviewer:* Line 381, should be "...similar metrics, e.g., for calculating…" (unless you are saying the following is an exhaustive list, in which case "i.e." is appropriate but the commas are still needed).

**Author Response:** Done (line 424).

*Reviewer:* Line 382, 394, 396 "which" should be "that".

**Author Response:** Done.

---

## Referee Report (RR1)

Re-evaluation of "jsmetrics v0.2.0: a Python package for metrics and algorithms used to identify or characterise atmospheric jet-streams", by Keel, et al.
Reviewed by Gloria Manney

**Recommendation:** The revised manuscript and code / documentation are all greatly improved and should be suitable for publication in GMD pending some further clarifications in the text.

**General Comments:** The online documentation of the jsmetrics software has been vastly improved; it is now straightforward to run any of the metrics, and there is a helpful set of examples for doing so. The manuscript is also much improved, and I believe this work is now sufficiently mature for publication in GMD. I do have quite a number of comments on issues or language that I feel still needs some clarification, but while these may be somewhat extensive, they are all IMO in the nature of "minor" revisions.  In a few cases that will be noted below I have already discussed appropriate modifications with the lead author; these concerns are included here to keep the online record of the review / revision process complete.

In general, I still think that the distinction drawn between "jet statistics" and "jet core algorithms" is too strong, since there can be (depending on the algorithm and the application of it) a large overlap in the information they provide. Nevertheless, as long this overlap is acknowledged and the capabilities of and the primary outputs of each group of algorithms are clearly described, this choice does not materially impact use of the package or interpretation of the results.  Several of my comments below focus on further clarification of this issue.

In the same vein, I still question the choice not to include jet core windspeed in the outputs for jet core algorithms (such as M11) that identify the jet core locations using windspeed (since that information is already available as it is essential to using the algorithm).  In this case, while this would not have been my choice, the authors now include examples showing how to get this from the "mask" of jet cores – so, as above, this does seriously impact use of the package.

**A Couple of Comments Re the Author Responses:**

Response to my general point (2), regarding the same issue mentioned just above:  Part of my concern is that the original discussion made it sound like the "jet statistics" algorithms provided more information than the jet core algorithms, when in fact the opposite is typically the case.  I think the revised discussion does improve this, but there are a couple of places where this could be clarified further, noted below; in particular, see my comment on Fig. 9 and the discussion thereof.

Regarding the authors' response about M11 implementation and the jupyter notebook detailing that (which was indeed very helpful; it would be excellent to be able to see something like this for other algorithms, though I appreciate that that may be too much work in cases where you haven't done something similar to this already), to paraphrase my exchange with the lead author (denoted as TK) on this implementation:

In[23]: I noted that JETPAC {the formal acronym for the package described by M11} has an undocumented feature in that, for each longitude slice without windspeeds > core_threshold (currently 40m/s), if there are regions with windspeed > edge_threshold, it catalogs those regions and the location / value of the max windspeed within them; if there are no windspeeds > edge_threshold, it catalogs a single maximum windspeed location (location, windspeed, other characteristics).  It appears that if you wanted to allow that feature (at least the core > max > edge option) in jsmetrics, you have that information here before you do the down selecting to remove the regions with no cores.

*TK responded that he would look into this.*

In[26]:  I questioned why even do this {downsample contour found for edge of jet region to get only the points above, below, equatorward, poleward of the jet core} when what you've got (the full region mask) provides more information than the original (above/below, poleward/equatorward)?! … I believe (if I've followed everything correctly) that you do have the option to retain and save this full mask, is that right?

*TK responded that the full mask is, indeed, included in the outputs.*

In[36]: It was not obvious to me in trying to go through this that the largest of the local maxima is always the one selected when one is / some {that are in the same "jet region"} are eliminated?

*TK answered that the function currently did not do this, but that he would correct it to do so.*

Regarding the "alternative with diagonals checked for jet cores", I had already realized that not checking the diagonals was something that may pose a big inefficiency in the original JETPAC implementation – though it does work as intended in the end, I guessed that there is a lot of unnecessary checking of multiple local maxima because of that (which could impact the speed).

*TK's response indicates that the option to check the diagonals is / will be included in jsmetrics.*
* * *
**Specific Comments On the Manuscript** (in order of appearance in the manuscript, not importance; line numbers are from tracked changes version):

Lines 12–13 (Abstract): Suggest rewording, something like "We classify the methods for characterising jet streams in the literature into three broad strategies: statistics...", since it really is a choice you made in how to group them in the package as opposed to a "proposal" for how they should be thought of in general.

Lines 28–29:  Suggest "We divide these common approaches into three broad types:"

Lines 30–31:  This sounds like each of these algorithms returns one and only one of latitude, speed, or width. I think some of them return more than one, right?  So this wording should be modified.

Lines 32–33:  The wording makes it sound rather like sinuosity is the only measure of waviness, when in fact there are many – again, greater care with the wording would be helpful.

Lines 34–35:  May want to say something about identifying the maximum windspeed and/or the region around that maximum, since that is the definition of "jet core".

Lines 40–41:  These references don't really seem like the best choices here, since, while they are mainly review papers, they generally cite few (most of them none) of the results from papers using metrics implemented in jsmetrics – that is, they don't demonstrate that the metrics you are implementing provide conflicting or confusing information.

Lines 61–64: This is a bit of a moving target, but this paper:
Spensberger et al 2023, DOI: 10.1175/JCLI-D-23-0080.1,
published (early online release) in *J. Clim.* since the original version of the jsmetrics manuscript is a very good choice that could be added for discussion of thermally / eddy driven jets; it also demonstrates a new way of identifying them using potential temperature (something that might easily be implemented in jsmetrics in the future).  There are a few other places below where I also suggest citing it.

Lines 63–64:  Suggest something like "...tropospheric jet streams but diagnostics included may identify either or both of the "primary" types...." to make it clear that one or both of eddy or thermally driven jets may be identified – i.e., that this statement simply says you are not identifying stratospheric jets.

Line 71: I don't think "synonymous" is what you mean here (that would say that these jets **are** cold waves, heat waves, etc).  Perhaps something like "..directly involved in {development | evolution} of…"

Line 75–78:  This seems out of place here.  Suggest joining this with the paragraph at the end of the previous section, then starting this section with something like "Despite their importance to climate studies, features of…"

 Line 83:  Should be "specific questions" and later in this line "and / or" since it is usually not just one characteristic if indeed they are developed for such a specific purpose.  Which M11 (aka JETPAC) definitely was not, and I expect that is the case for some of the others as well (especially jet core algorithms that provide a wealth of information).  JETPAC was developed (as previewed in M11) to be useful for many purposes; in addition to the papers (those cited here, along with another in which JETPAC diagnostics are correlated with Asian summer monsoon anticyclone characteristics, Manney et al, J Clim, 2021(b), DOI: 10.1175/JCLI-D-20-0729.1; and another in relation to the tropopause inversion layer, Peevey et al., 2014, JGR,

doi:10.1002/2014JD021808) that use it to study climatology, variability, and trends in the jet streams and related phenomena, it is also being used and has been used in studies of transport and STE, and for analysing UTLS composition variability / trends (e.g., Olsen et al, 2019, JGR, https://doi.org/10.1029/2019JD030435; Millán et al, 2023, AMT, https://doi.org/10.5194/amt-16-2957-2023).  My point is not that you should cite all these papers, but that some tropospheric jet diagnostics have been developed for very broad purposes.

Line 96:  Other methods have "evaluate latitudinal shifts, slowing or speeding up of the jet" as one of their primary purposes, e.g., M11 and PO13.

Line 124:  Spensberger & Spengler's method is not currently implemented, right?  And it doesn't IMO fit the category of "jet statistics" (see usage in Spensberger et al., 2023, mentioned above).

Line 136 and Table 2:  A brief description of what each of these metrics actually does would be helpful (i.e., what is calculated from what to get the metric).

Lines 169–170:  Seems odd to use Manney et al (2014) in these two reference lists when the discussion is about the algorithms described in Manney et al (2011) and the other references given are all the "methods" papers for that technique.

Line 187: Suggest "a method based on latitude to distinguish" – that will help clarify why is doesn't work very well most of the time.

Line 195: Would be good to mention Spensberger et al., 2023 (see citation info above), as well as the method introduced by Christenson et al (2017, J Clim, DOI: 10.1175/JCLI-D-16-0565.1), which both (in different implementations) use jet core potential temperature to distinguish eddy from thermally driven jets.

Lines 214–215:  This is as good a place as any to mention that allowing different vertical coordinates (e.g., altitude, potential temperature, and, especially, model levels) should be a high priority for future jsmetrics development.  E.g., JETPAC is typically run on reanalysis model levels because of the inadequacy of the "standard" pressure levels to capture the vertical structure (e.g., Manney et al, 2017, ACP); but is also sometimes run on pressure, altitude, or potential temperature levels when being used with other datasets to which those coordinates are native.

Line 291: "capable of" is not the right wording here, since most of the jet core algorithms are "capable of" this and you are excluding all of them as well.  You could just say something about showing metrics that look at lower tropospheric u-wind and leave it at that – that is sufficient to explain which methods you show here.

Figure 2: As I noted in my original review, you need to define in the caption what the width (top to bottom), length (side to side) of the shaded parts represent, what the thick lines near the

centre represent, and what the length of the thin lines means. Not everyone is familiar with a "violin plot" and the reader shouldn't have to stop and go look it up!

Line 297: What feature in Figure 2 shows the "Interquartile Range"? Also put this in the Figure 2 caption.

Line 323: Add Spensberger et al (2023, citation above) to this reference list, it has very good discussion of this.

Lines 335–336: Several papers have shown (unlike the European and Asian CAOs during that winter) that the stratosphere / SSW didn't have a very big impact on this event, see, e.g.,
Davis et al, 2022, https://doi.org/10.1038/s41467-022-28836-1;
Zhang et al 2022, https://doi.org/10.1029/2021GL096840;
Bolinger et al 2022, https://doi.org/10.1016/j.wace.2022.100461;
as well as others.

Lines 351–352: "only selecting cells of local 'maxima' " is not really "stricter conditions" if they are using the same wind speed threshold for a "core" to exist – it is just providing information only on the core itself rather than on both that and the surrounding "jet region".

Line 365–369: Identifying "jet centers" in latitude and longitude makes "jet centers" a very different beast than "jet cores", which are defined as a maximum in a horizontal coordinate / vertical coordinate plane (the core implementation in M11, for example, doesn't care what those horizontal and vertical coordinates are, just that it is given one of each). As such, K14 really does not fit the "jet core algorithm" category. Nor the "jet statistics" category. Not suggesting that you change it, but that you clarify better the fundamental difference of this method.

Line 399–400: This is consistent with very different jet behavior in different broad latitude regions, and the fact that both the North Atlantic and North Pacific have complex / highy variable jet patterns, whereas the region over Europe / Asia / W Pacific has a strong persistent subtropical jet (e.g., Koch et al., 2006; Manney et al, 2014; Spensberger & Spengler, 2020).

Lines 409–410: This result may or may not be realistic, because of the very coarse vertical resolution of the date used in the example and the different ways each algorithm interacts with that resolution – there is a huge amount of real atmospheric variability between regions, so "more consistent" isn't necessarily expected or realistic.

Line 431: One person's "unimportant feature" could possibly be another person's primary research question! Define "unimportant feature" (and / or choose a more precise wording).

Lines 434–449: As I said in my original review, M11 and PO13 (as well as some of the other "jet core" algorithms) are "purpose-built" to extract jet latitude – jet core latitude and altitude (or other "height" coordinate), along with windspeed, are the first and foremost outputs of these methods, and the ones that have been used most in following work with these methods.

Further, it is disingenuous to say you have not implemented finding the latitude in these methods, since you have a point flagged as the jet core (and have in fact used this to plot those latitudes in Figure 3!), thus all you have to do is to find the index into the latitude coordinate from that core "mask" and extract the latitude – perhaps one or two lines of code (not at all different in principle from extracting the core wind speed, which you have done in (one of the) examples in the online documentation). Further, the method you use to get the latitudes for these (far more complex than what I just suggested) will inherently reduce any appearance of bi-modality since it doesn't allow multiple jets at a given longitude and involves more averaging than the simpler procedure I suggested – thus I don't think you can say anything about bi-modality given the method you have used to get the latitude from the jet core algorithms (as long as they actually identify a core location, I don't think using a wind speed threshold should have much to do with it; though looking at upper troposphere vs lower troposphere could definitely be a factor). Also in these lines: In line 447, should be more specific and say that by "altitude of the methodology" you mean the jet core algorithms are looking at the upper troposphere and the jet statistics algorithms are looking at the lower troposphere – meaning a totally different definition of jets and looking at wind speeds in a totally different region, which clearly affects all of their characteristics.

Line 465: You might want to note here that Manney et al (2017, 2021a, b), Manney & Hegglin (2018), and PO13 all included extensive comparisons of different input reanalysis (which I assume is what you mean by "observational" since we don't have nice 3d gridded fields of actual observed winds!) datasets.
* * *
**Minor / Technical points** (typos, grammar, wording, etc):

Line 39: Suggest instead of "a confusing" either "an unclear message" or "apparently conflicting messages"

Line 45: "from" should be "using" or "upon"

Line 59: Either "positions…are" or "position…is"

Line 79: Suggest "identify and characterise" insead of "detect and then characterise"

Line 81: Add "e.g.," before references, since there are lots of others that discuss aspects of this.

Lines 84–85: Suggest "This initial set of metrics was included based on, first, ... and, second, the frequency of their usage in the literature."

Line 86: "of" should be "in".

Line 90: "which" should be "that", and, to be consistent with the "single value" in line 88, "and" would need to be "or" (see comment above re the description of "jet statistics").

Line 107: Delete "any"

Line 110: Again, " 'latitude' and 'speed' " is not a "single" metric!

Line 113: "While each jet statistic"

Line 193: Replace "which introduces a physical-based" with "by introducing a physically-based".

Line 210: Suggest "...due in part to…" (since this is by no means the only advantage of using xarray!!)

Line 224: Replace "nor" with "and / or" and delete comma.

Line 225: This is a less-than-obvious case, but needs to be either "methods' docstrings" or "method's docstring".

Line 228:  Add comma after "i.e.,"

Line 235: Delete "of" at end of line.

Line 257: "figure" should be "Figure".

Line 259: "rework and refactor" seems a bit redundant (since refactoring is a kind of reworking).

Lines 263–264: Now you are using "refactor" synonymously with "debug" (or "troubleshoot" or whatever term you prefer to use), whereas "refactor" is defined as "to improve internal code by making many small changes without altering the code's external behavior", which clearly implies that the code already produces the desired result.  Also, the sentence structure has errors here, disregarding any content changes, it should be: "After which we either refactor the method further if it fails the validation, or write unit tests, finish the documentation, and integrate the metric into the jsmetrics package if it succeeds."

Line 279: Should be "...from the ERA5…"

Line 286: "is" should be "are" ("data" is plural); also, "details" (which, grammatically, should be "detail" since "data" is plural) isn't the right word here – perhaps use "comprise" or "consist of".

Line 326: Change "We hope to express that" to something like "The above example demonstrates that…" or "We hope the above example demonstrates that…"  (I personally would leave out "We hope" since if you are publishing it you should express confidence in your results.)

Line 329: There should not be a comma.

Lines 333–334, and succeeding use: You use "*North American Cold Wave*" and other times "*Texas Cold Wave*". From what I've seen this event has been most frequently called the "Great Plains Cold Air Outbreak". Whichever term you choose, pick one and only one. (I also see no reason why it needs to be italicised.)

Line 334: Any of "...between 6 and 21 February…", "...from 6 through 21 February…", or "...from the 6th through the 21st of February…" would be correct (I favor the first as being most concise; whichever you use, try to be consistent in succeeding date range references).

Line 338: Delete colon after "levels".

LIne 341: Suggest "...some of the methods…"

Line 343: "...of the figures…"

Line 353: "P13" should be "PO13".

Line 360: Reword / correct: "...those cores; otherwise these jet cores in the same region will be considered part of the same core, at the location of the largest of the local wind speed maxima."

Line 364: "...cores in each may…"

Figure 4. Say in the caption what exactly the "Standard North Pacific Region" is.

Lines 373, 375, 376: See comment re lines 333–334 just above.

Line 397: "figure" should be "figures"

Line 403: Should be "...strongest and most variable…"

Line 418: Change "which all centre" to "all centred on"

Line 425: Delete "to" and delete comma after "that".

Line 436: "which" should be "that".

Line 462: Need a comma after "metric".

Line 482: Should be "...inputs, i.e., with…"

Lines 490–494: The sentence structure here has problems, and the sentence is too long and complex to follow clearly. Suggest breaking it up into two or more sentences and restructuring.

Line 506: "which" should be "that".

Line 511: "data is" should be "data are".

(Note that I also found a number of typos and small errors similar to these in the online documentation, so would suggest more careful proofreading of that.)

---

## Author Response (AR2)

*jsmetrics v0.2.0: a Python package for metrics and algorithms used to identify or characterise atmospheric jet-streams.*

**Response to Editor**

Dear Editor,

We are pleased to see that Reviewer #2 felt that our revised manuscript was much improved and that the online documentation was now sufficient. They have raised a series of issues/comments, and we respond to each of them below.

We feel we have dealt with all of the additional minor changes suggested by Reviewer #2 in their latest review. There are some suggestions that are beyond the scope of the software package in its current form, and having established a dialogue with Dr Gloria Manney, we hope to implement some of these suggestions in subsequent releases.

**Response to Reviewer #2**

We once again thank Dr. Gloria Manney for the help and advice she has provided in curating this manuscript. We are extreme grateful for all the time and effort that she has invested in understanding software package and helping us refine the text.

Please find our response to your comments below.
* * *
**Recommendation:**

*Reviewer: The revised manuscript and code / documentation are all greatly improved and should be suitable for publication in GMD pending some further clarifications in the text.*
**Author Response:** Thank you, and we hope that we have now made these further clarifications in the text.
* * *
**General Comments:**

*Reviewer: The online documentation of the jsmetrics software has been vastly improved; it is now straightforward to run any of the metrics, and there is a helpful set of examples for doing so. The manuscript is also much improved, and I believe this work is now sufficiently mature for publication in GMD. I do have quite a number of comments on issues or language that I feel still needs some clarification, but while these may be somewhat extensive, they are all IMO in the nature of "minor" revisions. In a few cases that will be noted below I have already discussed appropriate modifications with the lead author; these concerns are included here to keep the online record of the review / revision process complete.*

*In general, I still think that the distinction drawn between "jet statistics" and "jet core algorithms" is too strong, since there can be (depending on the algorithm and the application of it) a large overlap in the information they provide. Nevertheless, as long this overlap is acknowledged and the*

*capabilities of and the primary outputs of each group of algorithms are clearly described, this choice does not materially impact use of the package or interpretation of the results. Several of my comments below focus on further clarification of this issue.*

**Author Response:** We have made changes to the manuscript based on your comments about this issue (detailed throughout this response), and have written new descriptions for jet statistics and jet core algorithms, that we hope will clarify the distinction of these two categories. The new descriptions are:

*"Jet statistics — Statistics for isolating individual quantities synonymous with the jet stream from upper-level wind speed within a given time window"* (lines 26-27).

*"Jet core algorithms — Methods that return a mask of coordinates related to the jet location, e.g., identifying the maximum wind speed throughout the horizontal and/or vertical plane within a given time window"* (lines 30-31).

We do accept that methods from these two categories can offer the same kind of information (e.g. you could extract the latitude of jet cores provide by the jet core algorithms), but we hope to have clarified that different approaches are required to process and analyse the outputs of the jet statistics and jet core algorithms from jsmetrics. It felt appropriate for this work to distinguish categories based on types of outputs, and in this process, also describe what the software does and can be used for. In my own thesis (TK), I will treat the distinction between different jet stream metrics quite differently, because I will focusing on the scientific results provided by a particular use case of jsmetrics. We hope that this latest distinction between the categories will be clear for the reader.

*Reviewer: In the same vein, I still question the choice not to include jet core windspeed in the outputs for jet core algorithms (such as M11) that identify the jet core locations using windspeed (since that information is already available as it is essential to using the algorithm). In this case, while this would not have been my choice, the authors now include examples showing how to get this from the "mask" of jet cores – so, as above, this does seriously impact use of the package.*

**Author Response:** We recognise your concern here, but we would like to clarify that this decision was made for the software's efficiency and speed. We wanted to avoid including jet core wind speeds as well as a jet core mask by default, because it was not proven to scale well and we found it to be sub optimal for storage and computing resources. Including only a mask instead of the derived outputs of a mask (like jet core speed, latitude, altitude) means that we can save memory. We hope that offloading some of the procedures to extract other variables from the mask to the online examples will cover the broader uses of the package.
* * *
**A Couple of Comments Re the Author Responses:**

*Reviewer: Response to my general point (2), regarding the same issue mentioned just above: Part of my concern is that the original discussion made it sound like the "jet statistics" algorithms provided more information than the jet core algorithms, when in fact the opposite is typically the case. I think the revised discussion does improve this, but there are a couple of places where this could be clarified further, noted below; in particular, see my comment on Fig. 9 and the discussion thereof.*

**Author Response:** We hope that the new distinction of the two type of methods, mentioned above, is clear. As we mention later in this review, the discussion of the method used to create Figure 9 has also edited for clarification (lines 396-402).

*Reviewer: Regarding the authors' response about M11 implementation and the jupyter notebook detailing that (which was indeed very helpful; it would be excellent to be able to see something like this for other algorithms, though I appreciate that that may be too much work in cases where you haven't done something similar to this already), to paraphrase my exchange with the lead author (denoted as TK) on this implementation:*

*In[23]: I noted that JETPAC {the formal acronym for the package described by M11} has an undocumented feature in that, for each longitude slice without windspeeds > core_threshold (currently 40m/s), if there are regions with windspeed > edge_threshold, it catalogs those regions and the location / value of the max windspeed within them; if there are no windspeeds > edge_threshold, it catalogs a single maximum windspeed location (location, windspeed, other characteristics). It appears that if you wanted to allow that feature (at least the core > max > edge option) in jsmetrics, you have that information here before you do the down selecting to remove the regions with no cores.*

*TK responded that he would look into this.*
**Author Response:** This feature of M11 is now included in the latest version of jsmetrics, and if possible, we would be very keen to continue our dialogue with you to verify M11.

*Reviewer: In[26]: I questioned why even do this {downsample contour found for edge of jet region to get only the points above, below, equatorward, poleward of the jet core} when what you've got (the full region mask) provides more information than the original (above/below, poleward/equatorward)?! … I believe (if I've followed everything correctly) that you do have the option to retain and save this full mask, is that right?*

*TK responded that the full mask is, indeed, included in the outputs.*
**Author Response:** Yes, having this direct dialogue was really helpful.

*Reviewer:  In[36]: It was not obvious to me in trying to go through this that the largest of the local maxima is always the one selected when one is / some {that are in the same "jet region"} are eliminated?*

*TK answered that the function currently did not do this, but that he would correct it to do so.*
**Author Response:** The function has now been corrected, so this is now included in the latest version of jsmetrics.

*Reviewer: Regarding the "alternative with diagonals checked for jet cores", I had already realized that not checking the diagonals was something that may pose a big inefficiency in the original JETPAC implementation – though it does work as intended in the end, I guessed that there is a lot of unnecessary checking of multiple local maxima because of that (which could impact the speed).*

*TK's response indicates that the option to check the diagonals is / will be included in jsmetrics.*
**Author Response:** This diagonal checking is now included in the latest version of the M11 implementation in jsmetrics. This makes a small change to the jet cores shown by M11 in Figure 3.
* * *
**Specific Comments On the Manuscript (in order of appearance in the manuscript, not importance; line numbers are from tracked changes version):**

*Reviewer: Lines 12–13 (Abstract): Suggest rewording, something like "We classify the methods for characterising jet streams in the literature into three broad strategies: statistics...", since it really is a choice you made in how to group them in the package as opposed to a "proposal" for how*

*they should be thought of in general.*
**Author Response:** We have reworded this part of the abstract, and so it now reads:
*"We classify the methods for characterising jet streams in the literature into three broad strategies: statistics that isolate individual values from the wind speed profile (jet statistics), methods for quantifying the sinuosity of the upper air (waviness metrics), and algorithms that identify a mask related to the coordinates of fast flowing wind throughout the horizontal and/or vertical plane (jet core algorithms)"* (lines 9-12).

*Reviewer: Lines 28–29: Suggest "We divide these common approaches into three broad types:"*
**Author response:** Thanks for the suggestion, we have updated the text (line 25).

*Reviewer: Lines 30–31: This sounds like each of these algorithms returns one and only one of latitude, speed, or width. I think some of them return more than one, right? So this wording should be modified.*
**Author response:** We agree, and so we have changed the description of jet statistics to:
*"Statistics for isolating individual quantities synonymous with the jet stream from upper-level wind speed within a given time window (e.g. latitude, speed, width)"* (lines 26-27).
We hope the change of wording from *single* to *individual* is more applicable and better describes this category i.e., you can have a collection of individual quantities (latitude, speed, width, etc.).

*Reviewer: Lines 32–33: The wording makes it sound rather like sinuosity is the only measure of waviness, when in fact there are many – again, greater care with the wording would be helpful.*
**Author response:** Yes, we agree and have removed 'sinuosity' from the sentence here to reduce confusion. The new description for waviness metrics is:
*"Statistics and algorithms for determining the `waviness' of upper-level mean flow within a given time window. These metrics only have meaning at an integrated global scale"* (lines 28-29).

*Reviewer: Lines 34–35: May want to say something about identifying the maximum windspeed and/or the region around that maximum, since that is the definition of "jet core".*
**Author response:** We have refined our description of jet core algorithms in lines 30-31 to better account for all aspects of this category. This change should also account for your later comment about K14 (in reference to *lines 365–369*) to distinguish that method as a jet core algorithm which returns a mask of coordinates.

*Reviewer: Lines 40–41: These references don't really seem like the best choices here, since, while they are mainly review papers, they generally cite few (most of them none) of the results from papers using metrics implemented in jsmetrics – that is, they don't demonstrate that the metrics you are implementing provide conflicting or confusing information.*
**Author response:** We agree and have removed references from this sentence and reworded it to:
*"The differences between these types of approaches could lead to confusion about the trends shown in the planet's jet streams across a range of modelling and observational studies"* (line 32-33).

*Reviewer: Lines 61–64: This is a bit of a moving target, but this paper:*
*Spensberger et al 2023, DOI: 10.1175/JCLI-D-23-0080.1, published (early online release) in J. Clim. since the original version of the jsmetrics manuscript is a very good choice that could be added for discussion of thermally / eddy driven jets; it also demonstrates a new way of identifying them using potential temperature (something that might easily be implemented in jsmetrics in the future). There are a few other places below where I also suggest citing it.*
**Author response:** Thank you for suggesting this reference, we agree it is a very good choice as a citation for this manuscript and have now included it on lines 56, 60, 179-180, and 295-296.

*Reviewer: Lines 63–64: Suggest something like "...tropospheric jet streams but diagnostics included may identify either or both of the "primary" types...." to make it clear that one or both of eddy or thermally driven jets may be identified – i.e., that this statement simply says you are not identifying stratospheric jets.*

**Author response:** Thank you, we have reworded this sentence to.
*"Tropospheric jet streams in observations often exist in "merged states", especially across the mid-latitudes (Stendel et al., 2021), but diagnostics included in this package are not yet able to disaggregate the two "primary" types of jets"* (lines 56-58).

*Reviewer: Line 71: I don't think "synonymous" is what you mean here (that would say that these jets are cold waves, heat waves, etc). Perhaps something like "..directly involved in {development | evolution} of…"*

**Author response:** Thank you. This has been changed to: *"directly involved in the development of cold waves…"* (line 65).

*Reviewer: Line 75–78: This seems out of place here. Suggest joining this with the paragraph at the end of the previous section, then starting this section with something like "Despite their importance to climate studies, features of…"*

**Author response:** Okay, we have moved this content to the end of the introduction (lines 67-69), and the start of section 2 begins with *"Despite their importance…"*, like you suggest.

*Reviewer: Line 83: Should be "specific questions" and later in this line "and / or" since it is usually not just one characteristic if indeed they are developed for such a specific purpose. Which M11 (aka JETPAC) definitely was not, and I expect that is the case for some of the others as well (especially jet core algorithms that provide a wealth of information). JETPAC was developed (as previewed in M11) to be useful for many purposes; in addition to the papers (those cited here, along with another in which JETPAC diagnostics are correlated with Asian summer monsoon anticyclone characteristics, Manney et al, J Clim, 2021(b), DOI: 10.1175/JCLI-D-20-0729.1; and another in relation to the tropopause inversion layer, Peevey et al., 2014, JGR,doi:10.1002/2014JD021808) that use it to study climatology, variability, and trends in the jet streams and related phenomena, it is also being used and has been used in studies of transport and STE, and for analysing UTLS composition variability / trends (e.g., Olsen et al, 2019, JGR, https://doi.org/10.1029/2019JD030435; Millán et al, 2023, AMT, https://doi.org/10.5194/amt-16-2957-2023). My point is not that you should cite all these papers, but that some tropospheric jet diagnostics have been developed for very broad purposes.*

**Author response:** Thank you for pointing this out, we have corrected the text to "specific questions", and "and/or" (line 75).

*Reviewer: Line 96: Other methods have "evaluate latitudinal shifts, slowing or speeding up of the jet" as one of their primary purposes, e.g., M11 and PO13.*

**Author response:** We recognise this and agree, but we felt as though, for the remit of describing this software, it would be appropriate to describe what jet statistics are most useful for in this section and in isolation to the jet core algorithms uses.

*Reviewer: Line 124: Spensberger & Spengler's method is not currently implemented, right? And it doesn't IMO fit the category of "jet statistics" (see usage in Spensberger et al., 2023, mentioned above).*

**Author response:** It is not implemented, but we had made reference to this method in Table 5 under jet statistics originally. We agree with your opinion here and have place this method to jet core

algorithm in Table 5. We have also removed Spensberger & Spengler, 2020 reference from the text here (line 111).

*Reviewer: Line 136 and Table 2: A brief description of what each of these metrics actually does would be helpful (i.e., what is calculated from what to get the metric).*
**Author response:** Thank you for your suggestion here. We have now updated the text to include a brief overview of both of the waviness metrics included in jsmetrics (lines 123-126).

*Reviewer: Lines 169–170: Seems odd to use Manney et al (2014) in these two reference lists when the discussion is about the algorithms described in Manney et al (2011) and the other references given are all the "methods" papers for that technique.*
**Author response:** Yes, we agree and have replaced Manney et al. (2014) with Manney et al. (2011) (lines 157-158).

*Reviewer: Line 187: Suggest "a method based on latitude to distinguish" – that will help clarify why is doesn't work very well most of the time.*
**Author response:** Done (line 176).

*Reviewer: Line 195: Would be good to mention Spensberger et al., 2023 (see citation info above), as well as the method introduced by Christenson et al (2017, J Clim, DOI: 10.1175/JCLI-D-16-0565.1), which both (in different implementations) use jet core potential temperature to distinguish eddy from thermally driven jets.*
**Author response:** We have included a reference to the method in both of these papers and the updated text is:
*"Although not currently implemented in jsmetrics, Christenson et al. (2017) and Spensberger et al. (2023) propose methods which use the potential temperature of jet cores to distinguish eddy from thermally driven jets."* (lines 179-180).

*Reviewer: Lines 214–215: This is as good a place as any to mention that allowing different vertical coordinates (e.g., altitude, potential temperature, and, especially, model levels) should be a high priority for future jsmetrics development. E.g., JETPAC is typically run on reanalysis model levels because of the inadequacy of the "standard" pressure levels to capture the vertical structure (e.g., Manney et al, 2017, ACP); but is also sometimes run on pressure, altitude, or potential temperature levels when being used with other datasets to which those coordinates are native.*
**Author response:** We completely agree with your suggestion here, and have updated the text to include the following sentence:
*"Whilst the current iteration of jsmetrics is only compatible with data with standard pressure levels (plev), for future development of the package, it is a priority to include compatibility with other vertical coordinate systems."* (lines 199-200).

*Reviewer: Line 291: "capable of" is not the right wording here, since most of the jet core algorithms are "capable of" this and you are excluding all of them as well. You could just say something about showing metrics that look at lower tropospheric u-wind and leave it at that – that is sufficient to explain which methods you show here.*
**Author response:** We have removed "capable of" from this sentence (lines 278-279).

*Reviewer: Figure 2: As I noted in my original review, you need to define in the caption what the width (top to bottom), length (side to side) of the shaded parts represent, what the thick lines near thecentre represent, and what the length of the thin lines means. Not everyone is familiar with a "violin plot" and the reader shouldn't have to stop and go look it up!*

**Author response:** We have updated the figure caption and in text description to describe what the violin plot show:

*"The thicker black line in the centre of each violin plot indicates the interquartile range, and the thinner line indicates the 95% confidence interval. The white dot represents the median and the shading which forms the body of each violin is a Kernel Density Estimation, with wider sections representing a higher probability of occurence".* (lines 274-277)

*Reviewer: Line 297: What feature in Figure 2 shows the "Interquartile Range"? Also put this in the Figure 2 caption.*

**Author response:** As above, we have changed the in-text description and figure caption to describe which feature of the violin shows the Interquartile range.

*Reviewer: Line 323: Add Spensberger et al (2023, citation above) to this reference list, it has very good discussion of this.*

**Author response:** Done (line 296-297).

*Reviewer: Lines 335–336: Several papers have shown (unlike the European and Asian CAOs during that winter) that the stratosphere / SSW didn't have a very big impact on this event, see, e.g., Davis et al, 2022, https://doi.org/10.1038/s41467-022-28836-1; Zhang et al 2022, https://doi.org/10.1029/2021GL096840; Bolinger et al 2022, https://doi.org/10.1016/j.wace.2022.100461; as well as others.*

**Author response:** Thanks for pointing this out, we were wrong with our assumption that a SSW had affected this event, as such we have removed the reference to a SSW in this part (lines 304-305).

*Reviewer: Lines 351–352: "only selecting cells of local 'maxima' " is not really "stricter conditions" if they are using the same wind speed threshold for a "core" to exist – it is just providing information only on the core itself rather than on both that and the surrounding "jet region".*

**Author response:** We agree, and the text has been edited to:

*"Notably, S09, M11, P013, and K14 all use a 30 ms threshold (but not in the same way) and both S09 and PO13 select only cells of local 'maxima'; M11 and K14 also extract regions around each core/maxima"* (lines 317-319).

*Reviewer: Line 365–369: Identifying "jet centers" in latitude and longitude makes "jet centers" a very different beast than "jet cores", which are defined as a maximum in a horizontal coordinate / vertical coordinate plane (the core implementation in M11, for example, doesn't care what those horizontal and vertical coordinates are, just that it is given one of each). As such, K14 really does not fit the "jet core algorithm" category. Nor the "jet statistics" category. Not suggesting that you change it, but that you clarify better the fundamental difference of this method.*

**Author response:** We hope that our new definition of jet core algorithms and jet statistics (lines 30-31 & 26-27) clarifies the distinction between jet core algorithms and other types. When considering that all the jet core algorithms do are isolate coordinates throughout a given plane, then this makes sense that these coordinates can be latitude-longitude as well as latitude-vertical.

*Reviewer: Line 399–400: This is consistent with very different jet behaviour in different broad latitude regions, and the fact that both the North Atlantic and North Pacific have complex / highly variable jet patterns, whereas the region over Europe / Asia / W Pacific has a strong persistent subtropical jet (e.g., Koch et al., 2006; Manney et al, 2014; Spensberger & Spengler, 2020).*

**Author response:** Yes, we agree, this is a sensible interpretation. We edited the sentence to reflect that this a real phenomena rather than estimation issue from the metrics:

*"In Figure 5, the mean jet position varies more in the Northern Hemisphere (33.22-49.75◦N) than in the North Atlantic (44.62-49.55◦N), North Pacific (31.57-46.81◦N), or Southern Hemisphere (42.37-50.59◦S)"* (lines 360-362).

*Reviewer: Lines 409–410: This result may or may not be realistic, because of the very coarse vertical resolution of the date used in the example and the different ways each algorithm interacts with that resolution – there is a huge amount of real atmospheric variability between regions, so "more consistent" isn't necessarily expected or realistic.*

**Author response:** We have simplified this sentence to avoid confusion about 'consistency'. New sentence reads:

*"In particular, Figure 5 shows that some metrics show more variation in their estimates across multiple regions than others" (line 370)*

*Reviewer: Line 431: One person's "unimportant feature" could possibly be another person's primary research question! Define "unimportant feature" (and / or choose a more precise wording).*

**Author response:** We have chosen to reword this sentence and use the term 'noise'. The new sentence is: *"The comparison (Figure 8) demonstrates the losses and gains of time averaging: some features are diluted using the mean, while counts show more detail but can also include more noise"* (lines 387-389)

*Reviewer: Lines 434–449: As I said in my original review, M11 and PO13 (as well as some of the other "jet core" algorithms) are "purpose-built" to extract jet latitude – jet core latitude and altitude (or other "height" coordinate), along with windspeed, are the first and foremost outputs of these methods, and the ones that have been used most in following work with these methods. Further, it is disingenuous to say you have not implemented finding the latitude in these methods, since you have a point flagged as the jet core (and have in fact used this to plot those latitudes in Figure 3!), thus all you have to do is to find the index into the latitude coordinate from that core "mask" and extract the latitude – perhaps one or two lines of code (not at all different in principle from extracting the core wind speed, which you have done in (one of the) examples in the online documentation). Further, the method you use to get the latitudes for these (far more complex than what I just suggested) will inherently reduce any appearance of bi-modality since it doesn't allow multiple jets at a given longitude and involves more averaging than the simpler procedure I suggested – thus I don't think you can say anything about bi-modality given the method you have used to get the latitude from the jet core algorithms (as long as they actually identify a core location, I don't think using a wind speed threshold should have much to do with it; though looking at upper troposphere vs lower troposphere could definitely be a factor).*

**Author response:** We have updated the text to describe more specifically how we are creating an estimate for jet latitude for all the jet core algorithms, exclusively for the remit of this case study. We also have added a note that future version of jsmetrics could contain procedures that translate the outputs of jet core algorithms to jet statistics (lines 402-403).

The updated text on lines 396-403 is:
*"To create an estimate for jet latitude from the jet core algorithms, we first compute the estimation of jet cores using a given algorithm and use these locations as a mask to extract wind speed values for each day. Using these values, we then extract the zonally-averaged maximum wind speed and define the associated latitude as the jet latitude value at the native resolution. For consistency's sake, we use a single method to extract the latitude from the multidimensional field returned by the algorithms in this case study. This is the latitude of the maximum wind in the region (despite other options being available to do this for the multidimensional fields, e.g. Manney et al. (2011) would select all the indexes of returned jet cores). Future versions of jsmetrics could contain a variety of procedures that process the outputs of jet core algorithms into jet statistics"*

*Also in these lines: In line 447, should be more specific and say that by "altitude of the methodology" you mean the jet core algorithms are looking at the upper troposphere and the jet statistics algorithms are looking at the lower troposphere – meaning a totally different definition of jets and looking at wind speeds in a totally different region, which clearly affects all of their characteristics.*

**Author response:** Text changed, now reads:

*"This is most likely due to the altitude of the methodology, as the jet core algorithms are looking at the upper troposphere and the jet statistics algorithms are looking at the lower troposphere (Tables 1 & 3)".* (lines 405-407).

*Reviewer: Line 465: You might want to note here that Manney et al (2017, 2021a, b), Manney & Hegglin (2018), and PO13 all included extensive comparisons of different input reanalysis (which I assume is what you mean by "observational" since we don't have nice 3d gridded fields of actual observed winds!) datasets.*

**Author response:** We have changed "observational dataset" to "input reanalysis datasets" and included a note of the papers you mention here (lines 425-426).
* * *
**Minor / Technical points (typos, grammar, wording, etc):**

*Reviewer: Line 39: Suggest instead of "a confusing" either "an unclear message" or "apparently conflicting messages"*

**Author response:** We have reworded this sentence to *"The differences between these types of approaches could lead to confusion about the trends shown in the planet's jet streams across a range of modelling and observational studies."* (lines 32-33)

*Reviewer: Line 45: "from" should be "using" or "upon"*

**Author response:** done (line 38)

*Reviewer: Line 59: Either "positions…are" or "position…is"*

**Author response:** done, we went for 'position...is' (lines 51-52)

*Reviewer: Line 79: Suggest "identify and characterise" instead of "detect and then characterise"*

**Author response:** done (line 71).

*Reviewer: Line 81: Add "e.g.," before references, since there are lots of others that discuss aspects of this.*

**Author response:** done (line 74)

*Reviewer: Lines 84–85: Suggest "This initial set of metrics was included based on, first, ... and, second, the frequency of their usage in the literature."*

**Author response:** done (lines 76-78).

*Reviewer: Line 86: "of" should be "in".*

**Author response:** done (line 78).

*Reviewer: Line 90: "which" should be "that", and, to be consistent with the "single value" in line 88, "and" would need to be "or" (see comment above re the description of "jet statistics").*

**Author response:** we have replaced this with "and/or" (line 75).

*Reviewer: Line 107: Delete "any"*

**Author response:** done (removed from line 98).

*Reviewer: Line 110: Again, " 'latitude' and 'speed' " is not a "single" metric!*
**Author response:** We have changed to individual to be consistent with other references to jet statistics (line 100).

*Reviewer: Line 113: "While each jet statistic"*
**Author response:** done (line 103).

*Reviewer: Line 193: Replace "which introduces a physical-based" with "by introducing a physically-based".*
**Author response:** done (line 178).

*Reviewer: Line 210: Suggest "...due in part to…" (since this is by no means the only advantage of using xarray!!)*
**Author response:** Thank you, we have updated the text (lines 196-197).

*Reviewer: Line 224: Replace "nor" with "and / or" and delete comma.*
**Author response:** done (line 212).

*Reviewer: Line 225: This is a less-than-obvious case, but needs to be either "methods' docstrings" or "method's docstring".*
**Author response:** done, we went for 'method's docstring' (line 213).

*Reviewer: Line 228: Add comma after "i.e.,"*
**Author response:** done (line 216), and also changed on line 218.

*Reviewer: Line 235: Delete "of" at end of line.*
**Author response:** changed this to '...made a note of…' (line 223).

*Reviewer: Line 257: "figure" should be "Figure".*
**Author response:** done (line 245).

*Reviewer: Line 259: "rework and refactor" seems a bit redundant (since refactoring is a kind of reworking).*
**Author response:** changed to just 'refactor' (line 247).

*Reviewer: Lines 263–264: Now you are using "refactor" synonymously with "debug" (or "troubleshoot" or whatever term you prefer to use), whereas "refactor" is defined as "to improve internal code by making many small changes without altering the code's external behavior", which clearly implies that the code already produces the desired result. Also, the sentence structure has errors here, disregarding any content changes, it should be: "After which we either refactor the method further if it fails the validation, or write unit tests, finish the documentation, and integrate the metric into the jsmetrics package if it succeeds."*
**Author response:** We agree and have changed 'refactor' to 'debug', as this is closer to our meaning. Also, we have made the changes you suggest and the sentence now reads:
*"After which we either debug the method further if it fails the validation, or write unit tests, finish the documentation, and integrate the metric into the jsmetrics package if it succeeds"* (lines 251-252).

*Reviewer: Line 279: Should be "...from the ERA5…"*
**Author response:** Thanks, we have updated the text (line 266).

**Reviewer:** *Line 286: "is" should be "are" ("data" is plural); also, "details" (which, grammatically, should be "detail" since "data" is plural) isn't the right word here – perhaps use "comprise" or "consist of".*
**Author response:** Thank you, we have changed the text and we use 'consist of' (line 272).

**Reviewer:** *Line 326: Change "We hope to express that" to something like "The above example demonstrates that…" or "We hope the above example demonstrates that…" (I personally would leave out "We hope" since if you are publishing it you should express confidence in your results.)*
**Author response:** Thanks for your suggestion, we have updated the text to "The above example demonstrates that…" (line 299).

**Reviewer:** *Line 329: There should not be a comma.*
**Author response:** We have removed the comma after 'globe…' (line 301).

**Reviewer:** *Lines 333–334, and succeeding use: You use "North American Cold Wave" and other times "Texas Cold Wave". From what I've seen this event has been most frequently called the "Great Plains Cold Air Outbreak". Whichever term you choose, pick one and only one. (I also see no reason why it needs to be italicised.)*
**Author response:** Thanks for the suggestion, we have removed the italics and changed all mentions of this event to North American Cold Wave in the text.

**Reviewer:** *Line 334: Any of "...between 6 and 21 February…", "...from 6 through 21 February…", or "...from the 6th through the 21st of February…" would be correct (I favor the first as being most concise; whichever you use, try to be consistent in succeeding date range references).*
**Author response:** We have now chosen a consistent date range reference and changed all references to the same style. The style is like: *between 1200 UTC on 6th February and 1200 UTC on 21st February*. We will discuss further, as required, with the copy editor so it conforms to Copernicus style.

**Reviewer:** *Line 338: Delete colon after "levels".*
**Author response:** done (line 309).

**Reviewer:** *Line 341: Suggest "...some of the methods…"*
**Author response:** done (line 312).

**Reviewer:** *Line 343: "...of the figures…"*
**Author response:** done (lines 313-314).

**Reviewer:** *Line 353: "P13" should be "PO13".*
**Author response:** done (line 318).

**Reviewer:** *Line 360: Reword / correct: "...those cores; otherwise these jet cores in the same region will be considered part of the same core, at the location of the largest of the local wind speed maxima."*
**Author response:** This has been corrected (line 325).

**Reviewer:** *Line 364: "...cores in each may…"*
**Author response:** done (line 327).

**Reviewer:** *Figure 4. Say in the caption what exactly the "Standard North Pacific Region" is.*

**Author response:** Thank you for the suggestion, this has now been included.

*Reviewer: Lines 373, 375, 376: See comment re lines 333–334 just above.*
**Author response:** Thanks for the suggestion, we have removed the italics and changed all mentions of this event to North American Cold Wave in the text.

*Reviewer: Line 397: "figure" should be "figures"*
**Author response:** done (line 360).

*Reviewer: Line 403: Should be "...strongest and most variable…"*
**Author response:** changed (line 365).

*Reviewer: Line 418: Change "which all centre" to "all centred on"*
**Author response:** done (line 378).

*Reviewer: Line 425: Delete "to" and delete comma after "that".*
**Author response:** done (line 384).

*Reviewer: Line 436: "which" should be "that".*
**Author response:** done (line 395).

*Reviewer: Line 462: Need a comma after "metric".*
**Author response:** done (line 422).

*Reviewer: Line 482: Should be "...inputs, i.e., with…"*
**Author response:** done (line 442).

*Reviewer: Lines 490–494: The sentence structure here has problems, and the sentence is too long and complex to follow clearly. Suggest breaking it up into two or more sentences and restructuring.*
**Author response:** Thanks for the suggestion, we have broken this down into three sentences:

*"Finally, we note that some metrics may be too complex for the remit of this package (e.g. Kern et al., 2018; Kern and Westermann, 2019; Bösiger et al., 2022). When developing the package, we avoided metrics that use variables describing different aspects of the upper-level flow synonymous with (characteristics of) jet streams, such as wind shear (e.g. Lee et al., 2019) and magnitude of atmospheric waves (e.g. Chemke and Ming, 2020). Similarly, we did not include any potential metrics that require a training element to run and those that are currently very computationally expensive(e.g. Limbach et al., 2012; Molnos et al., 2017)"* (lines 450-455).

*Reviewer: Line 506: "which" should be "that".*
**Author response:** done (line 467).

*Reviewer: Line 511: "data is" should be "data are".*
**Author response:** done (line 472).

*Reviewer: (Note that I also found a number of typos and small errors similar to these in the online documentation, so would suggest more careful proofreading of that.)*
**Author response:** Thank you, we have also found a few and correct them since your review. We will also try to continually update the online docs with each iteration of the software.